# Large Language Model-driven Large Neighborhood Search for Large-Scale MILP Problems

**Huigen Ye** [1 2]  **Hua Xu** [1 2]  **An Yan** [1 2]  **Yaoyang Cheng** [1 2]

## Abstract

Large Neighborhood Search (LNS) is a widely used method for solving large-scale Mixed Integer Linear Programming (MILP) problems. The effectiveness of LNS crucially depends on the choice of the search neighborhood. However, existing strategies either rely on expert knowledge or computationally expensive Machine Learning (ML) approaches, both of which struggle to scale effectively for large problems. To address this, we propose LLM-LNS, a novel Large Language Model (LLM)-driven LNS framework for large-scale MILP problems. Our approach introduces a dual-layer self-evolutionary LLM agent to automate neighborhood selection, discovering effective strategies with scant small-scale training data that generalize well to large-scale MILPs. The inner layer evolves heuristic strategies to ensure convergence, while the outer layer evolves evolutionary prompt strategies to maintain diversity. Experimental results demonstrate that the proposed dual-layer agent outperforms state-of-the-art agents such as FunSearch and EOH. Furthermore, the full LLM-LNS framework surpasses manually designed LNS algorithms like ACP, ML-based LNS methods like CL-LNS, and large-scale solvers such as Gurobi and SCIP. It also achieves superior performance compared to advanced ML-based MILP optimization frameworks like GNN&GBDT and Light-MILPopt, further validating the effectiveness of our approach.

## 1. Introduction

Mixed Integer Linear Programming (MILP) is a versatile and widely used mathematical framework for solving complex optimization problems across various domains, including transportation management (Klanšek, 2015), bin packing (Fleszar, 2022), and production planning (Adrio et al., 2023). MILPs are challenging to solve efficiently due to their NP-hard nature (Kim et al., 2021) and the exponential growth of the search space as problem size increases (Vázquez et al., 2018). To address these challenges, researchers have developed two primary approaches (Zhang et al., 2023): exact algorithms, such as branch-and-bound, and heuristic-based approximation methods.

While exact algorithms like branch-and-bound (Boyd & Mattingley, 2007; Morrison et al., 2016) are effective for small to medium-sized problems, they struggle with the computational demands of larger instances. This has led to the rise of heuristic methods, particularly Large Neighborhood Search (LNS) (Ahuja et al., 2002; Mara et al., 2022), which iteratively improves solutions by destroying and repairing parts of the current solution, allowing for exploration of large neighborhoods without full re-optimization (Song et al., 2020; Ye et al., 2023a). However, LNS performance depends heavily on neighborhood selection, which is often hand-crafted and requires significant domain expertise. Designing these operators can be labor-intensive and prone to *cold-start issues*, where limited prior knowledge is available to guide the search (Zhang et al., 2023).

In recent years, machine learning (ML) techniques, including reinforcement learning (Wu et al., 2021; Song et al., 2020) and imitation learning (Sonnerat et al., 2021; Nair et al., 2020), have been applied to automate the design of neighborhood selection strategies. These methods aim to learn heuristic strategies from training datasets, reducing reliance on expert knowledge and allowing the algorithms to adapt to new, homogeneous instances. However, ML-based LNS approaches come with their own challenges. For reinforcement learning, *slow convergence* is a common issue (Beggs, 2005), particularly in large-scale MILP problems, due to the vast search space and the need for extensive exploration before identifying effective strategies. On the other hand, imitation learning requires large amounts of high-quality, labeled data, which can be *computationally expensive* to generate using expert algorithms (Huang et al., 2023b). As a result, both hand-crafted and ML-based methods struggle to efficiently solve large-scale MILP problems.

[1]Department of Computer Science and Technology, Tsinghua University, Beijing 100084, China [2]Beijing National Research Center for Information Science and Technology, Beijing 100084, China. Correspondence to: Hua Xu <xuhua@tsinghua.edu.cn>.

*Proceedings of the 42nd International Conference on Machine Learning*, Vancouver, Canada. PMLR 267, 2025. Copyright 2025 by the author(s).

The rise of Large Language Models (LLMs) offers a promising solution to these challenges. Unlike traditional hand-crafted methods, LLMs come pretrained with vast general knowledge, allowing them to reason about complex tasks and learn problem structures with minimal training data, thus avoiding *cold-start issues*. Additionally, LLMs can adapt to new problems through interactive reasoning, reducing the need for extensive exploration and addressing the *slow convergence* of reinforcement learning. Furthermore, LLMs can dynamically generate heuristic strategies without relying on large labeled datasets, which significantly reduces the *computational overhead* typically associated with imitation learning (Yang et al., 2024; Lange et al., 2024). While LLMs have shown potential in generating strategies for combinatorial optimization problems(Ye et al., 2024; Elhenawy et al., 2024), they often lack the problem-specific refinement needed to produce efficient heuristics without additional guidance (Plaat et al., 2024). Approaches like FunSearch (Romera-Paredes et al., 2024) and Evolution of Heuristic (EOH) (Liu et al., 2024) combine LLMs with evolutionary algorithms (Simon, 2013), but rely on fixed strategies, limiting solution diversity and leading to poor convergence due to insufficient directionality. This underscores the need for a more adaptive framework to fully harness LLMs for large-scale MILP problems.

In this paper, we propose LLM-LNS, a novel Large Language Model-driven Large Neighborhood Search framework designed specifically for solving large-scale MILP problems, which can discover effective neighborhood selection strategies for LNS with scant small-scale training data that generalize well to large-scale MILPs. The code of LLM-LNS is open-sourced at https://github.com/thuiar/LLM-LNS. Our key innovations are as follows:

- **Dual-layer Self-evolutionary LLM Agent**: We propose a novel LLM agent with a dual-layer self-evolutionary mechanism for automatically generating heuristic strategies. The inner layer evolves both thoughts and code representations of heuristic strategies, ensuring convergence, while the outer layer evolves evolutionary prompt strategies to maintain diversity, preventing the search process from getting trapped in local optima.

- **Differential Memory for Directional Evolution**: We introduce differential evolution in the agent to guide both crossover and variation. By feeding the fitness values of parent strategies back into the LLM, we leverage its memory to learn how to evolve from less effective to more effective strategies. This feedback mechanism enables the LLM to act as an optimizer, identifying promising directions and leading to more efficient improvements.

- **Application to Neighborhood Selection in LNS**: We

apply the proposed dual-layer LLM agent to the neighborhood selection strategy generation in LNS. By utilizing only a small amount of training data from small-scale problems, the LLM agent can discover new neighborhood selection strategies that generalize well to large-scale MILP problems.

- **Comprehensive Experimental Validation**: We validate the effectiveness of our proposed LLM-LNS at two levels. First, we test its agent's performance on heuristic generation tasks of combinatorial optimization problems, demonstrating its superiority over state-of-the-art methods such as FunSearch and EOH. Second, we evaluate its performance on large-scale MILP problems, where it outperforms traditional LNS methods, ML-based LNS methods, and leading solvers. Furthermore, our proposed LLM-LNS surpasses modern ML-based optimization frameworks for large-scale MILP, confirming the effectiveness of our LLM-LNS in solving large-scale optimization problems.

## 2. Related Work

### 2.1. Mixed Integer Linear Programming

Mixed Integer Linear Programming (MILP) problems represent a class of combinatorial optimization problems characterized by a linear objective function subject to a set of linear constraints, where some or all decision variables are restricted to integer values. An MILP can be defined as follows:

$$\min_x c^T x, \text{ s.t. } Ax \leq b, l \leq x \leq u, x_i \in \mathbb{Z}, i \in \mathcal{I}, \quad (1)$$

where $x$ represents the decision variables, with $n \in \mathbb{Z}$ denoting the dimensionality of the integer variables and $l, u, c \in \mathbb{R}^n$ corresponding to the lower bounds, upper bounds, and coefficients of the decision variables, respectively. The matrix $A \in \mathbb{R}^{m \times n}$ and the vector $b \in \mathbb{R}^m$ define the linear constraints of the problem. The set $\mathcal{I} \subseteq \{1, 2, \ldots, n\}$ denotes the indices of variables that are constrained to integer values. A feasible solution to the MILP problem satisfies all constraints, and the optimal solution minimizes the objective function value. (Artigues et al., 2015; Pisaruk, 2019)

### 2.2. Large Neighborhood Search

Large Neighborhood Search (LNS) is a widely used heuristic for solving MILP problems. It iteratively improves solutions by exploring predefined neighborhoods around a current solution. However, the effectiveness of LNS heavily relies on the neighborhood selection strategy, as poor choices can lead to stagnation in local optima.

Several approaches have been proposed to address this challenge. One common method is random-LNS (Song et al.,

2020), which randomly partitions integer variables into disjoint subsets and optimizes one subset in each iteration while fixing the others. However, random-LNS uses a fixed neighborhood size and overlooks correlations between decision variables, limiting its performance. To overcome these drawbacks, the Adaptive Constraint Partitioning (ACP) framework (Ye et al., 2023a) introduces a dynamic strategy that adjusts the neighborhood size, optimizing all decision variables associated with randomly selected constraints in each iteration. This ensures that highly correlated variables are optimized together, improving performance. Similar strategies have been explored in other works (Huang et al., 2023a; Han et al., 2023), but they still rely on manually designed heuristics, requiring expert knowledge and lacking adaptability to new problem instances.

To address this limitation, machine learning methods have been applied to automate neighborhood selection. Reinforcement learning (RL) approaches define reward functions based on solution improvements, allowing models to learn promising neighborhoods through interaction with the problem (Wu et al., 2021; Song et al., 2020; Nair et al., 2020). Imitation learning, on the other hand, uses large amount of large-scale sampling (Huang et al., 2023b; Zhou et al., 2023) or expert algorithms (Sonnerat et al., 2021) to guide the selection process. While these techniques reduce reliance on handcrafted strategies, RL struggles with convergence in large-scale MILP problems, and imitation learning requires extensive sampling, making it computationally expensive. This highlights the need for more efficient, automatically designed neighborhood selection strategies.

### 2.3. Large Language Model for Heuristic Design

The rise of Large Language Models (LLMs) has opened new possibilities for generating heuristic strategies to solve combinatorial optimization problems (Yang et al., 2024; Lange et al., 2024). LLMs excel at generating high-level ideas and reasoning over complex tasks, but they often lack problem-specific knowledge, limiting their ability to create effective heuristics without additional guidance (Plaat et al., 2024). To overcome these limitations, recent works have integrated LLMs with evolutionary algorithms (EA) to iteratively refine heuristics.

FunSearch (Romera-Paredes et al., 2024) is a notable attempt that combines LLMs with evolutionary frameworks. FunSearch uses LLMs to generate functions, which are then evolved through an evolutionary search process. This approach has demonstrated success in outperforming handcrafted algorithms on specific optimization problems. However, FunSearch is computationally expensive, often requiring millions of LLM queries to identify effective heuristic functions, which limits its practicality in many real-world applications. A more recent approach, Evolution of Heuristic

*Table 1.* Comparison of Features Between Difference Methods

|  | FunSearch | EOH | **LLM-LNS** |
| --- | --- | --- | --- |
| Heuristic Evolution | ✓ | ✓ | ✓ |
| Thought Evolution | ✗ | ✓ | ✓ |
| Prompt Evolution | ✗ | ✗ | ✓ |
| Directional Evolution | ✗ | ✗ | ✓ |

(EOH) (Liu et al., 2024), builds on the strengths of LLMs and evolutionary computation while addressing some of FunSearch's limitations. EOH introduces a novel evolutionary paradigm where heuristics, represented as natural language "thoughts," are translated into executable code by LLMs. These thoughts and their corresponding code are evolved within an EA framework, enabling the efficient generation of high-performance heuristics. As shown in Table 1, while FunSearch and EOH have advanced the integration of LLMs with evolutionary algorithms, they still have limitations. All methods focus on *Heuristic Evolution* for generating strategies, but FunSearch evolves only at the code level and lacks *Thought Evolution*. Meanwhile, EOH incorporates Thought Evolution but uses fixed evolutionary strategies, lacking *Prompt Evolution* to enhance solution diversity. Additionally, both methods lack *Directional Evolution*, where crossover operations are guided by differential memory to improve efficiency and adaptability. These limitations reduce their ability to guide the search effectively, often leading to premature convergence. These challenges highlight the need for more adaptive frameworks to fully harness LLMs in large-scale optimization tasks.

## 3. Method

In this section, we introduce LLM-LNS, a Large Language Model-driven Large Neighborhood Search framework designed to solve large-scale MILP problems. As shown in Figure 1, the framework is composed of two main components: a **Dual-layer Self-evolutionary LLM Agent** and a **Adaptive Large Neighborhood Search** process. For the framework's detailed pseudocode, see Appendix B.1.

### 3.1. Dual-layer Self-evolutionary LLM Agent

The Dual-layer Self-evolutionary LLM Agent is the core component of our framework, responsible for generating and evolving heuristic and prompt strategies. The **Dual-layer Self-evolutionary Structure** consists of an Inner Layer that evolves heuristic strategies to accelerate convergence, and an Outer Layer that evolves evolutionary prompt strategies to enhance diversity in heuristic generation. Another key innovation is the incorporation of **Differential Memory for Directional Evolution**, which accelerates convergence by learning the direction of improvement from less effective strategy to better ones. Together, these innovations ensure a balance between exploration and exploitation, significantly improving the efficiency and preventing stag-

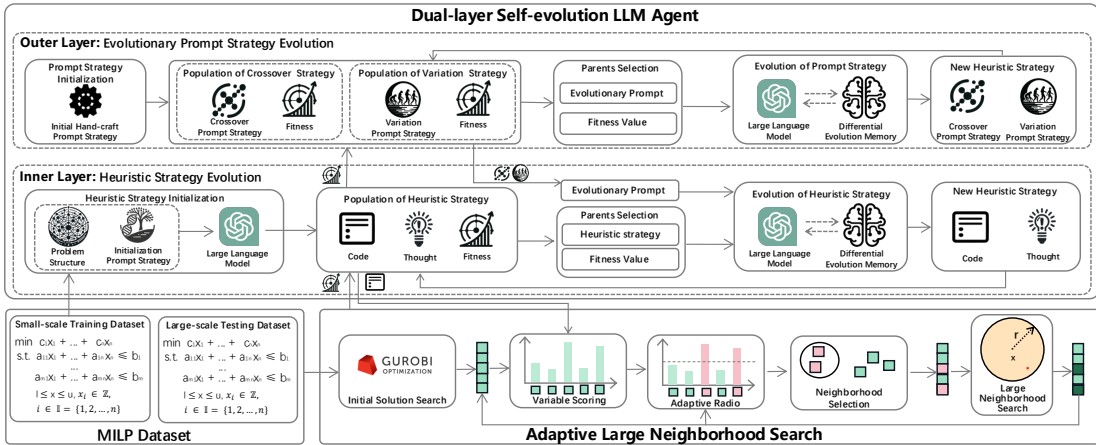

*Figure 1.* An overview of the proposed LLM-LNS framework. The framework consists of a dual-layer self-evolutionary LLM agent for solving large-scale MILP problems. In the outer layer, evolutionary prompt strategies are generated and passed to the inner layer, where heuristic strategies are evolved. A differential memory mechanism uses fitness feedback to refine these strategies across iterations. The refined strategies are fed into the Adaptive Large Neighborhood Search process, which iteratively improves solutions.

nation in local optima.

### 3.1.1. DUAL-LAYER SELF-EVOLUTIONARY STRUCTURE

The Dual-layer Self-evolutionary Structure is the core component of the LLM-LNS framework. It is designed to evolve both evolutionary prompt strategies and heuristic strategies in a synergistic manner, leveraging LLMs for automated heuristic design and refinement. This dual-layered structure mimics the heuristic development process of human experts, ensuring a balance between exploration and exploitation throughout the search process.

**Inner Layer: Heuristic Strategy Evolution.** The Inner Layer focuses on evolving heuristic strategies, which consist of both natural thought and corresponding code implementations, with an emphasis on *convergence*. Key aspects of Inner Layer, as illustrated in Figure 1 , include:

- *Initialization of Heuristic Strategies*: The initial set of heuristics is generated by feeding the structural information from small-scale training problems, along with an initialization prompt strategy, into the LLM. This produces the first generation of heuristic strategies.

- *Evolution of Heuristic Strategies*: In each generation, new heuristic strategies are evolved by selecting parent strategies from the current heuristic population. As detailed in Appendix B.3, strategies with higher fitness values are more likely to be selected as parents. These parents are then combined with evolutionary strategies, selected from the Outer Layer's population of prompt strategies (e.g., crossover or variation prompts), to guide the LLM in generating offspring strategies.

- *Evaluation and Final Selection*: After new heuristic strategies are generated, they are evaluated by integrating them into the Adaptive Large Neighborhood Search process, where each heuristic is applied to solve small-scale instances from the training dataset. The performance of each strategy is measured by its objective function value, which serves as its fitness score. After multiple iterations of evolution and evaluation, the best-performing heuristic strategies are identified based on their fitness.

**Outer Layer: Evolutionary Prompt Strategy Evolution.** The Outer Layer focuses on evolving evolutionary prompt strategies, which guide the LLM in generating new heuristic strategies. The emphasis in this layer is on *exploration* to maintain diversity and prevent premature convergence in the heuristic strategy population. The key stages of Outer Layer, as illustrated in Figure 1, include:

- *Initialization of Prompt Strategies*: The initial set of evolutionary prompt strategies is handcrafted and designed to perform basic crossover and variation operations, instructing the LLM on how to combine or modify existing heuristic strategies in the inner layer.

- *Evolution of Prompt Strategies*: As the evolution progresses, more complex prompt strategies are introduced to address stagnation in the heuristic population. Specifically, if the top-$l$ individuals in the heuristic population remain unchanged for $t$ consecutive generations, we infer that the evolution may have converged to a local optimum. This triggers the evolution of new prompt strategies. This systematic evolution of prompt

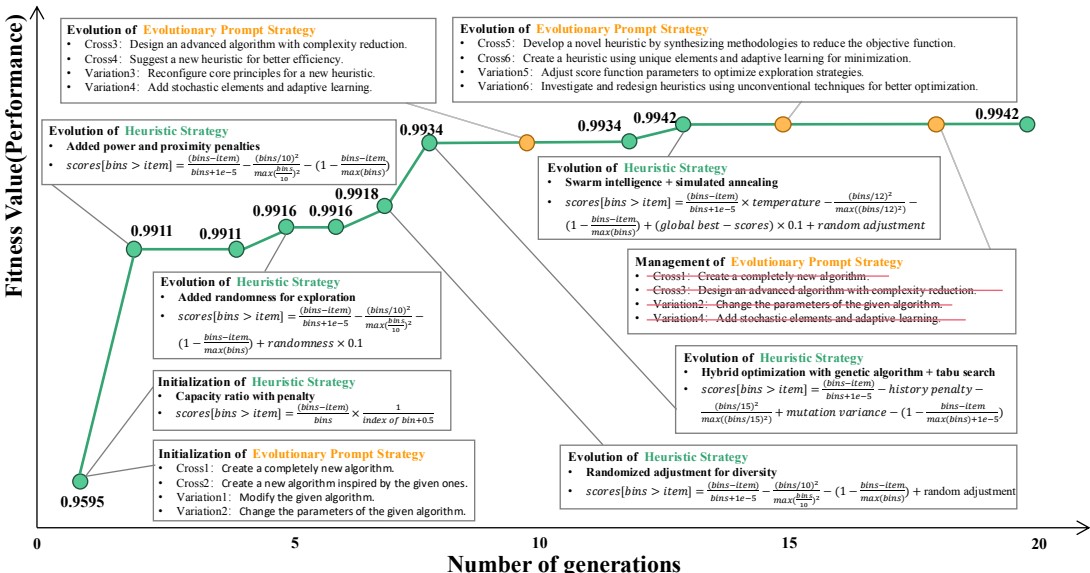

*Figure 2.* Evolution of Dual-layer Self-evolutionary LLM Agent for online bin packing. We outline the key thoughts of the best heuristics produced in some generations during the evolution of heuristic strategies. Additionally, we highlight the evolution of evolutionary prompt strategies, which dynamically adapt the prompt strategies to guide the LLM in generating more effective and diverse heuristics.

strategies helps ensure that the heuristic population does not get trapped in local optima.

- *Evaluation and Management of Prompt Strategies*: To ensure the efficiency and effectiveness of the prompt strategy population, each prompt strategy is evaluated based on the performance of the heuristic strategies it generates. Specifically, for each prompt strategy, the top-$k$ performing heuristic strategies it produces are tracked, and the average fitness score of these heuristics is used as the fitness score for the prompt strategy itself. This fitness-based evaluation allows us to manage the prompt population and control its size. As the number of prompt strategies increases over generations, under-performing strategies are pruned to prevent excessive growth and focus on the most effective strategies. This pruning process ensures that only the most effective prompt strategies continue to evolve, maintaining both diversity and efficiency in the evolutionary process. For parameter details, see Appendix C.

The synergy between the **Inner Layer** and **Outer Layer** drives rapid evolution of effective heuristics and novel evolutionary prompt strategies, as shown in Figure 2. Early generations focus on basic principles, but with the introduction of advanced prompt strategies, such as complexity reduction and adaptive learning, the system quickly adapts to overcome local optima. Notably, the sharp performance improvements between generations 5 to 15 demonstrate the framework's ability to autonomously discover and refine creative strategies, leading to continuous enhancements

in heuristic performance. This dual-layered approach ensures efficient exploration and exploitation, enabling the LLM-LNS framework to tackle large-scale problems with minimal human intervention. For a more detailed example of the evolution process, please refer to Appendix B.2.

### 3.1.2. DIFFERENTIAL MEMORY FOR DIRECTIONAL EVOLUTION

In our Dual-layer Self-evolutionary LLM Agent, both heuristic strategies and evolutionary prompt strategies evolve through a process that incorporates *Differential Memory for Directional Evolution*. This mechanism allows the LLM to leverage the fitness history of strategies, learning from the differences between higher- and lower-performing strategies to guide the generation of improved candidates. Differential memory enables the LLM to act as both a generator and an optimizer, dynamically refining strategies over successive generations.

At each generation $t$, the LLM is provided with a set of $m$ *strategy-thought-fitness* tuples:

$$S^{(t)} = \{\langle H_i^{(t)}, \text{thought}_i, f(H_i^{(t)})\rangle\}_{i=1}^{m}, \qquad (2)$$

where $H_i^{(t)}$ represents the $i$-th parent heuristic strategy selected for this generation, $\text{thought}_i$ is its corresponding natural language description, and $f(H_i^{(t)})$ is its fitness score. The size of $S^{(t)}$ is $m$, which is a predefined parameter representing the number of parent strategies used in a single evolutionary operation. These tuples encapsulate both the structural and performance information of the selected par-

ent strategies, providing the necessary context for generating offspring strategies.

To generate the next generation of strategies $H^{(t+1)}$, the LLM employs a *meta-prompt* $p_{\text{meta}}$, which combines two key components: a directive $p_{\text{learn}}$ that instructs the LLM to learn from the differences between higher- and lower-performing strategies, emphasizing traits that contribute to higher fitness; and an *evolutionary prompt strategy* $p_{\text{evo}}$, provided by the Outer Layer, which specifies the goals and rules for the evolutionary operation, such as crossover, mutation, or hybrid operations. The generation process can be formalized as:

$$H_i^{(t+1)} = \mathcal{M}(p_{\text{meta}} \| S^{(t)}), \tag{3}$$

where $\mathcal{M}$ is the LLM model, $p_{\text{meta}} = \langle p_{\text{learn}}, p_{\text{evo}} \rangle$ is the meta-prompt, and $S^{(t)}$ represents the strategy-thought-fitness tuples from the current generation. By integrating these components, the LLM generates new strategies $H^{(t+1)}$ that are informed by past evolutionary performance and aligned with the objectives defined by the Outer Layer. This iterative feedback-refinement loop ensures that the LLM dynamically balances exploration and exploitation. Differential memory accumulates across generations, enabling the LLM to focus on areas of the search space that demonstrate promise while avoiding stagnation in local optima. The result is an increasingly proficient evolution process, accelerating convergence toward optimal solutions while maintaining population diversity.

### 3.2. Adaptive Large Neighborhood Search

Adaptive Large Neighborhood Search (ALNS) dynamically adjusts neighborhood size and leverages the Dual-layer Self-evolutionary LLM Agent for variable scoring and selection. At each iteration $t$, the LLM agent computes scores $s_i^{(t)}$ for decision variables $x_i$ based on their potential to improve the objective value. The top-$k$ variables are selected to form the neighborhood $\mathcal{N}^{(t)}$:

$$\mathcal{N}^{(t)} = \{x_i \mid \text{rank}(s_i^{(t)}) \leq k\}, \tag{4}$$

where $\mathcal{N}^{(t)}$ is the neighborhood at iteration $t$, and $k$ is the current neighborhood size. A subproblem is then solved within $\mathcal{N}^{(t)}$, and the solution $\mathbf{x}$ is updated if an improvement is found.

The neighborhood size $k$ is adaptively adjusted based on search progress. If the improvement in the objective value falls below a threshold $\epsilon$ for $p$ consecutive iterations, $k$ is expanded to explore a broader search space $k \leftarrow \min(k_{\max}, k + \lceil u\% \cdot n \rceil)$, where $u\%$ is the adjustment rate and $n$ is the total number of decision variables. Conversely, if the time spent solving subproblems within the neighborhood exceeds a predefined limit, $k$ is reduced to focus on a smaller subset of variables $k \leftarrow \max(k_{\min}, k - \lceil u\% \cdot n \rceil)$.

The key innovation of ALNS lies in the use of the LLM agent to generalize variable selection strategies. Trained on small-scale MILP problems, the LLM agent learns to rank variables based on their impact on the objective function, enabling the generalization of these strategies to larger, more complex problems. This transfer of knowledge ensures that neighborhood selection is both adaptive and intelligent, dynamically balancing exploration and exploitation. By focusing computational resources on the most promising regions of the solution space, ALNS efficiently navigates the vast search space of large-scale MILPs. For a detailed description of the process and the corresponding pseudocode, please refer to Appendix B.4.

## 4. Experiment

To validate the effectiveness of the proposed LLM-LNS framework, we conduct two sets of experiments. First, we evaluate our proposed Dual-layer Self-evolutionary LLM Agent on heuristic generation tasks for combinatorial optimization problems, comparing it against methods like FunSearch (Romera-Paredes et al., 2024) and EOH (Liu et al., 2024). Second, we assess the full LLM-LNS framework on large-scale MILP problems, where it is compared against traditional LNS methods (e.g., ACP (Ye et al., 2023a)), ML-based LNS approaches (e.g., CL-LNS (Huang et al., 2023b)), the SOTA solvers like Gurobi (Gurobi Optimization, LLC, 2023) and SCIP (Maher et al., 2016), and modern ML optimization frameworks such as GNN&GBDT (Ye et al., 2023c) and Light-MILPopt (Ye et al., 2023b). More experimental results and details are provided in the Appendices C to F.

### 4.1. Heuristic Generation for Combinatorial Optimization Problems

In this section, we evaluate the performance of the Dual-layer Self-evolutionary LLM Agent in generating heuristic strategies for well-known combinatorial optimization problems. We focus on two widely studied problems: Online Bin Packing (Seiden, 2002) and the Traveling Salesman Problem (TSP) (Hoffman et al., 2013). Our method is compared against several hand-crafted heuristics, state-of-the-art ML-based methods, and other automatically designed heuristics.

#### 4.1.1. ONLINE BIN PACKING

The objective of the Online Bin Packing problem is to allocate a collection of items into the fewest possible bins of fixed capacity. We follow the experimental setup from Romera-Paredes et al. (2024), using Weibull distribution instances with varying numbers of items (1k to 10k) and bin capacities (100 and 500). The performance of each method is measured by the fraction of excess bins used, where lower values indicate better performance. We compare our method

*Table 2.* **Online Bin Packing Heuristic Comparison.** This table compares the performance of various bin packing heuristics based on the fraction of excess bins (lower values indicate better performance) across different Weibull distribution instances.

|  | 1k_C100 | 5k_C100 | 10k_C100 | 1k_C500 | 5k_C500 | 10k_C500 | Avg |
|---|---|---|---|---|---|---|---|
| First Fit | 5.32% | 4.40% | 4.44% | 4.97% | 4.27% | 4.28% | 4.61% |
| Best Fit | 4.87% | 4.08% | 4.09% | 4.50% | 3.91% | 3.95% | 4.23% |
| FunSearch | 3.78% | **0.80%** | **0.33%** | 6.75% | 1.47% | 0.74% | 2.31% |
| EOH | 4.48% | 0.88% | 0.83% | 4.32% | 1.06% | 0.97% | 2.09% |
| Ours | **3.58%** | 0.85% | 0.41% | **3.67%** | **0.82%** | **0.42%** | **1.63%** |

*Table 3.* **Traveling Salesman Problems Heuristic Performance Evaluation.** This table provides a comparison of the relative distance to the best-known solutions for different routing heuristics (lower values indicate better performance) on a subset of TSPLib benchmark instances.

|  | rd100 | pr124 | bier127 | kroA150 | u159 | kroB200 | Avg |
|---|---|---|---|---|---|---|---|
| NI | 19.91% | 15.50% | 23.21% | 18.17% | 23.59% | 24.10% | 20.75% |
| FI | 9.38% | 4.43% | 8.04% | 8.54% | 11.15% | 7.54% | 8.18% |
| Or-Tools | **0.01%** | 0.55% | 0.66% | 0.02% | 1.75% | 2.57% | 0.93% |
| AM | 3.41% | 3.68% | 5.91% | 3.78% | 7.55% | 7.11% | 5.24% |
| POMO | **0.01%** | 0.60% | 13.72% | 0.70% | 0.95% | 1.58% | 2.93% |
| LEHD | **0.01%** | 1.11% | 4.76% | 1.40% | 1.13% | 0.64% | 1.51% |
| EOH | **0.01%** | **0.00%** | 0.42% | 0.29% | **-0.01%** | **0.26%** | 0.16% |
| Ours | **0.01%** | **0.00%** | **0.01%** | **0.00%** | **-0.01%** | 0.44% | **0.08%** |

against several baselines, including hand-crafted heuristics First Fit (Tang et al., 2016) and Best Fit (Shor, 1991), which are widely used in practice, as well as automatically generated heuristics FunSearch (Romera-Paredes et al., 2024) and EOH (Liu et al., 2024), which represent state-of-the-art approaches.

As shown in Table 2, our method consistently achieves the best performance across different problem sizes and capacities, with an average excess bin fraction of 1.63%, outperforming both hand-crafted heuristics and automatically generated methods. In particular, our approach excels on the 10k items, capacity 500 instance, achieving a fraction of excess bins of 0.42%, outperforming FunSearch (0.74%) and EOH (0.97%), highlighting the strong scalability and generalization ability of our method, making it particularly effective in handling large-scale, high-capacity scenarios.

### 4.1.2. TRAVELING SALESMAN PROBLEM

The Traveling Salesman Problem (TSP) is a classic combinatorial optimization problem where the goal is to find the shortest route that visits all given locations exactly once. We evaluate our method on a subset of TSPLib benchmark instances (Reinelt, 1991), with performance measured by the relative distance to the best-known solutions (lower values indicate better performance). We compare our method against two types of baselines: hand-crafted heuristics and AI-generated heuristics. The hand-crafted heuristics include Nearest Insertion (NI) and Farthest Insertion (FI) (Rosenkrantz et al., 1977), two widely used constructive heuristics. We also include Google OR-Tools (Perron & Furnon), a popular solver, using its default settings and the recommended local search option. Beyond EOH (Liu et al., 2024), we compare against the Attention Model (AM) (Kool et al., 2018), POMO (Kwon et al., 2020), and LEHD (Luo

et al., 2023), all of which are ML-based methods.

As shown in Table 3, our method achieves the best average performance with a 0.08% gap to the best-known solutions, outperforming both hand-crafted heuristics and neural network-based methods. Notably, on the bier127 instance, our method achieves a relative distance of just 0.01% to the best-known solution, significantly outperforming EOH (0.42%) and other baselines, including LEHD (4.76%) and AM (5.91%). This substantial improvement highlights the effectiveness of our approach in solving challenging instances of the TSP.

It is important to note that both the Online Bin Packing and TSP problems use the same GPT-4o-mini LLM, with identical settings: 20 iterations and a population size of 20 for Online Bin Packing, and 10 for the TSP problem. Despite these identical settings, our method consistently outperforms EOH in both problems, showcasing the superior efficiency of the dual-layer self-evolutionary mechanism in exploring the solution space. This mechanism allows our method to dynamically adapt and refine solutions, resulting in better overall performance with the same computational resources. These results underscore the robustness and scalability of our approach, offering a promising direction for solving large-scale combinatorial optimization problems using LLMs.

### 4.2. Performance of LLM-LNS on Large-Scale MILP Problems

To validate the effectiveness of the proposed LLM-LNS framework for large-scale MILP problems, we evaluate its performance on four widely-used benchmark datasets: Set Covering (SC) (Caprara et al., 2000), Minimum Vertex Cover (MVC) (Dinur & Safra, 2005), Maximum Indepen-

*Table 4.* **Comparison of objective values on large-scale MILP instances across different methods.** For each instance, the best-performing objective value is highlighted in bold. The - symbol indicates that the method was unable to generate samples for any instance within 30,000 seconds, while * indicates that the GNN&GBDT framework could not solve the MILP problem.

| | $SC_1$(Min) | $SC_2$(Min) | $MVC_1$(Min) | $MVC_2$(Min) | $MIS_1$(Max) | $MIS_2$(Max) | $MIKS_1$(Max) | $MIKS_2$(Max) |
|---|---|---|---|---|---|---|---|---|
| Random-LNS | 16140.6 | 169417.5 | 27031.4 | 276467.5 | 22892.9 | 223748.6 | 36011.0 | 351964.2 |
| ACP | 17672.1 | 182359.4 | 26877.2 | 274013.3 | 23058.0 | 226498.2 | 34190.8 | 332235.6 |
| CL-LNS | - | - | 31285.0 | - | 15000.0 | - | - | - |
| Gurobi | 17934.5 | 320240.4 | 28151.3 | 283555.8 | 21789.0 | 216591.3 | 32960.0 | 329642.4 |
| SCIP | 25191.2 | 385708.4 | 31275.4 | 491042.9 | 18649.9 | 9104.3 | 29974.7 | 168289.9 |
| GNN&GBDT | 16728.8 | 252797.2 | 27107.9 | 271777.2 | 22795.7 | 227006.4 | * | * |
| Light-MILPOPT | 16108.1 | 160015.5 | 26950.7 | 269571.5 | 22966.5 | 230432.9 | 36125.5 | 362265.1 |
| LLM-LNS(Ours) | **15802.7** | **158878.9** | **26725.3** | **268033.7** | **23169.3** | **231636.9** | **36479.8** | **363749.5** |

dent Set (MIS) (Tarjan & Trojanowski, 1977), and Mixed Integer Knapsack Set (MIKS) (Atamtürk, 2003). Initially, LLM-LNS is trained on smalle-scale problems with tens of thousands of variables and constraints and then tested on large-scale instances with millions of variables and constraints to assess its scalability and generalization.

We compare LLM-LNS with several state-of-the-art baselines, including heuristic LNS methods like Random-LNS (Song et al., 2020), Adaptive Constraint Propagation (ACP) (Ye et al., 2023a), and the learning-based CL-LNS framework (Huang et al., 2023b). Additionally, we include traditional solvers like Gurobi (Gurobi Optimization, LLC, 2023) and SCIP (Maher et al., 2016), as well as modern ML-based frameworks such as GNN&GBDT (Ye et al., 2023c) and Light-MILPopt (Ye et al., 2023b). To ensure a fair comparison, Gurobi is used as the sub-solver in the neighborhood search step across all methods. For LLM-LNS, the neighborhood selection strategy is trained over 20 iterations on smaller problems before being applied to larger instances. Detailed results and discussions are provided in the Appendix F.

### 4.2.1. EFFECTIVENESS COMPARISON

In this section, we evaluate the effectiveness of LLM-LNS by comparing its performance with various state-of-the-art methods across different problem instances. To ensure a fair comparison, all methods are evaluated under the same computational time limit, and the final objective function values are used as the primary metric for comparison. This allows us to assess not only the solution quality but also the efficiency with which each method converges to the optimal or near-optimal solutions. The results, summarized in Table 4, show that LLM-LNS consistently outperforms traditional LNS-based heuristics and learning-based methods. Unlike hand-crafted LNS strategies, which are typically static and less effective as problem complexity increases, LLM-LNS dynamically adapts through its dual-layer self-evolutionary mechanism, enabling more efficient exploration of the solution space. Even compared to state-of-the-art learning-based LNS methods like CL-LNS, LLM-LNS demonstrates superior performance. Although CL-LNS represents one of the most advanced learning-based approaches, it often fails to

complete sampling within an acceptable time for large-scale instances, and even when results are obtained, the solution quality is significantly lower. This highlights the challenges faced by existing LNS-based methods when dealing with large and complex MILP problems, while underscoring the robustness and adaptability of LLM-LNS.

In addition, LLM-LNS shows a clear advantage over traditional solvers like Gurobi and SCIP, as well as learning-based methods such as GNN&GBDT and Light-MILPopt. While traditional solvers perform competitively on smaller instances, their performance degrades significantly as the problem size increases. Similarly, learning-based methods struggle with large-scale MILPs, finding it difficult to efficiently explore the exponentially growing solution space. In contrast, LLM-LNS consistently delivers superior results across both small and large-scale problems, offering a scalable and efficient solution. These findings suggest that LLM-LNS not only bridges the gap between traditional and learning-based methods, but also opens new avenues for scalable optimization in large-scale MILPs.

### 4.2.2. EFFICIENCY AND CONVERGENCE ANALYSIS

To further assess the efficiency and convergence behavior of LLM-LNS on large-scale MILP problems, we analyze both the objective-time trajectories and the primal integrals of all evaluated methods. The objective-time plots, shown in Figure 3, illustrate how quickly each method improves the solution quality over time, while the primal integral values, a common way in the MILP literature to quantify heuristic performance over time, summarized in Table 5, provide a holistic measure of convergence efficiency over the entire optimization period. Across all benchmark instances, LLM-LNS demonstrates the fastest convergence and most stable performance, consistently reaching high-quality solutions earlier than all baselines. This is particularly evident on large and complex instances like $SC_2$, $MVC_2$, and $MIKS_2$, where traditional solvers and learning-based methods either converge slowly or stagnate. LLM-LNS shows a sharp drop (for minimization) or rapid rise (for maximization) in the objective value early in the optimization process, indicating superior efficiency. The primal integral results (Table 5) further highlight its effectiveness, with LLM-LNS achieving

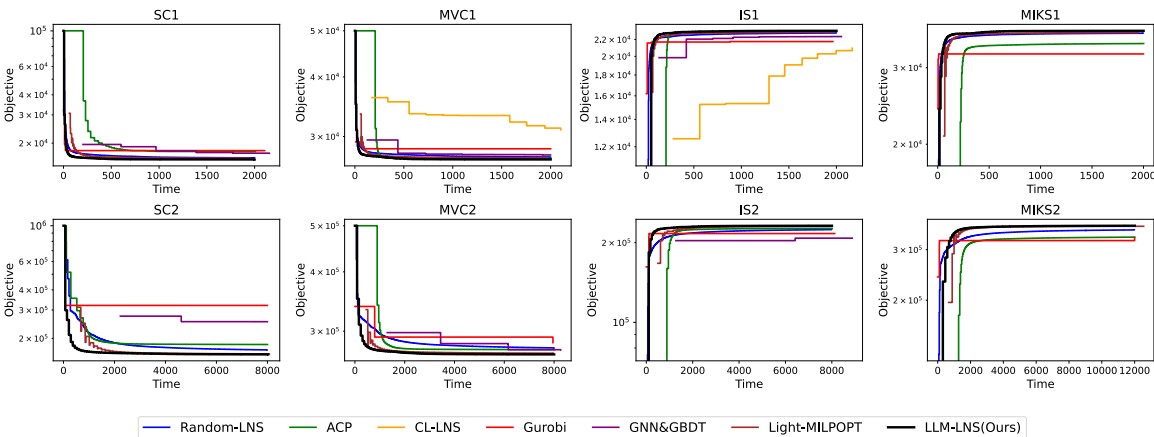

*Figure 3.* Convergence curves of objective values over time on large-scale MILP instances. Each subplot corresponds to one benchmark instance, illustrating how the objective value evolves over time for different methods using Gurobi as optimizer.

*Table 5.* **Comparison of primal integral on large-scale MILP instances across different methods.** For each instance, the best-performing primal integral is highlighted in bold.

| | $SC_1$(Min) | $SC_2$(Min) | $MVC_1$(Min) | $MVC_2$(Min) | $MIS_1$(Max) | $MIS_2$(Max) | $MIKS_1$(Max) | $MIKS_2$(Max) |
|---|---|---|---|---|---|---|---|---|
| Random-LNS | 3.41e+07 | 1.61e+09 | 5.50e+07 | 2.29e+09 | 4.47e+07 | 1.73e+09 | 7.08e+07 | 4.06e+09 |
| ACP | 5.43e+07 | 1.71e+09 | 5.90e+07 | 2.41e+09 | 4.11e+07 | 1.60e+09 | 6.02e+07 | 3.50e+09 |
| CL-LNS | - | - | 7.31e+07 | - | 3.17e+07 | - | - | - |
| Gurobi | 3.81e+07 | 2.65e+09 | 5.69e+07 | 2.36e+09 | 4.25e+07 | 1.75e+09 | 6.45e+07 | 3.86e+09 |
| SCIP | 5.18e+07 | 6.33e+09 | 6.39e+07 | 3.92e+09 | 3.34e+07 | 7.26e+07 | 5.77e+07 | 7.03e+08 |
| GNN&GBDT | 5.52e+07 | 3.75e+09 | 5.85e+07 | 2.55e+09 | 4.00e+07 | 1.40e+09 | * | * |
| Light-MILPOPT | 3.73e+07 | 1.80e+09 | 5.44e+07 | 2.27e+09 | 4.30e+07 | 1.70e+09 | 6.78e+07 | 4.03e+09 |
| **LLM-LNS(Ours)** | **3.27e+07** | **1.38e+09** | **5.42e+07** | **2.19e+09** | **4.48e+07** | **1.81e+09** | **7.17e+07** | **4.18e+09** |

the best scores on all instances, outperforming both classic solvers and learning-based frameworks. Notably, it achieves over 20% improvement in primal integral compared to the closest competitor on $SC_2$ and $MVC_2$, while CL-LNS fails to produce results on most large-scale instances, underscoring the scalability limitations of existing learning-based LNS frameworks.

While LLM-LNS offers advantages in efficiency and convergence, its simplicity also brings practical deployment benefits over traditional ML methods. LLM-based approaches often require less infrastructure and are easier to implement and maintain compared to neural networks, which need extensive training data and ongoing updates. This trade-off between performance and ease of deployment makes LLMs a compelling choice for real-world applications.

Overall, the experimental results demonstrate the effectiveness of our proposed innovations. In the first set of experiments, we validate the capability of the **Dual-layer Self-evolutionary LLM Agent** to autonomously generate competitive heuristic strategies for combinatorial optimization problems, consistently outperforming state-of-the-art methods such as FunSearch and EOH. Further supporting this, the ablation experiments presented in Appendix G confirm the effectiveness of each component of the Dual-layer Self-evolutionary Agent. This confirms the agent's

ability to balance exploration and exploitation, as guided by the **Differential Memory for Directional Evolution**. In the second set, we apply the **LLM-LNS framework** to large-scale MILP problems. LLM-LNS not only outperforms traditional LNS methods and advanced solvers like Gurobi and SCIP but also shows superior scalability compared to modern ML-based frameworks. It achieves faster convergence and better solution quality, especially on large instances like $SC_2$ and $MIKS_2$, where other methods struggle. These results highlight the success of our LLM agent to **neighborhood selection in LNS**, showcasing its ability to generalize across complex problems with minimal training data, while demonstrating clear advantages in both efficiency and convergence.

## 5. Conclusion

In this paper, we propose **LLM-LNS**, a Large Language Model-driven LNS framework for solving large-scale MILP problems, utilizing a dual-layer self-evolutionary LLM agent to automate heuristic strategy generation. Experiments show that LLM-LNS consistently outperforms traditional solvers, learning-based methods, and state-of-the-art LNS frameworks. Future work will explore new agent architectures and broader problems, aiming to further enhance the integration of LLMs with optimization techniques.

## Impact Statement

This paper presents work whose goal is to advance the field of Machine Learning. There are many potential societal consequences of our work, none which we feel must be specifically highlighted here.

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

## A. Overview of Appendix

This Appendix contains four sections, each addressing a specific aspect of the experimental setup and results. Below is a brief overview of each section:

- **Detailed Methodology of the Proposed LLM-LNS Framework** (Appendix B): This section provides a detailed explanation of the LLM-LNS framework, including the pseudocode of the framework, a detailed example of the dual-layer evolution process, the population management strategy, and the pseudocode of the Adaptive Large Neighborhood Search (ALNS) algorithm.

- **Parameter Settings** (Appendix C): This section describes key experimental parameters, including the number of top-performing heuristic strategies evaluated, thresholds for stagnation detection, and criteria for evolutionary convergence. Parameter values for Bin Packing (BP), Maximum Vertex Covering (MVC), and Mixed Integer Knapsack Set (MIKS) are also outlined.

- **Evolutionary Process of LLM-LNS** (Appendix D): This section explains the co-evolution of the inner and outer layers in the Dual-layer Self-Evolutionary LLM Agent. It includes comparisons between the Evolution of Heuristic (EoH) method and the proposed dual-layer approach for problems like Bin Packing and Traveling Salesman Problem (TSP).

- **Convergence Analysis of LLM-LNS** (Appendix E): This section analyzes the convergence behavior of the LLM-LNS method compared to EoH. Faster convergence rates, superior solution quality, and greater stability in problems like Online Bin Packing and Traveling Salesman Problem are demonstrated through graphs and figures.

- **Supplementary Experiments for LLM-LNS on Large-Scale MILP Problems** (Appendix F): This section presents the performance of LLM-LNS on large-scale Mixed Integer Linear Programming (MILP) problems, evaluated with different subsolvers (e.g., SCIP) and compared to traditional and learning-based methods. Error bar comparisons highlight solution consistency and reliability.

- **Ablation Study of the Dual-Layer Self-evolutionary LLM Agent** (Appendix G): This section evaluates the contributions of the dual-layer framework, analyzing the roles of Prompt Evolution (outer layer) and Directional Evolution (inner layer). Results from small- and large-scale datasets highlight their complementary effects on convergence, diversity, and performance.

- **Additional Validation Experiments** (Appendix H): This section presents experiments validating the stability, generalization, and robustness of LLM-LNS, with deeper insights into its scalability and consistency.

- **Limitations and Future Directions** (Appendix I): This section discusses the limitations of the proposed framework and outlines potential future directions to enhance its scalability and applicability.

These appendices provide a comprehensive overview of the experimental setup, evolutionary process, convergence analysis, and supplementary experiments, offering a deeper understanding of the performance and robustness of the LLM-LNS method in solving complex combinatorial optimization problems.

## B. Detailed Methodology of the Proposed LLM-LNS Framework

This appendix provides a detailed explanation of the methodology used in the LLM-LNS framework, with a focus on the evolutionary process and optimization strategies. The section is divided into four parts:

1. **Pseudocode of the Proposed LLM-LNS Framework**: A pseudocode representation of the overall LLM-LNS framework, providing an overview of its dual-layer self-evolutionary structure and integration with the Adaptive Large Neighborhood Search process.

2. **Detailed Example of Dual-layer Evolution Process**: A detailed example of the evolutionary process, highlighting the evolution of heuristic strategies and evolutionary prompt strategies over generations, as illustrated in Figure 4.

3. **Population Management Strategy**: A comprehensive description of the population management strategy employed to balance exploration and exploitation during the evolution.

4. **ALNS Pseudocode**: The complete pseudocode of the Adaptive Large Neighborhood Search (ALNS) algorithm, which was omitted from the main text due to space constraints.

---

**Algorithm 1** Pseudocode of the Proposed LLM-LNS Framework

---

**Require:** Small-scale MILP training dataset $\mathcal{D}_{\text{train}}$, large-scale MILP testing dataset $\mathcal{D}_{\text{test}}$, maximum outer iterations $T_{\text{outer}}$, maximum inner iterations $T_{\text{inner}}$, stagnation threshold $t_{\text{stagnation}}$, population size $N$, ALNS parameters

1: **Step 0: Initialization**
2: Initialize heuristic strategy population $\mathcal{H}_0$ using the LLM with initialization prompt strategies
3: Initialize prompt strategy populations $\mathcal{P}_{\text{cross}}$ and $\mathcal{P}_{\text{var}}$ with handcrafted prompts
4: Set stagnation counter $t_{\text{no\_improve}} \leftarrow 0$ and best fitness $\text{best\_fitness} \leftarrow \infty$
5: **for** $t_{\text{outer}} = 1$ to $T_{\text{outer}}$ **do**        ▷ **Outer Layer: Prompt Strategy Evolution**
6:     Fix current prompt strategies $\mathcal{P}_{\text{cross}}$ and $\mathcal{P}_{\text{var}}$
7:     **for** $t_{\text{inner}} = 1$ to $T_{\text{inner}}$ **do**        ▷ **Inner Layer: Heuristic Strategy Evolution**
8:        **Step 1: Generate New Heuristic Strategies**
9:        **for** each selected heuristic strategy $H$ in $\mathcal{H}_{t_{\text{inner}}-1}$ **do**
10:           Use fixed prompt strategies (from $\mathcal{P}_{\text{cross}}$ or $\mathcal{P}_{\text{var}}$) to guide the LLM
11:           Generate new heuristic strategies through crossover, variation, or hybrid operations
12:        **end for**
13:        Add newly generated heuristic strategies to $\mathcal{H}_{t_{\text{inner}}}$ and prune to maintain population size $N$
14:        **Step 2: Evaluate Heuristic Strategies**
15:        **for** each heuristic strategy $H$ in $\mathcal{H}_{t_{\text{inner}}}$ **do**
16:           Apply $H$ in the ALNS process to solve small-scale MILP instances from $\mathcal{D}_{\text{train}}$
17:           Compute fitness score $f(H)$ based on objective values
18:        **end for**
19:        Update $\mathcal{H}_{t_{\text{inner}}}$ by selecting top-$N$ heuristic strategies based on their fitness scores
20:        Increment $t_{\text{no\_improve}}$ if no improvement in best_fitness
21:     **end for**        ▷ End of Inner Layer
22:     **Step 3: Evaluate Prompt Strategies and Avoid Stagnation**
23:     **if** no improvement in best_fitness for $t_{\text{stagnation}}$ outer iterations **then**
24:        **for** each prompt strategy $P$ in $\{\mathcal{P}_{\text{cross}}, \mathcal{P}_{\text{var}}\}$ **do**
25:           Use LLM to generate new prompt strategies through crossover/variation prompts
26:           Evaluate new prompt strategies based on the fitness scores of the heuristics they generate
27:        **end for**
28:        Prune underperforming prompt strategies and retain top-$N$ strategies
29:        Reset $t_{\text{no\_improve}} \leftarrow 0$
30:     **end if**
31: **end for**        ▷ End of Outer Layer
32: **Step 4: Apply ALNS with Best Heuristic Strategy (Testing on $\mathcal{D}_{\text{test}}$)**
33: Select the best heuristic strategy $H^*$ from $\mathcal{H}_{T_{\text{inner}}}$
34: Use $H^*$ to guide the ALNS process for solving large-scale MILP instances from $\mathcal{D}_{\text{test}}$
35: **return** Best heuristic strategy $H^*$ and corresponding solution

---

## B.1. Pseudocode of the Proposed LLM-LNS Framework

The pseudocode outlined in Algorithm 1 follows a dual-layer structure, with an **Outer Layer** responsible for evolving prompt strategies to maintain diversity and an **Inner Layer** focusing on evolving heuristic strategies for improved convergence. The process is divided into the following steps:

- **Initialization.** The framework initializes two key populations: (1) the **heuristic strategy population** $\mathcal{H}_0$, generated by the LLM using initial prompt strategies and small-scale MILP problem structures; and (2) the **prompt strategy populations** $\mathcal{P}_{\text{cross}}$ and $\mathcal{P}_{\text{var}}$, handcrafted to perform basic crossover and variation operations. A stagnation counter and the best fitness value are also initialized to monitor the evolution process.

- **Outer Layer: Prompt Strategy Evolution.** This layer maintains and evolves the prompt strategy populations $\mathcal{P}_{\text{cross}}$ and $\mathcal{P}_{\text{var}}$. At the beginning of each outer iteration, the current prompt strategies are fixed and passed to the Inner Layer for heuristic strategy evolution. After the inner loop completes, the Outer Layer evaluates whether the prompt strategies need to evolve to avoid stagnation. If no improvement in fitness is observed over multiple outer iterations, new prompt strategies are generated using the LLM, evaluated, and updated to ensure diversity in heuristic generation.

- **Inner Layer: Heuristic Strategy Evolution.** Under the guidance of fixed prompt strategies, the Inner Layer evolves the heuristic strategy population $\mathcal{H}$. This process involves:

    - **Generating New Heuristic Strategies.** Using prompt strategies (crossover or variation) from the Outer Layer, the LLM generates offspring heuristic strategies by applying operations such as crossover or mutation to parent

strategies. The offspring strategies are added to the population, and low-performing strategies are pruned to maintain a fixed population size $N$.

- **Evaluating Heuristic Strategies.** Each heuristic strategy is integrated into the ALNS process and applied to small-scale MILP instances. The fitness of each strategy is calculated based on its performance (e.g., objective value). High-performing strategies are selected to form the next generation, ensuring the population improves over time.

- **Step 4: Apply ALNS with Best Heuristic Strategy.** After the dual-layer evolution completes, the best-performing heuristic strategy $H^*$ is selected from the final heuristic strategy population $\mathcal{H}$. This strategy is applied to large-scale MILP problems in the testing dataset $\mathcal{D}_{\text{test}}$ using the ALNS process. The heuristic guides the ALNS framework in dynamically selecting and exploring variable neighborhoods, enabling efficient navigation of the solution space for large-scale optimization problems.

The dual-layer framework provides a robust method for solving large-scale MILP problems by combining the exploration capabilities of prompt strategy evolution in the Outer Layer with the optimization power of heuristic strategy evolution in the Inner Layer. This structure ensures a balance between exploration and exploitation, while the LLM efficiently generates creative and effective strategies for both layers.

## B.2. Detailed Example of Dual-layer Evolution Process

This subsection provides a detailed example of the evolution process for the **Dual-layer Self-evolutionary LLM Agent**, corresponding to Figure 4. This example illustrates how heuristic strategies and evolutionary prompt strategies are iteratively refined over generations to improve performance, with specific focus on the interplay between the Inner and Outer Layers.

### B.2.1. INNER LAYER: HEURISTIC STRATEGY EVOLUTION

The Inner Layer evolves heuristic strategies iteratively, starting from basic initializations and progressively incorporating advanced techniques. The evolution process includes the following key stages:

- **Generation 0: Initialization.** The heuristic strategies are initialized using simple capacity-based methods derived from problem structures. For instance, a heuristic employs a capacity ratio with penalties:

$$\text{scores}[\text{bins} > \text{item}] = \frac{\text{bins} - \text{item}}{\text{bins}} \times \frac{1}{\text{index of bin} + 0.5}. \tag{5}$$

At this stage, the fitness value is **0.9595**. The Outer Layer initializes basic prompt strategies (e.g., Cross1 and Variation1) to generate variations of these heuristics, creating a foundation for further exploration.

- **Generation 5: Exploration with Randomness.** To encourage exploration, randomness is introduced into the heuristic scoring functions. This adjustment enables the system to escape local optima, resulting in a fitness value of **0.9916**. For example:

$$\text{scores}[\text{bins} > \text{item}] = \frac{\text{bins} - \text{item}}{\text{bins} + 10^{-5}} - \frac{(\frac{\text{bins}}{10})^2}{\max(\frac{\text{bins}}{10}^2)} \left(1 - \frac{\text{bins} - \text{item}}{\max(\text{bins})}\right) + \text{randomness} \times 0.1. \tag{6}$$

This improvement is guided by Outer Layer prompts such as Cross2, which emphasizes "creating new algorithms inspired by existing ones," and Variation2, which "adjusts parameters of given algorithms for better exploration."

- **Generation 8: Hybrid Optimization.** Advanced hybrid optimization methods, such as genetic algorithms combined with tabu search, are introduced. These methods allow the system to explore more diverse solutions while avoiding local optima. The fitness value improves to **0.9934**. An example heuristic adapts historical penalties to refine its search direction:

$$\begin{aligned} \text{scores}[\text{bins} > \text{item}] = & \frac{\text{bins} - \text{item}}{\text{bins} + 10^{-5}} - \text{history penalty} - \frac{(\text{bins}/15)^2}{\max((\text{bins}/15)^2)} \\ & + \text{mutation variance} - \left(1 - \frac{\text{bins} - \text{item}}{\max(\text{bins}) + 10^{-5}}\right). \end{aligned} \tag{7}$$

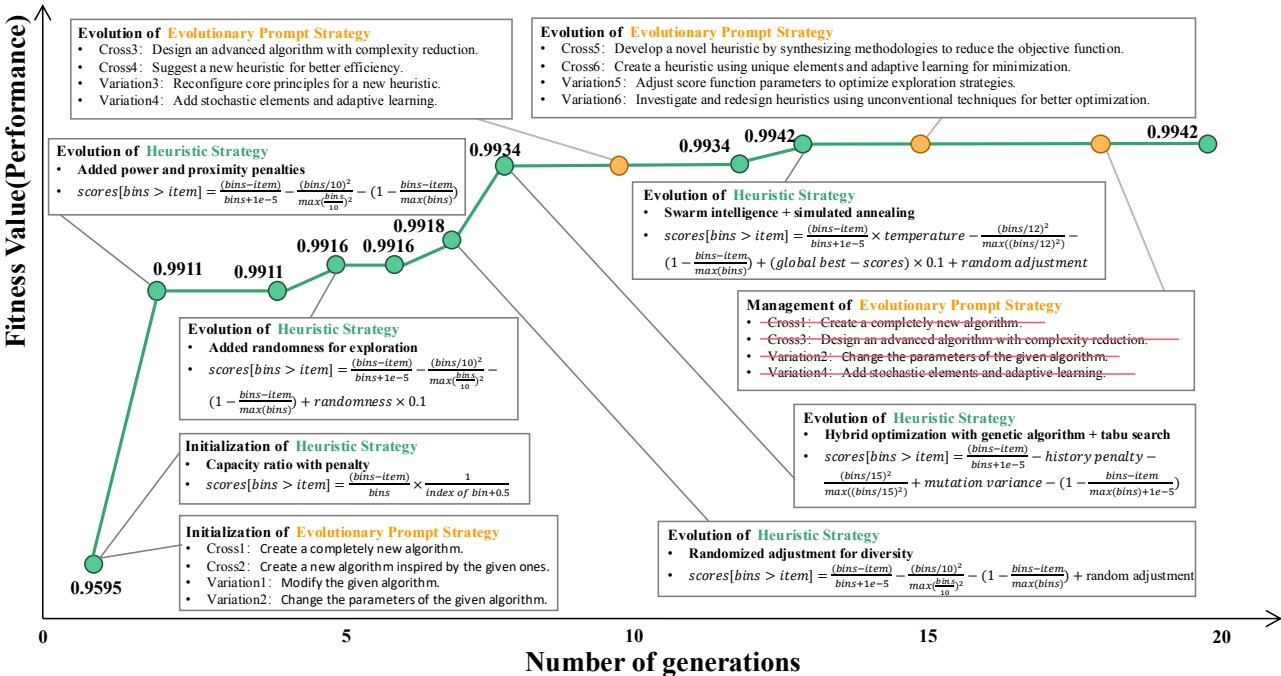

*Figure 4.* Evolution of Dual-layer Self-evolutionary LLM Agent for online bin packing. We outline the key thoughts of the best heuristics produced in some generations during the evolution of heuristic strategies. Additionally, we highlight the evolution of evolutionary prompt strategies, which dynamically adapt the prompt strategies to guide the LLM in generating more effective and diverse heuristics.

Outer Layer prompts evolve to introduce complexity-reducing techniques (e.g., Cross3: "Reconfigure heuristic principles to reduce computational complexity while maintaining effectiveness"), enabling the generation of lighter yet more powerful heuristics.

- **Generation 12: Swarm Intelligence and Diversity Increase.** By combining swarm intelligence methods (e.g., simulated annealing) with randomized adjustments, the heuristic strategies achieve a fitness value of **0.9942**. The scoring function now incorporates global optimization techniques:

$$\text{scores[bins > item]} = \frac{\text{bins} - \text{item}}{\text{bins} + 10^{-5}} \times \text{temperature} - \frac{(\text{bins}/12)^2}{\max((\text{bins}/12)^2)} - \left(1 - \frac{\text{bins} - \text{item}}{\max(\text{bins})}\right) + (\text{global best} - \text{scores}) \times 0.1 + \text{random adjustment.} \tag{8}$$

Prompts such as Variation4 ("Add stochastic elements and adaptive learning") guide this evolution, introducing diversity and enabling broader exploration of the solution space.

### B.2.2. OUTER LAYER: EVOLUTIONARY PROMPT STRATEGY EVOLUTION

The Outer Layer evolves evolutionary prompt strategies to guide the LLM in generating new heuristic strategies. The evolution of prompts adapts to the performance of the Inner Layer, as follows:

- **Generation 1: Initialization.** Basic prompt strategies, such as Cross1 ("Create a completely new algorithm") and Variation1 ("Modify the given algorithm"), are handcrafted to initialize the system. These prompts enable the LLM to generate simple heuristics based on problem structures.

- **Generation 6: Addressing Stagnation.** As performance stagnates, advanced prompt strategies are introduced. For example, Cross3 emphasizes reducing algorithmic complexity:
  - *Example Prompt:* "Reconfigure heuristic principles to reduce computational complexity while maintaining effectiveness."

- **Generation 10: Adaptive Learning.** Prompts such as Variation5 and Variation6 incorporate stochastic adjustments and adaptive learning to overcome local optima:

  - *Example Prompt:* "Enhance diversity by introducing stochastic elements to refine exploration strategies."

  These prompts enable the generation of heuristics that combine exploration and exploitation effectively.

- **Generation 18: Pruning Underperforming Strategies.** Underperforming prompt strategies are pruned based on their fitness scores, ensuring that only the most effective prompts are retained. This step streamlines the search process and focuses computational resources on high-performing strategies.

### B.2.3. SYNERGY BETWEEN LAYERS

The synergy between the Inner and Outer Layers is evident in the performance improvements across generations, as shown in Figure 4. Each major improvement corresponds to a refinement of both heuristic strategies and prompt strategies:

- Between **Generations 1 and 5**, randomness introduced by Variation2 enables broader exploration, significantly boosting fitness from **0.9595** to **0.9916**.

- Between **Generations 5 and 8**, hybrid optimization techniques (e.g., genetic algorithms) guided by Cross3 prompts enable heuristics to escape local optima, achieving a fitness of **0.9934**.

- Between **Generations 8 and 12**, swarm intelligence combined with stochastic adjustments (guided by Variation4) further enhances diversity, pushing the fitness to **0.9942**.

Overall, the Outer Layer's refinement of prompt strategies drives the Inner Layer's ability to generate increasingly sophisticated heuristics, demonstrating the power of the dual-layer evolution process.

### B.3. Population Management Strategy

To ensure the effectiveness and diversity of strategies within the LLM-LNS framework, we employ a population management strategy that balances exploration and exploitation during each generation. This strategy governs the selection of parent strategies for evolutionary operations (e.g., crossover and mutation) and the replacement of poorly performing strategies to maintain a high-quality population.

### B.3.1. SELECTION OF EVOLUTIONARY STRATEGIES

At each generation, the framework uses a probabilistic sampling mechanism to select $m$ parent strategies from the population for crossover and mutation. The probability of selecting a strategy is determined by its fitness value, which reflects its performance in achieving the optimization objective. Specifically, let the population contain $n$ strategies with fitness values ranked in descending order as $f_1, f_2, \ldots, f_n$. The probability of selecting the $i$-th strategy is given by:

$$P_i = \frac{1}{i + 1 + n}, \quad i = 1, 2, \ldots, n, \tag{9}$$

where $i$ represents the rank of the strategy (starting from 0), and $n$ is the population size. This ranking-based probability distribution ensures that higher-fitness strategies are more likely to be selected while preserving some randomness to allow lower-fitness strategies to participate. Such randomness enhances exploration by preventing premature convergence to local optima.

Using this probability distribution, we sample $m$ parent strategies for evolutionary operations. These operations generate new candidate strategies, which are evaluated and integrated into the population based on their fitness values.

### B.3.2. MANAGEMENT OF POORLY PERFORMING STRATEGIES

After each generation, the population is updated to maintain a fixed size while ensuring diversity and quality. Let the current population be $P = \{s_1, s_2, \ldots, s_n\}$, where each strategy $s_i$ has a fitness value $f(s_i)$. The goal is to construct a new population $P'$ such that:

---

**Algorithm 2** Adaptive Large Neighborhood Search (ALNS)

---

**Require:** Initial solution $\mathbf{x}_0$, initial neighborhood size $k$, time limit $T$, threshold $\epsilon$, iteration limit $p$, minimum and maximum neighborhood sizes $k_{\min}, k_{\max}$, decision variable count $n$, adjustment rate $u\%$ (percentage)

 1: Initialize solution $\mathbf{x} \leftarrow \mathbf{x}_0$, set time $t \leftarrow 0$
 2: **while** $t < T$ **do**
 3:     Compute variable scores $s_i$ using the LLM agent
 4:     Select top-$k$ variables to form neighborhood $\mathcal{N}$
 5:     Solve subproblem within $\mathcal{N}$ using a solver
 6:     Update solution $\mathbf{x}$ if an improvement is found
 7:     **if** time spent in neighborhood exceeds predefined limit **then**
 8:         $k \leftarrow \max(k_{\min}, k - \lceil u\% \cdot n \rceil)$              $\triangleright$ Reduce neighborhood size by $u\%$ of $n$
 9:     **else if** improvement in objective value $< \epsilon$ for $p$ consecutive iterations **then**
10:         $k \leftarrow \min(k_{\max}, k + \lceil u\% \cdot n \rceil)$              $\triangleright$ Expand neighborhood size by $u\%$ of $n$
11:     **end if**
12:     Update time $t$
13: **end while**
14: **return** $\mathbf{x}$

---

- $P'$ contains at most size strategies, where size is a predefined parameter,

- Strategies with duplicate fitness values are removed,

- The highest-fitness strategies are retained.

The population update process is as follows: 1. Remove strategies with invalid or undefined fitness values. 2. Eliminate duplicate strategies by retaining only one instance of strategies with the same fitness value. 3. Rank the remaining strategies by fitness value in descending order and select the top size strategies to form the new population $P'$.

This management process ensures that the population remains diverse while focusing on high-quality strategies, avoiding redundancy and inefficiency. By preserving the highest-fitness strategies and introducing new candidates through evolutionary operations, the framework achieves a balance between exploration and exploitation.

### B.3.3. FITNESS EVALUATION

The fitness value of a strategy is determined by its optimization performance on a set of small-scale training problems. Specifically, the fitness value $f(s_i)$ for a strategy $s_i$ is calculated as the average objective value achieved across multiple problem instances:

$$f(s_i) = \frac{1}{|I|} \sum_{j \in I} \text{Obj}(s_i, I_j), \tag{10}$$

where $I$ is the set of training problem instances, and $\text{Obj}(s_i, I_j)$ represents the objective value achieved by strategy $s_i$ on instance $I_j$. This evaluation method ensures that strategies are assessed based on consistent and robust performance metrics.

### B.3.4. SUMMARY

The population management strategy in the LLM-LNS framework combines fitness-based selection, diversity preservation, and rigorous fitness evaluation. By maintaining a high-quality and diverse population, the framework progressively improves the quality of strategies across generations. This strategy, together with the LLM's ability to generalize and optimize, enables the LLM-LNS framework to efficiently navigate large and complex search spaces, balancing exploration and exploitation to achieve superior optimization performance.

### B.4. ALNS Pseudocode

This subsection presents the pseudocode for the Adaptive Large Neighborhood Search (ALNS) algorithm, which dynamically adjusts the neighborhood size and leverages the Dual-layer Self-evolutionary LLM Agent to efficiently solve large-scale MILP problems.

**Explanation of the ALNS Algorithm.** The Adaptive Large Neighborhood Search (ALNS) algorithm is designed to iteratively improve solutions for large-scale MILP problems by dynamically adjusting the search space based on feedback

from the optimization process. The key steps of the algorithm are as follows:

- **Initialization:** The algorithm begins with an initial solution $\mathbf{x}_0$ and an initial neighborhood size $k$. A time limit $T$ and other parameters, such as threshold $\epsilon$, iteration limit $p$, and neighborhood size bounds $k_{\min}$ and $k_{\max}$, are predefined.

- **Variable Scoring and Neighborhood Selection:** At each iteration, the Dual-layer Self-evolutionary LLM Agent computes scores $s_i$ for the problem's decision variables $x_i$, indicating their potential to improve the objective value. The top-$k$ variables are then selected to form the neighborhood $\mathcal{N}$:

$$\mathcal{N} = \{x_i \mid \text{rank}(s_i) \leq k\}. \tag{11}$$

- **Subproblem Solving:** Within the selected neighborhood $\mathcal{N}$, a subproblem is solved using a solver. If the solution $\mathbf{x}$ improves the objective value, the solution is updated.

- **Dynamic Adjustment of Neighborhood Size:**
    - **Reduction:** If the time spent solving subproblems within the neighborhood exceeds a predefined limit, the neighborhood size $k$ is reduced to focus on a smaller subset of variables:

$$k \leftarrow \max(k_{\min}, k - \lceil u\% \cdot n \rceil), \tag{12}$$

    where $u\%$ is the adjustment rate and $n$ is the total number of decision variables.
    - **Expansion:** If the improvement in the objective value is less than $\epsilon$ for $p$ consecutive iterations, the neighborhood size is expanded to explore a broader search space:

$$k \leftarrow \min(k_{\max}, k + \lceil u\% \cdot n \rceil). \tag{13}$$

- **Termination:** The process continues until the time limit $T$ is reached, at which point the best solution $\mathbf{x}$ is returned.

The ALNS algorithm's dynamic adjustment of neighborhood size ensures efficient exploration and exploitation of the solution space. By leveraging the LLM agent to score variables, the algorithm generalizes variable selection strategies learned from small-scale MILP problems, enabling its application to larger, more complex problems. The combination of intelligent variable scoring and adaptive neighborhood adjustment allows ALNS to efficiently navigate the vast search space, improving solution quality while maintaining computational efficiency.

## C. Experimental Settings

In this section, we detail the parameter settings used in our experiments for both the Dual-layer Self-evolutionary LLM Agent and the Adaptive Large Neighborhood Search (ALNS). We also provide an overview of the standard MILP problem instances used in this study. Furthermore, we will describe the specific function signature that our self-evolutionary LLM agent is designed to evolve.

### C.1. Dual-layer Self-evolutionary LLM Agent Parameters

The following key parameters were used for the evolutionary process of the LLM agent:

- $h$: Represents the number of top-performing heuristic strategies used to evaluate each prompt strategy. For each prompt strategy, the top-$h$ heuristics it generates are tracked, and their average fitness score is used as the fitness score for the prompt strategy. In our experiments, $h$ is set to half of the population size. Specifically:
    - For **Bin Packing (BP)** and **Traveling Salesman Problem (TSP)**, the population sizes are 20 and 10, respectively, so $h$ is set to 10 and 5.
    - For the four MILP problems—**Maximum Vertex Covering (MVC)**, **Set Covering (SC)**, **Independent Set (IS)**, and **Mixed Integer Knapsack Set (MIKS)**—the population size is 4, so $h$ is set to 2.

- $l$: Denotes the number of top individuals in the heuristic population that are monitored for stagnation. If the top-$l$ individuals remain unchanged for $t$ generations, we infer that the evolution has potentially converged to a local optimum, triggering the introduction of new prompt strategies. In all our experiments, $l$ is set to 4.

- $t$: The number of consecutive generations during which the top-$l$ individuals must remain unchanged before stagnation is detected. In all our experiments, $t$ is set to 3.

## C.2. Adaptive Large Neighborhood Search (ALNS) Parameters

For ALNS, we use the following parameters:

- **Neighborhood size** $k$: Set to half of the decision variable count $n$. This represents the number of decision variables selected to form the search neighborhood in each iteration.

- **Time limit** $T$: To ensure robustness and avoid excessive computation, we impose a maximum time limit per LNS iteration: 100 seconds for problems with approximately 100K variables and 200 seconds for problems with around 1M variables. This time limit is applied uniformly across all LNS-based baselines to prevent slow subproblem solves, ensuring efficiency and fair comparison in our experiments.

- **Threshold** $\epsilon$: Represents the minimum improvement in the objective function to continue exploring the current neighborhood. We set $\epsilon = 1\text{e-}3$.

- **Iteration limit** $p$: The number of consecutive iterations with improvements below the threshold $\epsilon$ before expanding the neighborhood size. We set $p = 3$.

- **Minimum and maximum neighborhood sizes** $k_{\min}, k_{\max}$: These are set to $k_{\min} = 0$ and $k_{\max} = n$ (the total number of decision variables in the problem).

- **Adjustment rate** $u\%$: Specifies the percentage of decision variables $n$ by which the neighborhood size is adjusted during expansion or reduction. In our experiments, we set $u\% = 10$.

## C.3. Datasets for Heuristic Evolution

To ensure a fair comparison with state-of-the-art methods such as EOH, we adopted the same dataset configurations as those used in EOH for heuristic evolution. For example, in the online bin packing problem, the evaluation dataset consists of five sets of instances, each containing 5,000 items generated from a Weibull distribution. These instances cover a wide range of item counts and container capacities, ensuring the diversity and representativeness of the problem settings. Similarly, for the traveling salesman problem (TSP), we utilized 64 randomly selected instances from TSP100, which were also used in EOH's experiments. These instances provide a well-established basis for evaluating heuristic performance in combinatorial optimization tasks.

For MILP problems, we followed a similar design approach to that used in the online bin packing problem. Specifically, we employed five small-scale MILP problems, each involving tens of thousands of decision variables and linear constraints. These smaller-scale problems serve as a foundation for heuristic evolution, allowing the method to generalize effectively to larger-scale MILP problems with hundreds of thousands or even millions of decision variables. This demonstrates the scalability and practical applicability of our approach when addressing large-scale optimization challenges.

## C.4. Experimental Settings for Algorithm Design

Our proposed dual-layer agent framework is designed to evolve heuristics for solving combinatorial optimization problems, specifically targeting Online Bin Packing (BP) and the Travelling Salesman Problem (TSP). The dual-layer architecture is responsible for learning and refining heuristic strategies for these problems, enabling efficient and scalable solutions. Below, we provide detailed descriptions of the experimental settings for each problem.

For **Online Bin Packing**, we adopt the settings described in (Romera-Paredes et al., 2024) and (Liu et al., 2024) to design heuristics for determining suitable bin allocations for incoming items (Angelopoulos, 2023). The task of the dual-layer agent is to design a scoring function that assigns items to bins. The inputs to the agent include the size of the item and the remaining capacities of the bins, while the output is a set of scores for the bins. The item is then assigned to the bin with the highest score. This process is iterated for each incoming item, allowing the agent to dynamically adapt its scoring strategy based on the evolving state of the bins.

For the **Travelling Salesman Problem (TSP)**, we use the dual-layer agent to design heuristics for Guided Local Search (GLS) (Voudouris et al., 2010). GLS introduces perturbations and dynamically adjusts the objective landscape to help escape local optima, enabling broader exploration of the solution space. A critical task in GLS is updating the distance matrix to guide the local search towards more promising regions. In this context, the dual-layer agent is tasked with producing

*Table 6.* The size of one real-world case study in the internet domain and four widely used NP-hard benchmark MILPs.

| Problem | Scale | Number of Variables | Number of Constraints |
|---|---|---|---|
| SC (Minimize) | $SC_1$ | 200000 | 200000 |
| | $SC_2$ | 2000000 | 2000000 |
| MVC (Minimize) | $MVC_1$ | 100000 | 300000 |
| | $MVC_2$ | 1000000 | 3000000 |
| MIS (Maximize) | $MIS_1$ | 100000 | 300000 |
| | $MIS_2$ | 1000000 | 3000000 |
| MIKS (Maximize) | $MIKS_1$ | 200000 | 200000 |
| | $MIKS_2$ | 2000000 | 2000000 |

heuristics for updating the distance matrix. The inputs include the current distance matrix, the current route, and the number of edges, while the output is an updated distance matrix. GLS then applies local search operators iteratively on the updated landscape to refine the solution. In our experiments, we utilize two common local search operators: the relocate operator and the 2-opt operator, which are widely recognized for their effectiveness in TSP optimization (Arnold & Sörensen, 2019).

These settings are aligned with those used in EOH to ensure fair comparisons and reproducibility. Detailed descriptions of the inputs, outputs, and operators are provided in the appendix of the manuscript to further clarify our experimental configurations.

We also emphasize that no seed heuristics, expert-written code, or prior knowledge were manually introduced during the experiments. All heuristic strategies were initialized automatically by the large language model (LLM), ensuring fairness in the comparisons.

## C.5. MILP Problem Overview

We use a set of standard problem instances based on four canonical MILP problems: Maximum Independent Set (MIS), Minimum Vertex Covering (MVC), Set Covering (SC), and Mixed Integer Knapsack Set (MIKS). Below are the formal definitions of these problems.

To ensure the robustness and fairness of our evaluation, the generation and selection of these instances follow a rigorous procedure. The instances are not arbitrarily selected but are systematically generated based on standard formulations of these canonical MILP problems. For each problem class and size category, a substantial set of instances is initially generated by randomly sampling parameters from a consistent underlying distribution specific to that problem class. Subsequently, these generated instances are randomly partitioned into training and testing sets, preventing selection bias and ensures a more reliable assessment of our framework's generalization capabilities across different problem types and scales.

To evaluate the scalability and generalization ability of our framework, we design experiments on both small-scale training instances and large-scale testing instances. Specifically:

- The **small-scale training instances** are designed with sizes corresponding to 1% of the decision variables and constraints of the large-scale testing instances. These small-scale problems are used to train and evolve heuristic and prompt strategies.

- The **large-scale testing instances** are significantly larger, featuring up to $10^6$ decision variables and $3 \times 10^6$ constraints (as shown in Table 6). These instances are used to evaluate the generalization ability of the strategies evolved during training.

Our experimental results demonstrate that the proposed framework achieves strong generalization, effectively solving large-scale MILP problems even when trained solely on small-scale instances. This highlights the ability of the framework to transfer knowledge learned from small-scale problems to much larger, real-world problems.

**Maximum Independent Set problem (MIS)**: The Maximum Independent Set problem has applications in network design, where one might need to select the largest subset of mutually non-interacting entities, such as devices in a wireless network to avoid interference. Another common application is in social network analysis, where independent sets can represent groups of users who do not have direct connections, useful for targeting non-overlapping communities.

Consider an undirected graph $\mathcal{G} = (\mathcal{V}, \mathcal{E})$, where a subset of nodes $\mathcal{S} \subseteq \mathcal{V}$ is called an independent set if no edge $e \in \mathcal{E}$ exists between any pair of nodes in $\mathcal{S}$. The MIS problem seeks to find an independent set of maximum cardinality. The binary decision variable $x_v$ indicates whether node $v \in \mathcal{V}$ is part of the independent set ($x_v = 1$) or not ($x_v = 0$). The problem can be formulated as:

$$
\begin{aligned}
\max \quad & \sum_{v \in \mathcal{V}} x_v \\
\text{s.t.} \quad & x_u + x_v \leq 1, \quad \forall (u, v) \in \mathcal{E}, \\
& x_v \in \{0, 1\}, \quad \forall v \in \mathcal{V}.
\end{aligned}
\tag{14}
$$

**Minimum Vertex Covering problem (MVC)**: The Minimum Vertex Covering problem is widely used in resource allocation, where one needs to ensure that every interaction (edge) between pairs of objects (nodes) is covered by a resource. For example, in network security, this problem can be used to efficiently place security agents or sensors such that all communication links are monitored.

Given an undirected graph $\mathcal{G} = (\mathcal{V}, \mathcal{E})$, a subset of nodes $\mathcal{S} \subseteq \mathcal{V}$ is called a covering set if for any edge $e \in \mathcal{E}$, at least one of its endpoints is included in $\mathcal{S}$. The MVC problem aims to find a covering set of minimum cardinality. The binary decision variable $x_v$ indicates whether node $v \in \mathcal{V}$ is part of the covering set ($x_v = 1$) or not ($x_v = 0$). The problem is formulated as:

$$
\begin{aligned}
\min \quad & \sum_{v \in \mathcal{V}} x_v \\
\text{s.t.} \quad & x_u + x_v \geq 1, \quad \forall (u, v) \in \mathcal{E}, \\
& x_v \in \{0, 1\}, \quad \forall v \in \mathcal{V}.
\end{aligned}
\tag{15}
$$

**Set Covering problem (SC)**: The Set Covering problem is fundamental in facility location, where one must select the minimum number of locations (subsets) to serve all customers (elements of the universal set). It is also used in airline crew scheduling, where the goal is to assign the minimum number of crews to cover all flights.

Given a finite universal set $\mathcal{U} = \{1, 2, \ldots, n\}$ and a collection of $m$ subsets $S_1, \ldots, S_m$ of $\mathcal{U}$, each subset $S_i$ is associated with a cost $c_i$. The SC problem involves selecting a combination of these subsets such that every element in $\mathcal{U}$ is covered by at least one of the selected subsets, while minimizing the total cost. The binary decision variable $x_i$ indicates whether subset $S_i$ is selected ($x_i = 1$) or not ($x_i = 0$). The problem is formulated as:

$$
\begin{aligned}
\min \quad & \sum_{i=1}^{m} c_i x_i \\
\text{s.t.} \quad & \sum_{i=1}^{m} x_i \cdot \mathbf{1}_{\{j \in S_i\}} \geq 1, \quad \forall j \in \mathcal{U}, \\
& x_i \in \{0, 1\}, \quad \forall i \in \{1, \ldots, m\}.
\end{aligned}
\tag{16}
$$

**Mixed Integer Knapsack Set problem (MIKS):** The Mixed Integer Knapsack Set problem arises in various applications such as logistics, resource allocation, and portfolio optimization. It captures scenarios where some items can be selected fractionally, while others must be either fully selected or excluded. For instance, in supply chain management, certain goods can be split and shipped in parts, whereas others must be shipped as complete units.

The MIKS problem extends the classical knapsack formulation by incorporating both continuous and binary decision variables, along with multiple capacity constraints. Given $N$ items and $M$ resource dimensions, each item consumes capacity in one or more dimensions, and the total usage in each dimension must not exceed the available capacity. The objective is to maximize the total value of selected items. Let $x_i$ denote the selection variable for item $i$, where $x_i = 1$ indicates full selection, and $0 \leq x_i \leq 1$ allows partial selection. The problem is formulated as:

$$\max \quad \sum_{i=1}^{N} c_i x_i$$

$$\text{s.t.} \quad \sum_{i:j\in S_i} x_i \leq 1, \quad \forall j \in \{1, 2, \ldots, M\},$$

$$0 \leq x_i \leq 1, \quad \forall i \in \{1, 2, \ldots, N\},$$

$$x_i \in \{0, 1\} \text{ or } [0, 1], \quad \forall i \in \{1, 2, \ldots, N\}.$$

$(17)$

### C.6. MILP Problem Generation Process

The generation of MILP problems follows a structured approach to ensure consistency and diversity across problem instances. The process is broken down into several steps, as outlined below.

**Step 1: Random Graph and Set Generation**

For the generation of the problems, we first define the underlying structure of the problem using random graphs or sets, depending on the problem type. Specifically, we:

- For the Maximum Independent Set (MIS) and Minimum Vertex Covering (MVC) problems, we randomly generate an undirected graph $\mathcal{G} = (\mathcal{V}, \mathcal{E})$, where $\mathcal{V}$ is a set of vertices and $\mathcal{E}$ is a set of edges. The number of vertices and edges is chosen based on the desired problem scale.

- For the Set Covering (SC) problem, we generate a collection of sets $\mathcal{S}$ covering elements from a universal set $\mathcal{U}$. Each subset is randomly selected, and the number of sets is based on the problem size.

- For the Mixed Integer Knapsack Set (MIKS) problem, we generate a list of items with associated weights, profits, and capacities. The items are randomly selected, with some variables being continuous and others integer-based.

**Step 2: Define Objective Function and Constraints**

Once the underlying graph or sets are generated, the objective function and constraints are formulated based on the problem type:

- In the MIS and MVC problems, the objective function is formulated to either maximize the number of nodes in the independent set (MIS) or minimize the number of nodes in the vertex cover (MVC). Constraints are added to enforce the independence or covering conditions.

- For SC, the goal is to minimize the total cost of selected subsets while ensuring all elements in $\mathcal{U}$ are covered. Constraints are added to ensure that each element is covered by at least one subset.

- In MIKS, we define a profit-weight relationship for each item, where we aim to maximize the profit while ensuring the total weight does not exceed a given capacity.

**Step 3: Randomization and Scaling**

The parameters such as the number of nodes, edges, sets, and constraints are randomly scaled to generate problem instances of varying sizes. The scaling process ensures that problem instances span a broad range of complexities and sizes:

- For each problem class, the number of nodes (or sets) and edges (or item types) are randomly sampled from a predefined distribution, such as uniform or Gaussian, to create diverse instances.

- The size of the instances is controlled to generate both small-scale and large-scale problem instances, where small-scale instances are used for training and large-scale instances are used for testing.

**Step 4: Validation and Evaluation**

Finally, the generated problem instances are validated to ensure they adhere to the correct problem formulations. This includes checking if the constraints are satisfied and whether the generated instances match the intended problem characteristics. The performance of the model is then evaluated on the testing instances, assessing both solution quality and computational efficiency.

This systematic process ensures that the generated MILP instances cover a wide range of problem types and sizes, providing a reliable basis for evaluating optimization algorithms and models.

### C.7. Evolved Heuristic Function Signature

To evaluate the effectiveness of our proposed dual-layer self-evolutionary LLM agent, we tested its ability to design heuristic functions across three distinct classes of optimization problems: Online Bin Packing (bp_online), Traveling Salesman Problem (TSP), and general Mixed Integer Linear Programming (MILP) instances. Below, we detail the specific input and output signature of the function that our agent evolved for each problem type.

#### C.7.1. ONLINE BIN PACKING (BP_ONLINE)

For the **Online Bin Packing** problem, the self-evolutionary agent was tasked with designing a heuristic scoring function with the signature `def score(item, bins):`.

The fundamental goal of this function is to evaluate a set of available bins and assign a score to each, indicating its suitability for the current `item` that needs to be packed. In each step of the packing process, the `item` is placed into the bin that receives the highest score among feasible options. According to the problem specification, bins whose remaining capacity equals the overall maximum bin capacity are not considered for placing the current item, effectively prioritizing the use of already partially filled bins. The overarching objective is to minimize the total number of bins ultimately utilized.

- **Inputs**:
    - `item` (int): The size of the current item to be placed.
    - `bins` (NumPy array): An array representing the remaining capacities of all feasible bins. These are bins that have sufficient space to accommodate the current `item`.

- **Output**:
    - `scores` (NumPy array): An array of the same dimension as `bins`, where each element is the calculated score for the corresponding bin.

The agent was encouraged to develop a scoring logic of sufficient complexity to potentially outperform simpler heuristics and to ensure the internal consistency of the evolved function.

#### C.7.2. TRAVELING SALESMAN PROBLEM (TSP)

For the **Traveling Salesman Problem**, the agent's task was to devise a heuristic strategy for dynamically updating the edge distance matrix. The evolved function has the signature `def update_edge_distance(edge_distance, local_opt_tour, edge_n_used):`.

The primary objective of this function is to modify the perceived distances between nodes based on the characteristics of a recently found local optimal tour and historical edge usage. By altering the `edge_distance` matrix, the heuristic aims to help the search process escape local optima and explore different regions of the solution space, ultimately facilitating the discovery of a tour with a minimized total distance. The function is expected to return the modified distance matrix.

- **Inputs**:
    - `edge_distance` (NumPy array): The current matrix representing the distances between nodes.
    - `local_opt_tour` (NumPy array): An array of node IDs constituting the local optimal tour found in a previous iteration.
    - `edge_n_used` (NumPy array): A matrix detailing the frequency with which each edge has been included in explored solutions or permutations.

- **Output**:

    - `updated_edge_distance` (NumPy array): The new edge distance matrix, adjusted by the evolved heuristic.

All input and output arrays are NumPy arrays. This dynamic adjustment is crucial for guiding the neighborhood search effectively.

### C.7.3. MIXED INTEGER LINEAR PROGRAMMING (MILP)

For **Mixed Integer Linear Programming (MILP)** problems, the self-evolutionary agent was tasked with designing a heuristic to guide the neighborhood selection process within a Large Neighborhood Search (LNS) framework. The objective is to intelligently score decision variables to determine which ones should be relaxed and re-optimized to improve the current solution. The signature of the evolved function is:
`def select_neighborhood(n, m, k, site, value, constraint, initial_solution, current_solution, objective_coefficient):`.

The primary goal of this function is to analyze the current state of the MILP problem—including its structure, constraints, objective function, and current solution—and assign a numerical score to each decision variable. These scores (`neighbor_score`) subsequently inform the LNS mechanism on which variables to include in the neighborhood (i.e., to "free" for re-optimization). The agent was encouraged to develop strategies that could consider variable correlations (e.g., by assigning similar scores to variables involved in the same constraint) or incorporate randomness to escape local optima, aiming to iteratively enhance the `current_solution`.

- **Inputs**:

    - `n` (int): The total number of decision variables in the MILP instance.
    - `m` (int): The total number of constraints in the MILP instance.
    - `k` (NumPy array): An array of length `m`, where `k[i]` denotes the number of decision variables participating in the $i$-th constraint.
    - `site` (list of NumPy arrays): A list of `m` NumPy arrays. For the $i$-th constraint, `site[i][j]` indicates the index (ID) of the $j$-th decision variable involved in that constraint.
    - `value` (list of NumPy arrays): A list of `m` NumPy arrays. For the $i$-th constraint, `value[i][j]` specifies the coefficient of the $j$-th decision variable (as identified by `site[i][j]`) in that constraint.
    - `constraint` (NumPy array): An array of length `m`, where `constraint[i]` is the right-hand side (RHS) value for the $i$-th constraint.
    - `initial_solution` (NumPy array): An array of length `n`, holding the values of the decision variables from an initial feasible solution.
    - `current_solution` (NumPy array): An array of length `n`, representing the values of the decision variables in the current incumbent solution that LNS aims to improve.
    - `objective_coefficient` (NumPy array): An array of length `n`, where `objective_coefficient[i]` is the coefficient of the $i$-th decision variable in the objective function.

- **Output**:

    - `neighbor_score` (NumPy array): An array of length `n`, generated by the evolved function. Each element `neighbor_score[i]` corresponds to the calculated score for the $i$-th decision variable, influencing its likelihood of being selected for the LNS neighborhood.

All inputs are provided as NumPy arrays, except for `n` and `m` (integers), and `site` and `value` (lists of NumPy arrays). The evolved function is responsible for creating and populating the `neighbor_score` NumPy array.

### C.8. Training Setup and Generalization in TSP and Bin Packing

To ensure fair comparisons and demonstrate the generalization ability of our approach, we closely followed the experimental setups used in prior work such as EOH and FunSearch.

For the TSP experiments, we used the same five TSPLib instances—d198, eil76, rat99, rl1889, and u1060—as the training set for evolving our policies. None of the other TSPLib instances used in evaluation were included in training. This separation guarantees that performance improvements are not due to overfitting specific instances. We will update Table 17 to clearly mark the training instances. For the Bin Packing task, we adopted the Weibull 5k test dataset as the training data, consistent with the setup in EOH and FunSearch. This ensures a comparable training environment across methods while allowing us to test the ability of our LLM agent to generalize to different instance distributions and sizes.

## D. Evolutionary Process of LLM-LNS

### D.1. Evolutionary Process Overview

In this appendix, we provide a detailed breakdown of the experimental results and the evolution of heuristic strategies generated by our proposed **Dual-layer Self-Evolutionary LLM Agent**. The following sections offer a comprehensive analysis of how the inner and outer layers of the LLM agent collaborate to generate and refine heuristic strategies across various combinatorial optimization problems, including **Online Bin Packing (bp_online)**, the **Traveling Salesman Problem (TSP)**, and large-scale MILP instances such as **Maximum Vertex Covering (MVC)**, **Set Covering (SC)**, **Independent Set (IS)**, and **Mixed Integer Knapsack Set (MIKS)**.

- **Inner and Outer Layer Prompt Initialization and Evolution**: As shown in Sec. D.2, our approach leverages a dual-layer architecture, where the **inner layer** evolves heuristic strategies by modifying solution components, while the **outer layer** evolves the prompt structure guiding the inner layer, balancing exploration and exploitation. The inner layer prompts iteratively generate heuristics by scoring decision variables based on their contributions to the objective function and constraints, with randomness included to avoid local optima. This enables the LLM to reason about the problem structure and generate high-quality strategies, even without extensive domain expertise. The **outer layer** maintains diversity by evolving prompt structures to prevent premature convergence on suboptimal solutions. Both layers adapt based on past performance, allowing the LLM to refine its strategy generation over time.

- **Heuristic Improvement Through Dual-layer Self-evolutionary LLM Agent**: As shown in Sec. D.3, we demonstrates the progression of heuristic strategies, starting from initial random strategies and gradually evolving into more effective ones through the dual-layer self-evolutionary process. The initial strategies are simple and focus on ranking decision variables based on their contributions to the objective function and constraints. Over time, the LLM agent introduces additional complexity, such as incorporating randomness and penalizing larger deviations from the current solution, improving the robustness of the generated heuristics. The progression of the population is guided by the outer layer, which adjusts the structure and focus of prompts to encourage exploration and avoid premature convergence. The inner layer then refines specific solution components in response to the prompts, iteratively improving the performance of the heuristic strategies. As seen from the evolution of objective scores, the dual-layer system enables the generation of increasingly effective heuristics, balancing exploration with exploitation to achieve superior results in various problem instances.

- **Heuristic Strategies for Bin Packing Online: EoH vs. Dual-Layer Self-Evolution LLM Agent**: As shown in Sec. D.4, both the *Evolution of Heuristic (EoH)* method and our Dual-layer Self-Evolution LLM Agent utilize LLM-based evolutionary processes to generate heuristic strategies for the *Bin Packing Online* problem. The strategy generated by EoH approach, while leveraging LLM to evolve heuristics, focuses primarily on a hybrid scoring system that combines utilization ratios, dynamic adjustments, and an exponentially decaying factor. This method is effective but tends to rely on a more static set of features and parameters, which limits its adaptability across diverse problem instances. In contrast, our Dual-layer Self-Evolution LLM Agent incorporates a more dynamic and adaptive strategy. By combining nonlinear capacity scaling, relative size assessment, and historical penalties for overutilized bins, our approach allows for greater flexibility and adaptability. Specifically, the generated heuristics dynamically adjust based on remaining capacity, item size, and previous bin usage, thereby balancing local search with global optimization. This adaptability enables our agent to discover and refine more efficient strategies that minimize the number of bins used. The results clearly demonstrate that while both methods use LLM-based evolution, our dual-layer approach consistently outperforms the EoH method in terms of solution quality and computational efficiency. The dual-layer system's ability to evolve both the heuristic strategies and the prompt structures ensures that it can fine-tune solutions more effectively, leading to superior bin utilization and fewer bins required overall. This highlights the strength of our approach in generating more robust and context-aware heuristics.

- **Heuristic Strategies for Traveling Salesman Problem (TSP): EoH vs. Dual-Layer Self-Evolution LLM Agent**: Similar to the *Bin Packing Online* problem, both the *Evolution of Heuristic (EoH)* method and our Dual-layer Self-Evolution LLM Agent use LLM-based evolutionary processes to generate heuristic strategies for the *Traveling Salesman Problem (TSP)*. As shown in Sec. D.5, the strategy generated by EoH method employs a randomized approach that adjusts the edge distance matrix by increasing the distances of a random proportion of edges, while rewarding a smaller subset of unused edges. This method encourages exploration but tends to apply uniform adjustments without fully accounting for the global structure of the solution. In contrast, strategy generated by our Dual-layer Self-Evolution LLM Agent introduces a more sophisticated edge distance adjustment mechanism. It dynamically explores alternative routes by incorporating an inverse frequency factor, which penalizes frequently used edges and rewards less frequently used ones. This adaptive mechanism gradually resets excessively amplified distances, promoting diversification and improving the exploration of the solution space. Furthermore, it balances exploitation by focusing on refining the most promising routes based on past tours, leading to faster convergence towards a global optimum. The results clearly demonstrate that while both methods are effective in exploring new routes, the dual-layer approach consistently outperforms the EoH method in terms of solution quality and convergence speed. By incorporating a more nuanced edge adjustment process and dynamically adapting to the problem context, the Dual-layer Self-Evolution LLM Agent achieves superior results in minimizing the total distance, making it a more robust and efficient solution for the TSP.

- **Evolutionary Path of the Dual-Layer Self-Evolution LLM Agent**: As illustrated in Sec. D.6, we trace the evolutionary process of the LLM agent in solving Maximum Vertex Cover (MVC) problem, detailing how heuristic strategies evolve step by step through the inner and outer layers, gradually converging to optimized solutions. Initially, the agent generates simple heuristics that focus on ranking decision variables based on their impact on the objective function and constraint violation, incorporating randomness to encourage exploration. These early strategies serve as a foundation for further refinement. As the process evolves, the outer layer refines the prompt instructions, guiding the inner layer to develop more sophisticated heuristics. The LLM begins to incorporate additional factors, such as the absolute difference from the initial solution and a more nuanced treatment of constraints. This results in improved exploration of the solution space, as well as better handling of both the objective function and constraints. In the later stages, the agent integrates more advanced techniques, such as hybrid methods combining genetic algorithms with local search, to enhance convergence speed and solution quality. The final heuristics represent a co-evolutionary approach that balances exploration and exploitation, leading to significantly optimized solutions. The evolution of prompts, from the initial simplistic forms to highly specialized instructions, demonstrates the power of the dual-layer architecture in improving both the heuristic strategies and the problem-solving process itself.

- **Evolutionary Result of the Dual-Layer Self-Evolution LLM Agent**: Finally, we present the results achieved by the LLM agent after the completion of the entire evolutionary process across three challenging combinatorial optimization problems: Set Covering (SC), Maximum Independent Set (MIS), and Mixed Integer Knapsack Set (MIKS). As detailed in Sec. D.7, the final heuristics generated by the Dual-layer Self-Evolution LLM Agent are compared with those produced by traditional methods and state-of-the-art approaches, demonstrating significant improvements in solution quality and computational efficiency. For the Set Covering problem (SC), the LLM agent's final heuristic achieves a superior balance between minimizing the number of selected sets and satisfying the constraints. By dynamically adjusting penalties and incorporating random exploration, the agent efficiently navigates the solution space, outperforming traditional methods in both the objective score and constraint satisfaction. In the Maximum Independent Set (MIS) problem, the LLM agent leverages simulated annealing principles combined with adaptive scoring of decision variables. This approach not only ensures thorough exploration but also accelerates convergence towards high-quality solutions. The agent's ability to balance objective contributions with constraint violations leads to a considerable reduction in the total error, as reflected in the final objective score. Lastly, for the Mixed Integer Knapsack Set (MIKS) problem, the LLM agent adopts a hybrid strategy that integrates genetic algorithms and simulated annealing. This allows for a more diversified search process, strategically selecting decision variables based on their contributions to the objective function and constraint interactions. The agent's solution demonstrates a significant improvement over existing methods, particularly in how it dynamically adapts to varying problem constraints while maintaining computational efficiency.

In summary, the proposed Dual-layer Self-Evolutionary LLM Agent effectively generates and refines heuristic strategies for diverse combinatorial optimization problems. Leveraging the complementary roles of its inner and outer layers, it balances exploration and exploitation to discover high-quality, context-aware strategies. Its adaptability in evolving both problem-solving heuristics and guiding prompts ensures superior solution quality and computational efficiency. From online

bin packing to large-scale MILP problems, the agent consistently outperforms traditional and state-of-the-art methods, demonstrating robustness, scalability, and evolutionary refinement.

## D.2. Inner and Outer Layer Prompt Initialization and Evolution

---

**Prompt for Generating Initial Heuristic Strategies**
Given an initial feasible solution and a current solution to a Mixed-Integer Linear Programming (MILP) problem, with variables' lower_bound, upper_bound and coefficient in objective function. We want to improve the current solution using Large Neighborhood Search (LNS).

The task can be solved step-by-step by starting from the current solution and iteratively selecting a subset of decision variables to relax and re-optimize. In each step, most decision variables are fixed to their values in the current solution, and only a small subset is allowed to change. You need to score all the decision variables based on the information I give you, and I will choose the decision variables with high scores as neighborhood selection. To avoid getting stuck in local optima, the choice of the subset can incorporate a degree of randomness.

First, describe your new algorithm and main steps in one sentence. The description must be inside a brace. Next, implement it in Python as a function named select_neighborhood. This function should accept 5 input(s): 'initial_solution', 'current_solution', 'lower_bound', 'upper_bound', 'objective_coefficient'. The function should return 1 output(s): 'neighbor_score'. 'initial_solution', 'current_solution', 'lower_bound', 'upper_bound' and 'objective_coefficient' are numpy arrays. 'neighbor_score' is also a numpy array that you need to create manually. The i-th element of the arrays corresponds to the i-th decision variable. All are Numpy arrays. I don't give you 'neighbor_score' so that you need to create it manually. The length of the 'neighbor_score' array is the same as the length of the other arrays.

Do not give additional explanations.

---

**(Cross) Initial Prompt for Heuristic Strategies Evolution**
Given an initial feasible solution and a current solution to a Mixed-Integer Linear Programming (MILP) problem, with variables' lower_bound, upper_bound and coefficient in objective function. We want to improve the current solution using Large Neighborhood Search (LNS).

The task can be solved step-by-step by starting from the current solution and iteratively selecting a subset of decision variables to relax and re-optimize. In each step, most decision variables are fixed to their values in the current solution, and only a small subset is allowed to change. You need to score all the decision variables based on the information I give you, and I will choose the decision variables with high scores as neighborhood selection. To avoid getting stuck in local optima, the choice of the subset can incorporate a degree of randomness.

I have 5 existing algorithm's thought, objective function value with their codes as follows: No.1 algorithm's thought, objective function value, and the corresponding code are: ...
No.2 algorithm's thought, objective function value, and the corresponding code are: ...
...
No.5 algorithm's thought, objective function value, and the corresponding code are: ...

Please help me create a new algorithm that has a totally different form from the given ones.

First, describe your new algorithm and main steps in one sentence. The description must be inside a brace. Next, implement it in Python as a function named select_neighborhood. This function should accept 5 input(s): 'initial_solution', 'current_solution', 'lower_bound', 'upper_bound', 'objective_coefficient'. The function should return 1 output(s): 'neighbor_score'. 'initial_solution', 'current_solution', 'lower_bound', 'upper_bound' and 'objective_coefficient' are numpy arrays. 'neighbor_score' is also a numpy array that you need to create manually. The i-th element of the arrays corresponds to the i-th decision variable. All are Numpy arrays. I don't give you 'neighbor_score' so that you need to create it manually. The length of the 'neighbor_score' array is the same as the length of the other arrays.

Do not give additional explanations.

---

**(Cross) Initial Prompt Strategies**
1. Please help me create a new algorithm that has a totally different form from the given ones.

2. Please help me create a new algorithm that has a totally different form from the given ones but can be motivated from them.

---

**(Cross) Prompt for Prompt Strategies Evolution**
We are working on solving a minimization problem. Our objective is to leverage the capabilities of the Language Model (LLM) to generate heuristic algorithms that can efficiently tackle this problem. We have already developed a set of initial prompts and observed the corresponding outputs. However, to improve the effectiveness of these algorithms, we need your assistance in carefully analyzing the existing prompts and their results. Based on this analysis, we ask you to generate new prompts that will help us achieve better outcomes in solving the minimization problem.

I have 5 existing prompts with objective function value as follows:
No.1 prompt's tasks assigned to LLM, and objective function value are: ...
No.2 prompt's tasks assigned to LLM, and objective function value are: ...
...
No.5 prompt's tasks assigned to LLM, and objective function value are: ...

Please help me create a new prompt that has a totally different form from the given ones but can be motivated from them.

Please describe your new prompt and main steps in one sentence. Do not give additional explanations.

---

## D.3. Heuristic Improvement Through Dual-layer Self-evolutionary LLM Agent

**Heuristic 1 (Obj Score: 5375.52145)**
Rank decision variables based on their penalty contribution and the difference from current solution, incorporating randomness in scoring.

```python
import numpy as np
def select_neighborhood(n, m, k, site, value,
        constraint, initial_solution,
        current_solution, objective_coefficient):
    neighbor_score = np.zeros(n)
    variable_difference = np.zeros(n)
    for i in range(m):
        lhs = sum(value[i][j] * current_solution[
            site[i][j]] for j in range(k[i]))
        penalty = max(0, lhs - constraint[i])
        for j in range(k[i]):
            var_index = site[i][j]
            difference = current_solution[
                var_index] - initial_solution[
                var_index]
            neighbor_score[var_index] += penalty *
                difference
    neighbor_score += objective_coefficient * np.
        random.rand(n)
    return neighbor_score
```

**Heuristic 2 (Obj Score: 5383.05876)**
Rank decision variables based on their objective contribution and impact on current solution deviation, with randomness included in the scoring process.

```python
import numpy as np
def select_neighborhood(n, m, k, site, value,
        constraint, initial_solution,
        current_solution, objective_coefficient):
    neighbor_score = np.zeros(n)
    variable_contribution = np.zeros(n)
    for i in range(m):
        lhs = sum(value[i][j] * current_solution[
            site[i][j]] for j in range(k[i]))
        deviation = lhs - constraint[i]
        for j in range(k[i]):
            var_index = site[i][j]
            contribution = value[i][j] * (
                initial_solution[var_index] -
                current_solution[var_index])
            neighbor_score[var_index] +=
                contribution
    neighbor_score += objective_coefficient + np.
        random.rand(n)
    return neighbor_score
```

**Heuristic 3 (Obj Score: 5384.8486)**
This modified algorithm ranks decision variables based on their contribution to the total current solution's objective function value and their degree of constraint satisfaction.

```python
import numpy as np
def select_neighborhood(n, m, k, site, value,
        constraint, initial_solution,
        current_solution, objective_coefficient):
    neighbor_score = np.zeros(n)
    for i in range(m):
        lhs = sum(value[i][j] * current_solution[
            site[i][j]] for j in range(k[i]))
        for j in range(k[i]):
            if lhs > constraint[i]:
                neighbor_score[site[i][j]] +=
                    objective_coefficient[site[i
                    ][j]] * (lhs - constraint[i])
            else:
                neighbor_score[site[i][j]] +=
                    objective_coefficient[site[i
                    ][j]] * (constraint[i] - lhs)
    neighbor_score += np.random.rand(n) * 0.1
    return neighbor_score
```

**Heuristic 4 (Obj Score: 5384.95417)**
Rank decision variables by their contribution to the objective function and difference from initial values, while also weighing their frequency of use in the constraints.

```python
import numpy as np
def select_neighborhood(n, m, k, site, value,
        constraint, initial_solution,
        current_solution, objective_coefficient):
    score = np.zeros(n)
    frequency = np.zeros(n)
    for i in range(m):
        lhs = sum(value[i][j] * current_solution[
            site[i][j]] for j in range(k[i]))
        deviation = lhs - constraint[i]
        for j in range(k[i]):
            var_index = site[i][j]
            contribution = value[i][j] * np.abs(
                initial_solution[var_index] -
                current_solution[var_index])
            score[var_index] += contribution
            frequency[var_index] += 1
    neighbor_score = score / (frequency + 1e-5) +
        objective_coefficient + np.random.rand(n)
    return neighbor_score
```

**Prompt Designed by LLM**
Develop an algorithm that combines the strengths of existing heuristics while introducing random perturbations to enhance exploration and minimize the objective function more effectively.

**Heuristic (Obj Score: 5374.19865)**
Rank decision variables based on their contribution to the objective function and incorporate the absolute difference from the initial solution while adding a degree of randomness to the scores.

```python
import numpy as np

def select_neighborhood(n, m, k, site, value, constraint, initial_solution, current_solution,
        objective_coefficient):
    neighbor_score = np.zeros(n)
    for i in range(m):
        lhs = sum(value[i][j] * current_solution[site[i][j]] for j in range(k[i]))
        for j in range(k[i]):
            var_index = site[i][j]
            difference = np.abs(current_solution[var_index] - initial_solution[var_index])
            neighbor_score[var_index] += (constraint[i] - lhs) * difference
    neighbor_score += objective_coefficient * np.random.rand(n)
    return neighbor_score
```

## D.4. Heuristic Strategies for Bin Packing Online: EoH vs. Dual-Layer Self-Evolution LLM Agent

---

**Heuristic Designed by EoH**

**Description**

The heuristic incorporates a weighted average of the utilization ratio, dynamic adjustment, and an exponentially decaying factor, with different parameter settings to minimize the number of used bins.

**Code**

```python
import numpy as np
def heuristic(item, bins):
    diff = bins-item # remaining capacity
    exp = np.exp(diff) # exponent term
    sqrt = np.sqrt(diff) # square root term
    ulti = 1-diff/bins # utilization term
    comb = ulti * sqrt # combination of utilization and square root
    adjust = np.where(diff > (item * 3), comb + 0.8, comb + 0.3)
        # hybrid adjustment term to penalize large bins
    hybrid_exp = bins / ((exp + 0.7) *exp)
        # hybrid score based on exponent term
    scores = hybrid_exp + adjust
        # sum of hybrid score and adjustment
    return scores
```

---

**Heuristic Designed by Dual-layer Self-evolution LLM Agent**

**Description**

The new algorithm employs a hybrid optimization strategy that combines nonlinear penalties for historical usage, adaptive capacity scaling, and a relative size assessment, facilitating a balance between local and global search for optimal bin assignment.

**Code**

```python
import numpy as np
def score(item, bins):
    feasible_bins = bins[bins > item]
    scores = np.zeros_like(bins)
    if len(feasible_bins) == 0:
        return scores
    # Nonlinear capacity scaling that enhances the desire for larger
        spaces
    remaining_capacity = feasible_bins - item
    capacity_scaling = np.log1p(remaining_capacity) * (
        remaining_capacity / np.max(remaining_capacity))
    # Relative size assessment: quadratic term comparing item size with
        bin capacities
    relative_size_effect = (item ** 2 / feasible_bins) * 50  # Scale to
        moderate impact
    # Nonlinear penalty based on historical usage counts to deter
        overutilization
    historical_count = np.arange(len(feasible_bins)) + 1  # Simulating
        historical usage
    penalty_factor = np.power(1.5, historical_count)  # Exponential
        penalty for higher usage
    # Combining scores: enhanced capacity scaling, moderated size
        assessment, and historical penalties
    scores[bins > item] = capacity_scaling - relative_size_effect -
        penalty_factor
    return scores
```

## D.5. Heuristic Strategies for Traveling Salesman Problem: EoH vs. Dual-Layer Self-Evolution LLM Agent

**Heuristic Designed by EoH**

**Description**

This algorithm uses a randomized approach to update the edge distance matrix by randomly selecting a proportion of edges to increase their distances while uniformly rewarding a smaller proportion of unused edges to encourage exploration.

**Code**

```python
import numpy as np
def update_edge_distance(edge_distance, local_opt_tour, edge_n_used):
    N = edge_distance.shape[0]
    updated_edge_distance = edge_distance.copy()
    # Parameters for randomization
    increase_factor = 2.0
    decrease_factor = 0.9
    random_selection_ratio = 0.3  # percentage of edges to randomly adjust
    # Identify all edges used in the local optimal tour
    used_edges = set()
    for i in range(len(local_opt_tour)):
        start = local_opt_tour[i]
        end = local_opt_tour[(i + 1) % len(local_opt_tour)]
        used_edges.add((min(start, end), max(start, end)))
    # Randomly select a proportion of edges to increase distance
    all_edges = [(i, j) for i in range(N) for j in range(N) if i != j]
    np.random.shuffle(all_edges)
    num_edges_to_increase = int(len(all_edges) * random_selection_ratio)
    for edge in all_edges[:num_edges_to_increase]:
        start, end = edge
        # If the edge is used in the local optimal tour, apply a higher increase
        if (min(start, end), max(start, end)) in used_edges:
            updated_edge_distance[start, end] *= increase_factor
            updated_edge_distance[end, start] *= increase_factor
        else:
            updated_edge_distance[start, end] *= decrease_factor
            updated_edge_distance[end, start] *= decrease_factor
    return updated_edge_distance
```

**Heuristic Designed by Dual-layer Self-evolution LLM Agent**

**Description**

The new algorithm refines the edge distance adjustment mechanism by incorporating an acceptance heuristic that dynamically explores alternative routes while gradually resetting excessively amplified distances, thus promoting diversification and improved convergence towards a global optimum.

**Code**

```python
import numpy as np
def update_edge_distance(edge_distance, local_opt_tour, edge_n_used):
    # Create a copy of the edge distance matrix for updates
    updated_edge_distance = np.copy(edge_distance)
    # Extract the number of nodes
    num_nodes = edge_distance.shape[0]
    # Calculate the inverse frequency factor for each edge
    inverse_frequency_factor = np.max(edge_n_used) - edge_n_used + 1
    # Update the edge distance based on the local optimal tour
    for i in range(len(local_opt_tour)):
        # Get the current and next node in the local optimal tour
        current_node = local_opt_tour[i]
        next_node = local_opt_tour[(i + 1) % len(local_opt_tour)]
        # Apply the inverse frequency factor to decrease the edge weight
        updated_edge_distance[current_node, next_node] *= inverse_frequency_factor[
            current_node, next_node]
        updated_edge_distance[next_node, current_node] *= inverse_frequency_factor[
            next_node, current_node]
    return updated_edge_distance
```

## D.6. Evolutionary Path of the Dual-Layer Self-Evolution LLM Agent

**Heuristic (Obj Score: 5400.48176)**
The algorithm ranks decision variables based on their impact on the objective function and how they relate to the violated constraints, incorporating a degree of randomness.

**Code**

```python
import numpy as np
def select_neighborhood(n, m, k, site, value, constraint,
    initial_solution, current_solution, objective_coefficient):
    neighbor_score = np.zeros(n)
    violated_constraints = 0
    for i in range(m):
        lhs = sum(value[i][j] * current_solution[site[i][j]] for
            j in range(k[i]))
        if lhs > constraint[i]:
            violated_constraints += 1
            for j in range(k[i]):
                neighbor_score[site[i][j]] +=
                    objective_coefficient[site[i][j]]
    if violated_constraints > 0:
        neighbor_score /= violated_constraints
    randomness = np.random.rand(n) * 0.1
    neighbor_score += randomness
    return neighbor_score
```

**Heuristic (Obj Score: 5374.19865)**
Rank decision variables based on their contribution to the objective function and incorporate the absolute difference from the initial solution while adding a degree of randomness to the scores.

**Code**

```python
import numpy as np
def select_neighborhood(n, m, k, site, value, constraint,
    initial_solution, current_solution, objective_coefficient):
    neighbor_score = np.zeros(n)
    for i in range(m):
        lhs = sum(value[i][j] * current_solution[site[i][j]] for
            j in range(k[i]))
        for j in range(k[i]):
            var_index = site[i][j]
            difference = np.abs(current_solution[var_index] -
                initial_solution[var_index])
            neighbor_score[var_index] += (constraint[i] - lhs) *
                difference
    neighbor_score += objective_coefficient * np.random.rand(n)
    return neighbor_score
```

**Heuristic (Obj Score: 5373.34904)**
Develop a co-evolutionary heuristic approach that integrates genetic algorithms with local search techniques to enhance convergence speed and minimize the objective function for the specified problem.

**Code**

```python
import numpy as np
def select_neighborhood(n, m, k, site, value, constraint,
    initial_solution, current_solution, objective_coefficient):
    neighbor_score = np.zeros(n)
    for i in range(m):
        lhs = sum(value[i][j] * current_solution[site[i][j]] for
            j in range(k[i]))
        for j in range(k[i]):
            var_index = site[i][j]
            difference = np.abs(current_solution[var_index] -
                initial_solution[var_index])
            neighbor_score[var_index] += (constraint[i] - lhs) *
                difference
    random_adjustment = np.random.rand(n)
    adaptive_mutation_rate = np.clip(np.abs(objective_coefficient
        ), 0.1, 1.0)
    neighbor_score += adaptive_mutation_rate * random_adjustment
    return neighbor_score
```

**Initial Prompts**

- (Cross) Please help me create a new algorithm that has a totally different form from the given ones.

- (Cross) Please help me create a new algorithm that has a totally different form from the given ones but can be motivated from them.

- (Variation) Please assist me in creating a new algorithm that has a different form but can be a modified version of the algorithm provided.

- (Variation) Please identify the main algorithm parameters and assist me in creating a new algorithm that has a different parameter settings of the score function provided.

**Current Prompts**

- (Cross) Develop a modified heuristic algorithm that utilizes a hybrid approach, combining elements of simulated annealing and genetic algorithms, to optimize the given minimization problem.

- (Cross) Design a modified heuristic algorithm for the minimization problem by incorporating elements of simulated annealing with a unique cooling schedule.

- (Variation) Please identify the main algorithm parameters and assist me in creating a new algorithm that has a different parameter settings of the score function provided.

- (Variation) Develop an algorithm that combines the strengths of existing heuristics while introducing random perturbations to enhance exploration and minimize the objective function more effectively.

**Final Prompts**

- (Cross) Develop a hybrid heuristic algorithm for the minimization problem that combines genetic algorithms with tabu search to enhance local search capabilities while maintaining diversity in the solution population.

- (Cross) Develop a co-evolutionary heuristic approach that integrates genetic algorithms with local search techniques to enhance convergence speed and minimize the objective function for the specified problem.

- (Variation) Design a novel optimization strategy that integrates genetic algorithms with dynamic programming principles to enhance the search for optimal solutions, focusing on adaptive mutation rates to effectively minimize the objective function value.

- (Variation) Design a novel optimization framework that integrates particle swarm optimization with genetic algorithms, focusing on adaptive mutation strategies to enhance convergence speed and minimize the objective function value.

## D.7. Evolutionary Result of the Dual-Layer Self-Evolution LLM Agent

### D.7.1. EVOLUTIONARY RESULT OF SET COVERING PROBLEM

---

**Heuristic (Obj Score: 3339.39339)**

This algorithm computes scores based on the penalty incurred by each variable when deviating from the current solution and evaluates the impact on constraint satisfaction.

**Code**

```python
import numpy as np
def select_neighborhood(n, m, k, site, value, constraint, initial_solution, current_solution, objective_coefficient):
    neighbor_score = np.zeros(n)
    for i in range(m):
        lhs_value = sum(value[i][j] * current_solution[site[i][j]] for j in range(k[i]))
        for j in range(k[i]):
            variable_index = site[i][j]
            if lhs_value >= constraint[i]:
                penalty = lhs_value - constraint[i]
                contribution = penalty * value[i][j]
                neighbor_score[variable_index] += contribution
            else:
                contribution = value[i][j]
                neighbor_score[variable_index] -= contribution
    costs = np.abs(current_solution - initial_solution) * (objective_coefficient + 1e-5)
    with np.errstate(divide='ignore', invalid='ignore'):
        neighbor_score = np.divide(neighbor_score, costs, where=costs != 0)
    neighbor_score -= np.min(neighbor_score)
    neighbor_score /= np.max(neighbor_score) if np.max(neighbor_score) != 0 else 1
    rand_factor = np.random.rand(n) * 0.1
    neighbor_score += rand_factor
    return neighbor_score
```

---

**Final Prompts**

- (Cross) Please help me create a new algorithm that has a totally different form from the given ones.

- (Cross) Please help me create a new algorithm that has a totally different form from the given ones but can be motivated from them.

- (Variation) Please assist me in creating a new algorithm that has a different form but can be a modified version of the algorithm provided.

- (Variation) Please identify the main algorithm parameters and assist me in creating a new algorithm that has a different parameter settings of the score function provided.

---

### D.7.2. EVOLUTIONARY RESULT OF MAXIMUM INDEPENDENT SET PROBLEM

---

**Heuristic (Obj Score: -4634.0636)**

This new heuristic approach combines the principles of simulated annealing with the adaptive scoring of decision variables based on their contributions to violated constraints while incorporating randomness to enhance exploration of the solution space.

**Code**

```python
import numpy as np
def select_neighborhood(n, m, k, site, value, constraint, initial_solution, current_solution, objective_coefficient):
    neighbor_score = np.zeros(n)
    current_objective_value = np.dot(current_solution, objective_coefficient)
    variable_contributions = np.zeros(n)
    for i in range(m):
        lhs_value = sum(value[i][j] * current_solution[site[i][j]] for j in range(k[i]))
        if lhs_value > constraint[i]:
            for j in range(k[i]):
                var_index = site[i][j]
                variable_contributions[var_index] += (value[i][j] * (current_solution[var_index] == 1))
    for index in range(n):
        improvement = objective_coefficient[index] - variable_contributions[index]
        neighbor_score[index] = improvement + (current_solution[index] * 0.5)
    temperature = np.random.uniform(0.1, 1.0)
    randomness = np.random.uniform(-temperature, temperature, size=n)
    neighbor_score += randomness
    return neighbor_score
```

---

**Final Prompts**

- (Cross) Develop a novel hybrid algorithm that combines local search and simulated annealing techniques to explore the solution space and minimize the objective function more effectively.

- (Cross) Design a novel optimization algorithm inspired by the existing methods, focusing on adaptive parameter tuning to enhance convergence toward better solutions.

- (Variation) Design a novel heuristic approach inspired by the principles of simulated annealing to optimize the following problem parameters.

- (Variation) Please identify the main algorithm parameters and assist me in creating a new algorithm that has a different parameter settings of the score function provided.

---

### D.7.3. EVOLUTIONARY RESULT OF MIXED INTEGER KNAPSACK SET PROBLEM

**Heuristic (Obj Score: -3612.99096)**
This novel algorithm enhances diversity in the solution search process by strategically selecting decision variables based on both their objective contributions and constraint interactions, while incorporating a degree of random exploration.

**Code**

```python
import numpy as np
def select_neighborhood(n, m, k, site, value, constraint, initial_solution, current_solution, objective_coefficient):
    neighbor_score = np.zeros(n)
    contribution_scores = objective_coefficient * current_solution
    neighbor_score += contribution_scores
    for i in range(m):
        lhs_value = sum(value[i][j] * current_solution[site[i][j]] for j in range(k[i]))
        if lhs_value > constraint[i]:
            for j in range(k[i]):
                var_index = site[i][j]
                penalty = (lhs_value - constraint[i]) / max(1, np.sum(value[i]))
                neighbor_score[var_index] -= penalty * value[i][j] * np.random.uniform(0.8, 1.2)
    local_search_factor = (initial_solution - current_solution) ** 2
    neighbor_score += local_search_factor
    randomness = np.random.rand(n) * 0.1
    neighbor_score += randomness
    if np.max(neighbor_score) > 0:
        neighbor_score /= np.max(neighbor_score)
    return neighbor_score
```

**Final Prompts**
- (Cross) Design a hybrid heuristic algorithm that combines elements of genetic algorithms and simulated annealing to explore the solution space efficiently.

- (Cross) Develop a multi-phase heuristic optimization strategy that integrates particle swarm optimization with tabu search to dynamically adapt search parameters and enhance convergence rates.

- (Variation) Develop an algorithm that incorporates a novel optimization strategy, diverging from previous approaches, to enhance the objective function's outcome by exploring alternative parameter tuning techniques.

- (Variation) Please identify the main algorithm parameters and assist me in creating a new algorithm that has a different parameter settings of the score function provided.

## E. Convergence Analysis of LLM-LNS

### E.1. Evolutionary Progress in Combinatorial Optimisation Problem

Across both two combinatorial optimization problems Online Bin Packing and Traveling Salesman Problem, LLM-LNS consistently shows superior convergence and final solution quality compared to EOH.

In the Online Bin Packing problem shown in Figure 5, LLM-LNS shows better convergence behavior from the early stages. As the generations progress, LLM-LNS steadily improves and consistently outperforms EOH. The reduced variance in later generations highlights the stability of the LLM-LNS approach, which efficiently balances exploration and exploitation. Its dual-layer structure allows it to thoroughly explore the solution space, avoiding premature convergence and reaching a higher overall objective score. In contrast, EOH exhibits larger fluctuations and fails to achieve the same level of performance, indicating its limitations in maintaining robust progress during the evolutionary process.

In the Traveling Salesman Problem shown in Figure 6, although LLM-LNS starts with a less favorable initial population compared to EOH, it quickly demonstrates its advantage. Initially, EOH performs better, but it stagnates after the first 8 generations, showing little improvement afterward. Meanwhile, LLM-LNS continues to refine its solutions and steadily decreases the objective score. This indicates that the dual-layer structure of LLM-LNS effectively prevents it from getting trapped in local optima, maintaining a high level of exploration even in later generations. By the end of the evolutionary process, LLM-LNS surpasses EOH, achieving better overall results.

In both problems, LLM-LNS's ability to maintain diversity early in the process, combined with its strong convergence in later stages, gives it a clear advantage over EOH. The dual-layer evolutionary strategy ensures that LLM-LNS avoids stagnation, allowing for continuous improvement and ultimately leading to superior performance in solving combinatorial optimization problems.

### E.2. Convergence Analysis of Generations

In the Online Bin Packing Problem, we conducted 100 generations of iterative training using the proposed dual-layer strategy. Figure 7 shows the convergence trends for both the training and testing scores over these 100 generations. The results

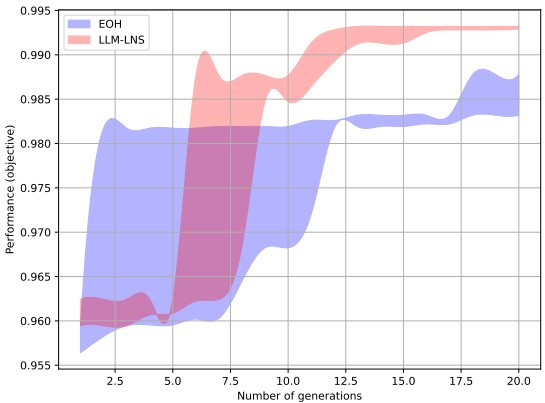

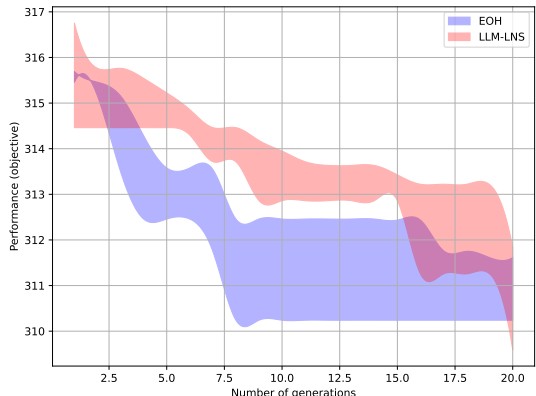

*Figure 5.* Evolutionary Progress of Heuristic Strategies in Online Bin Packing

*Figure 6.* Evolutionary Progress of Heuristic Strategies in Traveling Salesman Problem

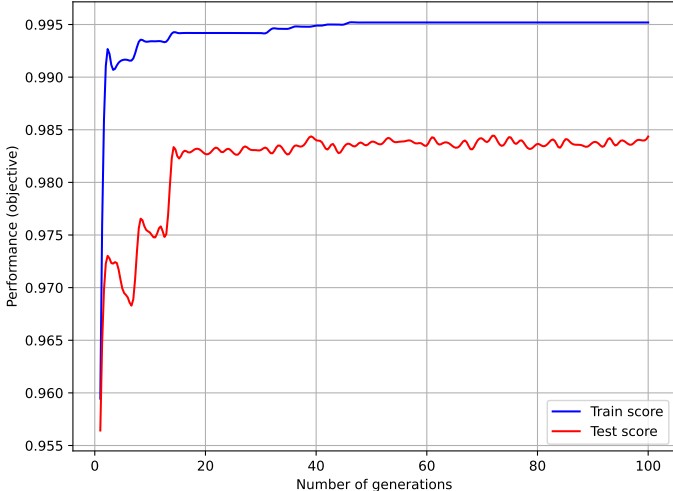

*Figure 7.* Convergence of Training and Testing Scores in 100-Generation of Online Bin Packing Problem.

provide interesting insights into the behavior of our model during the evolutionary process, particularly in terms of how the training and testing losses evolve differently.

The training loss demonstrates a clear and consistent downward trend throughout the generations. Initially, the training score starts relatively high, but quickly drops within the first few generations. This rapid initial improvement indicates that the evolutionary algorithm is highly effective at optimizing the objective function within the training set. As the generations progress, the training score continues to decrease, eventually converging to a very low value. This steady decline suggests that the model is successfully adapting to the problem, continually refining its population and reducing the training objective. The absence of significant fluctuations in later generations implies that the model has reached a stable state, effectively minimizing the training loss with little variance.

On the other hand, the testing loss follows a somewhat different pattern. Initially, we observe a sharp decline in the testing score, which mirrors the behavior of the training score. However, after this initial drop, the testing score does not continue to improve as steadily as the training score. Instead, it stabilizes around a certain value and begins to exhibit small fluctuations.

*Table 7.* Comparison of objective values on large-scale MILP instances across different methods using SCIP as optimizer. For each instance, the best-performing objective value is highlighted in bold. The - symbol indicates that the method was unable to generate samples for any instance within 30,000 seconds, while * indicates that the GNN&GBDT framework could not solve the MILP problem.

| | $SC_1$ | $SC_2$ | $MVC_1$ | $MVC_2$ | $MIS_1$ | $MIS_2$ | $MIKS_1$ | $MIKS_2$ |
|---|---|---|---|---|---|---|---|---|
| Random-LNS | 16164.2 | 171655.6 | 27049.6 | 277255.3 | 22892.9 | 222076.8 | 691.7 | 6870.1 |
| ACP | 17743.4 | 192791.2 | 27432.9 | 281862.4 | 23058.0 | 216008.8 | 29879.2 | 7913.5 |
| CL-LNS | - | - | 31285.0 | - | 15000.0 | - | - | - |
| Gurobi | 17934.5 | 320240.4 | 28151.3 | 283555.8 | 21789.0 | 216591.3 | 32960.0 | 329642.4 |
| SCIP | 25191.2 | 385708.4 | 31275.4 | 491042.9 | 18649.9 | 9104.3 | 29974.7 | 168289.9 |
| GNN&GBDT | 16728.8 | 261174.0 | 27107.9 | 271777.2 | 22795.7 | 227006.4 | * | * |
| Light-MILPOPT | 16147.2 | 166756.0 | 26956.8 | 269771.3 | 22963.6 | 230278.1 | 36125.5 | **357483.8** |
| LLM-LNS(Ours) | **15950.2** | **161732.8** | **26763.4** | **268825.5** | **23137.19** | **230682.8** | **36147.7** | 350468.7 |

This behavior suggests that while the model is able to generalize to a degree, it encounters more variability in the testing data compared to the training data. These fluctuations could be attributed to the inherent complexity or diversity of the unseen test instances, which the model has not been directly optimized for.

This phenomenon is reminiscent of the behavior observed during neural network training, where the training loss continues to decrease as the model becomes more specialized in fitting the training data, while the testing loss reaches a plateau and may exhibit some fluctuations. In this case, the testing loss reflects the model's ability to generalize beyond the training set. The fact that the testing score does not continue to decrease beyond a certain point suggests that the model may have reached its limit in terms of generalization, possibly due to overfitting to the training data. However, the steady fluctuations in the testing score indicate that the model remains adaptable and does not suffer from severe overfitting, as there is no significant increase in the testing loss.

Overall, the divergence between the training and testing scores in later generations highlights the trade-off between optimization and generalization. While the dual-layer evolutionary strategy is highly effective at optimizing the training set, it must also balance the need for generalization to unseen data. The oscillation of the testing score around a stable value suggests that the model is reasonably robust but may benefit from additional techniques to further enhance its generalization performance, such as regularization or early stopping strategies in future iterations.

In summary, the convergence analysis of the 100-generation experiment reveals that while the training loss continues to decrease, the testing loss stabilizes with slight fluctuations. This behavior is indicative of a model that has successfully optimized for the training data while maintaining a reasonable level of generalization, akin to patterns observed in neural network training processes.

## F. Supplementary Experiments for LLM-LNS on Large-Scale MILP Problems

### F.1. Performance of LLM-LNS Using SCIP as the Subsolver

In this supplementary set of experiments, we further evaluate the performance of LLM-LNS by incorporating SCIP as the subsolver for large-scale MILP problems. The results, summarized in Table 7, provide a comprehensive comparison across various methods using SCIP, offering deeper insights into the robustness and adaptability of LLM-LNS when faced with different solver strategies.

As seen in the results, LLM-LNS continues to demonstrate superior performance across most instances, consistently outperforming traditional LNS-based methods, learning-based frameworks such as GNN&GBDT, and even advanced solvers like Gurobi and SCIP. The highlighted bold values indicate that LLM-LNS achieves the best objective values in the majority of cases, reinforcing its scalability and effectiveness in large-scale MILP problems.

However, an interesting observation arises in the MIKS instances, where Light-MILPopt outperforms LLM-LNS. This can be attributed to the unique challenges posed by MIKS in large-scale settings. Specifically, MIKS requires significantly more resources for neighborhood searches as the problem size increases, compared to smaller-scale instances. SCIP, as an optimizer, employs a different strategy for solving MIKS, which likely influences the performance of LLM-LNS when scaling to larger instances. In smaller-scale problems, LLM-LNS may have learned more aggressive strategies that are effective in those scenarios, but these strategies may lead to timeout issues in larger instances due to the increased computational complexity and extended iteration times required for SCIP. As a result, the overall improvement in performance is limited in these larger MIKS problems.

*Table 8.* Comparison of standard deviation values on large-scale MILP instances across different methods using Gurobi as optimizer.

| | $SC_1$ | $SC_2$ | $MVC_1$ | $MVC_2$ | $MIS_1$ | $MIS_2$ | $MIKS_1$ | $MIKS_2$ |
|---|---|---|---|---|---|---|---|---|
| Random-LNS | 37.5 | 258.1 | 88.4 | 243.0 | 72.1 | 243.0 | 98.2 | 584.0 |
| ACP | 38.4 | 1039.3 | 71.6 | 403.5 | 60.3 | 928.8 | 118.2 | 649.2 |
| CL-LNS | - | - | 617.7 | - | 277.5 | - | - | - |
| Gurobi | 28.8 | 143.4 | 77.2 | 287.3 | 48.8 | 147.5 | 69.0 | 225.7 |
| SCIP | 13823.6 | 298211.7 | 107.3 | 262.0 | 57.5 | 85.8 | 73.2 | 242313.7 |
| GNN&GBDT | 360.1 | 3800.4 | 93.8 | 950.4 | 119.3 | 4738.8 | * | * |
| Light-MILPOPT | 1.0 | 145.7 | 79.4 | 209.4 | 52.1 | 133.1 | 41.7 | 272.5 |
| LLM-LNS(Ours) | 17.7 | 144.2 | 79.7 | 198.1 | 55.2 | 147.6 | 70.2 | 170.4 |

*Table 9.* Comparison of standard deviation values on large-scale MILP instances across different methods using SCIP as optimizer.

| | $SC_1$ | $SC_2$ | $MVC_1$ | $MVC_2$ | $MIS_1$ | $MIS_2$ | $MIKS_1$ | $MIKS_2$ |
|---|---|---|---|---|---|---|---|---|
| Random-LNS | 18.8 | 250.3 | 79.0 | 234.8 | 72.1 | 401.7 | 18.1 | 36.2 |
| ACP | 30.8 | 6338.3 | 77.2 | 217.6 | 60.3 | 946.4 | 1829.7 | 943.8 |
| CL-LNS | - | - | 617.7 | - | 277.5 | - | - | - |
| Gurobi | 28.8 | 143.4 | 77.2 | 287.3 | 48.8 | 147.5 | 69.0 | 225.7 |
| SCIP | 13823.6 | 298211.7 | 107.3 | 262.0 | 57.5 | 85.8 | 73.2 | 242313.7 |
| GNN&GBDT | 51.4 | 5587.6 | 91.4 | 474.0 | 80.0 | 660.4 | * | * |
| Light-MILPOPT | 37.7 | 693.4 | 77.3 | 216.9 | 51.6 | 151.7 | 80.0 | 1045.8 |
| LLM-LNS(Ours) | 20.4 | 169.5 | 82.6 | 188.7 | 54.3 | 75.9 | 68.7 | 1197.5 |

Despite these challenges, LLM-LNS still exhibits competitive performance in MIKS, managing to outperform many other methods, including Gurobi and traditional LNS strategies. The occasional time-out or reduced efficiency in MIKS does not overshadow the fact that LLM-LNS remains a robust and scalable solution across a wide range of large-scale MILP problems.

In conclusion, these supplementary experiments highlight the adaptability and robustness of LLM-LNS when using different subsolvers, including SCIP. Although challenges remain in specific problem instances like MIKS, LLM-LNS consistently delivers superior performance across most problem types, demonstrating its ability to generalize across solvers and problem scales. The results reinforce the notion that LLM-LNS effectively bridges the gap between traditional solvers and learning-based methods, offering a scalable solution for large-scale combinatorial optimization problems.

### F.2. Comparison of Standard Deviation Values

The comparison of standard deviation (SD) values across different methods using both Gurobi and SCIP as sub-optimizers reveals several key insights into the stability of various approaches when solving large-scale MILP problems. Standard deviation reflects the consistency of the solutions; lower values indicate that the method is more stable and produces less variation in different runs.

As shown in Table 8, for the experiments using Gurobi, LLM-LNS consistently demonstrates low standard deviation values across most instances, indicating that it not only achieves superior objective values but does so with high stability. For example, in $SC_1$, $MVC_2$, and $MIKS_2$, LLM-LNS has SD values of 17.7, 198.1, and 170.4, respectively, which are comparable to or lower than other methods. Light-MILPopt also shows excellent stability in $SC_1$ and $MIKS_1$, with SD values of 1.0 and 41.7, respectively, although its performance fluctuates more in other instances. In contrast, Random-LNS and ACP exhibit higher variability, especially in $SC_2$ and $MIKS_2$, where ACP's SD reaches as high as 1039.3 and 649.2, respectively, suggesting a lack of robustness in these instances. Gurobi itself also shows moderate consistency, while methods like CL-LNS fail to generate results for certain instances, indicating poor scalability for large problems.

As shown in Table 9, when SCIP is used as the optimizer, the trends remain somewhat similar. LLM-LNS continues to show stable performance, particularly in $SC_1$ and $MVC_2$, with SD values of 20.4 and 188.7, respectively. However, SCIP itself exhibits extremely high variability in some instances, particularly in $SC_2$ and $MIKS_2$, with SD values exceeding 298,000 and 242,000, respectively, which suggests that SCIP struggles with certain large-scale MILPs. This instability in SCIP could be due to its aggressive strategies or solver configurations being less suited to these specific problem instances. Light-MILPopt again demonstrates relatively stable performance in most instances, although its SD increases significantly in some cases, such as $MIKS_2$. GNN&GBDT and ACP also show considerable fluctuations, with ACP having an SD of 6338.3 in $SC_2$, further highlighting its instability in large-scale settings.

In summary, LLM-LNS not only consistently outperforms other methods in terms of objective values but also maintains strong stability across a wide range of instances, particularly when compared to methods like Random-LNS, ACP, and SCIP.

*Table 10.* Comparison of error bar on large-scale MILP instances across different methods using Gurobi as optimizer.

| | $SC_1$ | $SC_2$ | $MVC_1$ | $MVC_2$ | $MIS_1$ | $MIS_2$ | $MIKS_1$ | $MIKS_2$ |
|---|---|---|---|---|---|---|---|---|
| Random-LNS | 65.4 | 318.3 | 142.1 | 350.8 | 104.4 | 333.6 | 158.9 | 808.8 |
| ACP | 56.8 | 1787.2 | 120.6 | 574.8 | 83.6 | 1233.0 | 173.7 | 742.7 |
| CL-LNS | - | - | 892.6 | - | 406.3 | - | - | - |
| Gurobi | 39.7 | 252.7 | 119.6 | 349.0 | 64.7 | 183.1 | 103.8 | 319.7 |
| SCIP | 25238.2 | 533457.2 | 165.2 | 402.1 | 96.9 | 103.6 | 94.6 | 433463.8 |
| GNN&GBDT | 511.3 | 5504.8 | 148.7 | 1522.6 | 160.1 | 7887.9 | * | * |
| Light-MILPOPT | 1.4 | 206.4 | 121.6 | 289.8 | 78.8 | 216.6 | 63.3 | 420.1 |
| LLM-LNS(Ours) | 27.9 | 187.9 | 125.4 | 289.8 | 82.2 | 199.3 | 111.7 | 259.2 |

*Table 11.* Comparison of error bar on large-scale MILP instances across different methods using SCIP as optimizer.

| | $SC_1$ | $SC_2$ | $MVC_1$ | $MVC_2$ | $MIS_1$ | $MIS_2$ | $MIKS_1$ | $MIKS_2$ |
|---|---|---|---|---|---|---|---|---|
| Random-LNS | 33.2 | 362.1 | 123.3 | 368.2 | 104.4 | 531.3 | 26.1 | 51.5 |
| ACP | 46.1 | 10845.3 | 106.0 | 324.1 | 83.6 | 1371.4 | 3253.2 | 1055.6 |
| CL-LNS | - | - | 892.6 | - | 406.3 | - | - | - |
| Gurobi | 39.7 | 252.7 | 119.6 | 349.0 | 64.7 | 183.1 | 103.8 | 319.7 |
| SCIP | 25238.2 | 533457.2 | 165.2 | 402.1 | 96.9 | 103.6 | 94.6 | 433463.8 |
| GNN&GBDT | 72.6 | 7349.2 | 147.2 | 678.6 | 100.4 | 1076.6 | * | * |
| Light-MILPOPT | 66.6 | 1223.3 | 118.5 | 305.6 | 79.1 | 239.4 | 124.2 | 1473.9 |
| LLM-LNS(Ours) | 31.7 | 231.2 | 131.9 | 266.7 | 68.9 | 94.7 | 105.9 | 1868.3 |

This robustness makes LLM-LNS a strong candidate for solving large-scale MILP problems effectively and consistently.

## F.3. Comparison of Error Bar

The error bar comparison across different methods using Gurobi and SCIP as optimizers provides insights into the variability and confidence in solutions across large-scale MILP instances. Error bars quantify the uncertainty or inconsistency in the results, with smaller values indicating more reliable and consistent performance.

As shown in Table 10, for methods using Gurobi, LLM-LNS again demonstrates strong reliability with relatively small error bars across most instances. For example, in $SC_1$, $MVC_2$, and $MIKS_2$, LLM-LNS has error bars of 27.9, 289.8, and 259.2, respectively. These values are noticeably smaller than those for methods like Random-LNS and ACP, which exhibit much larger error bars, reflecting greater instability. Light-MILPopt also shows excellent performance with particularly low error bars in $SC_1$ (1.4) and $MIKS_1$ (63.3), but its error increases significantly in some other instances. Notably, SCIP exhibits extremely large error bars in several instances, such as $SC_2$ and $MIKS_2$, where the error bars exceed 533,000 and 433,000, respectively, indicating significant inconsistency in its performance on these large-scale problems. GNN&GBDT also shows high error bars, suggesting that its performance is less reliable across different runs.

As shown in Table 11, when using SCIP as the optimizer, LLM-LNS continues to demonstrate relatively low error bars, particularly in $SC_1$, $MVC_2$, and $MIKS_1$, where the values are 31.7, 266.7, and 105.9, respectively. These results are significantly more stable compared to methods like ACP and GNN&GBDT, which show very high error bars in instances like $SC_2$ (error bar of 10845.3 for ACP) and $MIKS_2$. SCIP itself again shows extremely high error bars for instances such as $SC_2$ and $MIKS_2$, further highlighting its instability in handling large-scale problems. Light-MILPopt performs well in some instances but also shows considerable variation in others, with error bars as high as 1473.9 in $MIKS_2$.

Overall, LLM-LNS consistently demonstrates lower error bars across both optimizers, Gurobi and SCIP, indicating that it provides more reliable and consistent solutions for large-scale MILP problems. This makes it a strong candidate for scenarios where both solution quality and stability are critical.

## F.4. Convergence Analysis

In this section, we analyze the convergence performance of our proposed approach, our proposed LLM-LNS, in comparison to several baseline methods for solving large-scale MILP problems, including Random-LNS, ACP, Gurobi, GNN&GBDT, and Light-MILPOPT. The experimental results are shown in Figures 8 through 11, which include instances of four different problem types: Set Covering (SC), Maximum Vertex Covering (MVC), Independent Set (IS), and Mixed Integer Knapsack Set (MIKS). We evaluate both medium-scale and large-scale instances using two solvers as sub-optimizer, Gurobi and SCIP.

The analysis of the convergence curves reveals several important observations:

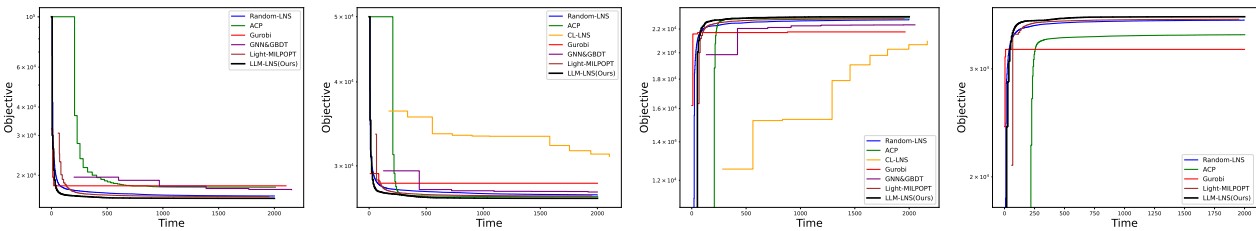

*Figure 8.* Time-objective value graphs of medium-scale problems using Gurobi: $SC_1$, $MVC_1$, $IS_1$, and $MIKS_1$.

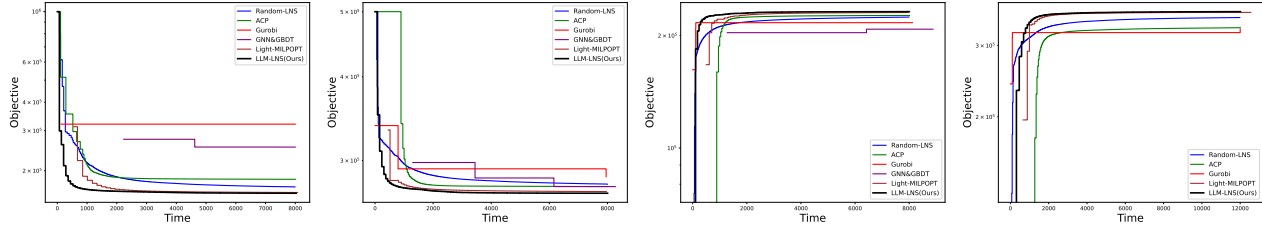

*Figure 9.* Time-objective value graphs of large-scale problems using Gurobi: $SC_2$, $MVC_2$, $IS_2$, and $MIKS_2$.

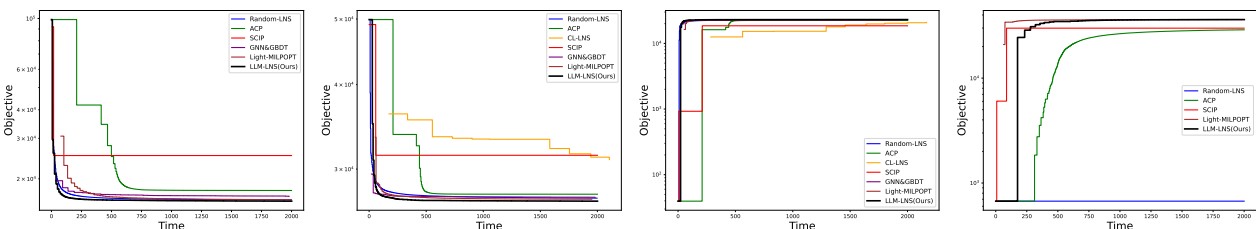

*Figure 10.* Time-objective value graphs of medium-scale problems using SCIP: $SC_1$, $MVC_1$, $IS_1$, and $MIKS_1$.

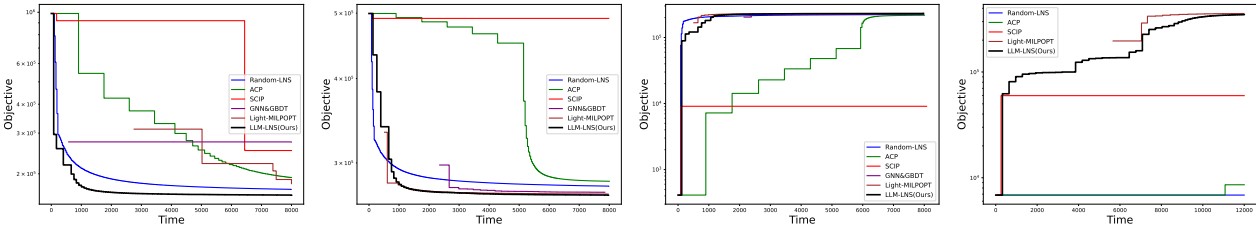

*Figure 11.* Time-objective value graphs of large-scale problems using SCIP: $SC_2$, $MVC_2$, $IS_2$, and $MIKS_2$.

- **Faster Initial Convergence**: For nearly all problem instances, the **LLM-LNS** approach demonstrates a significantly faster initial convergence compared to the baseline methods. The objective value drops sharply within the first few time steps, indicating that our method can quickly identify high-quality solutions. In contrast, methods like **Random-LNS** and **ACP** exhibit slower initial convergence, requiring more time to achieve similar reductions in the objective value.

- **Superior Final Objective Value**: Across both medium- and large-scale problem instances, our proposed LLM-LNS consistently achieves lower final objective values compared to the other methods. This is particularly evident in the large-scale instances (e.g., $SC_2$, $MVC_2$, $IS_2$, and $MIKS_2$), where the superiority of our method becomes more pronounced. While methods such as Random-LNS and ACP plateau early, often with suboptimal solutions, our proposed LLM-LNS continues to improve the solution even after other methods have stagnated.

- **Stable Convergence Behavior**: The convergence curves of our proposed LLM-LNS exhibit smooth and gradual decreases in the objective value, indicating stable optimization behavior. In contrast, some of the baseline methods,

*Table 12.* Performance comparison of LLM-LNS with additional LNS methods on MILP tasks. Results are reported as objective values (lower is better).

| Method | $SC_1$ | $SC_2$ | $MVC_1$ | $MVC_2$ | $MIS_1$ | $MIS_2$ | $MIKS_1$ | $MIKS_2$ |
|---|---|---|---|---|---|---|---|---|
| Random-LNS | 16140.6 | 169417.5 | 27031.4 | 276467.5 | 22892.9 | 223748.6 | 36011.0 | 351964.2 |
| ACP | 17672.1 | 182359.4 | 26877.2 | 274013.3 | 23058.0 | 226498.2 | 34190.8 | 332235.6 |
| Least-Integral | 22825.3 | 228188.0 | 29818.0 | 306567.1 | 20106.9 | 195782.2 | 27196.9 | 241663.4 |
| Most-Integral | 50818.2 | 519685.5 | 35340.5 | 327742.4 | 14584.4 | 157686.5 | 31235.3 | 314621.6 |
| RINS | 26116.2 | 261176.3 | 26851.3 | 306215.6 | 23069.7 | 201178.1 | 30049.1 | 299953.4 |
| LLM-LNS (Ours) | **15802.7** | **158878.9** | **26725.3** | **268033.7** | **23169.3** | **231636.9** | **36479.8** | **363749.5** |

especially Random-LNS and GNN&GBDT, show more erratic convergence patterns, characterized by large and sudden jumps in the objective value. This suggests that our method is more robust and avoids the instability that can arise in heuristic-based search strategies.

- **Scalability**: The performance gap between our proposed LLM-LNS and the baseline methods becomes even more pronounced in large-scale problem instances. For example, in the large-scale $MIKS_2$ and $SC_2$ instances, our proposed LLM-LNS outperforms all other methods by a significant margin, converging to a much lower objective value within a shorter time frame. This demonstrates the scalability of our method, as it remains effective even as the problem size increases, whereas the performance of other methods, such as Light-MILPOPT and ACP, degrades considerably.

- **Comparison with Exact Solvers**: When compared to the exact solver Gurobi, our proposed LLM-LNS shows comparable or even superior performance, particularly in terms of convergence speed. While Gurobi tends to find solutions that improve gradually over time, our proposed LLM-LNS reaches competitive solutions much faster, which is crucial in time-constrained scenarios. This highlights the practical advantage of our method in scenarios where computational resources or time are limited.

In summary, the experimental results demonstrate that **LLM-LNS** has clear advantages in terms of convergence speed, final solution quality, and robustness compared to both heuristic-based and exact optimization methods. Our approach is particularly well-suited for large-scale MILP problems, where it consistently outperforms the baseline methods by a significant margin.

### F.5. Baseline Comparisons with Additional LNS Methods

To evaluate the effectiveness of the proposed LLM-LNS framework, we conducted comprehensive comparisons with several LNS methods that utilize different heuristic scoring functions. Specifically, we incorporated **Least-Integral** (Nair et al., 2020), **Most-Integral** (Berthold, 2006), and **RINS** (Danna et al., 2005), which are classical scoring functions commonly used in LNS frameworks, alongside the state-of-the-art methods **ACP** (Ye et al., 2023a) and classic method **Random-LNS** (Song et al., 2020).

The results, summarized in Table 12, demonstrate that our proposed LLM-LNS consistently outperforms all baseline methods across a variety of MILP tasks, including Set Covering (SC), Maximum Vertex Cover (MVC), Maximum Independent Set (MIS), and Mixed Integer Knapsack Set (MIKS). This advantage highlights the superior ability of LLM-LNS to balance exploration diversity and solution convergence.

From the results in Table 12, several key observations can be made. Among the classical LNS methods, RINS generally achieves better results compared to Least-Integral and Most-Integral, as it leverages neighborhood-based improvements combined with partial solutions. However, these methods still fall significantly behind ACP, Random-LNS, and our proposed LLM-LNS, particularly on larger problem instances such as $SC_2$, $MVC_2$, and $MIKS_2$. For example, on the $SC_2$ problem, RINS achieves an objective value of 261176.3, compared to 158878.9 for LLM-LNS, highlighting the limitations of traditional scoring functions in handling large-scale MILP problems.

ACP and Random-LNS perform much better than the classical scoring-based methods due to their adaptiveness and ability to leverage heuristic diversity. However, even these state-of-the-art baselines are consistently outperformed by LLM-LNS across all tasks. For instance:

- On the $SC_1$ problem, LLM-LNS achieves an objective value of 15802.7, compared to 16140.6 for Random-LNS and 17672.1 for ACP.

- On the $MIS_2$ problem, LLM-LNS achieves 231636.9, compared to 223748.6 for Random-LNS and 226498.2 for ACP.

The clear performance advantage of LLM-LNS can be attributed to its dual-layer architecture, which combines prompt evolution and heuristic strategy optimization to balance search diversity and convergence. The outer layer generates diverse prompts that broaden the search space, avoiding premature convergence to suboptimal solutions. Meanwhile, the inner layer refines heuristic strategies and accelerates convergence by leveraging the evolved prompts. This interaction ensures that LLM-LNS adapts effectively to different problem scales and complexities.

Moreover, the results highlight the scalability of LLM-LNS. While ACP and Random-LNS demonstrate reasonable performance on smaller tasks, their effectiveness diminishes as the problem size increases. In contrast, LLM-LNS maintains its performance advantage across both small-scale (e.g., $SC_1$) and large-scale (e.g., $SC_2$) tasks, showcasing its robustness and adaptability. This scalability is directly enabled by the dynamic feedback loop between the two layers, ensuring continuous refinement of both the search space (via prompt evolution) and the solution strategies (via heuristic evolution).

In summary, the experimental results validate the effectiveness of LLM-LNS over both classical and state-of-the-art LNS baselines. Its dual-layer mechanism provides superior generalization and adaptability, making it a powerful framework for solving diverse and large-scale MILP problems.

## G. Ablation Study of the Dual-Layer Self-evolutionary LLM Agent

This section presents the results of the ablation study conducted to analyze the contributions of the dual-layer framework components: **Prompt Evolution** (outer layer) and **Directional Evolution** (inner layer). The study evaluates the effects of removing or isolating each component on the overall performance of the framework. Specifically, we compare the following variations:

- **Base (EOH):** The baseline Evolution of Heuristic (EOH) method without any modifications.

- **Base + Dual Layer:** The EOH method with the dual-layer structure (**Prompt Evolution** in the outer layer).

- **Base + Differential:** The EOH method with the **Directional Evolution** mechanism (inner layer).

- **Ours:** The complete dual-layer framework incorporating both **Prompt Evolution** and **Directional Evolution**.

We evaluate these variations on datasets of different scales to observe their impact on both small-scale and large-scale problems. The datasets include **1k_C100**, **5k_C100**, **10k_C100**, **1k_C500**, **5k_C500**, and **10k_C500**, representing combinatorial optimization instances of varying sizes. The results are summarized in Table 13.

**Key Observations:**

- **Impact of Prompt Evolution:** Adding the dual-layer structure (Base + Dual Layer) significantly improves performance on large-scale problems, as the outer layer enhances the diversity of the search process through prompt optimization. This is particularly evident in the **10k_C500** dataset, where the error rate decreases from **0.97%** (Base) to **0.39%**.

- **Impact of Directional Evolution:** Incorporating the differential evolution mechanism (Base + Differential) improves performance on small-scale problems by accelerating convergence through more effective crossover and mutation strategies. For example, on the **1k_C100** dataset, the error rate decreases from **4.48%** (Base) to **2.64%**.

- **Synergy of Both Components:** The complete dual-layer framework (Ours) achieves the most balanced improvements across datasets, particularly for larger-scale problems. However, on small-scale datasets like **1k_C100**, the additional exploration introduced by Prompt Evolution can slightly increase the error rate compared to Base + Differential (from **2.64%** to **3.58%**).

These results validate the complementary roles of **Prompt Evolution** and **Directional Evolution** in enhancing both diversity and convergence, demonstrating the effectiveness of the dual-layer framework for solving combinatorial optimization problems of varying scales.

*Table 13.* Ablation study results on various datasets. The table compares the baseline (EOH), the addition of the dual-layer structure (Prompt Evolution, outer layer), the addition of the differential evolution mechanism (Directional Evolution, inner layer), and the complete method (Ours). The best results for each dataset are highlighted in bold.

|  | 1k_C100 | 5k_C100 | 10k_C100 | 1k_C500 | 5k_C500 | 10k_C500 |
|---|---|---|---|---|---|---|
| Base (EOH) | 4.48% | 0.88% | 0.83% | 4.32% | 1.06% | 0.97% |
| Base + Dual Layer | 3.78% | 0.93% | **0.40%** | 3.91% | 0.92% | **0.39%** |
| Base + Differential | **2.64%** | 0.94% | 0.69% | **2.54%** | 0.94% | 0.70% |
| Ours | 3.58% | **0.85%** | 0.41% | 3.67% | **0.82%** | 0.42% |

*Table 14.* Stability evaluation of multiple runs on Bin Packing tasks. Results are reported as error rates (%).

| Method | 1k_C100 | 5k_C100 | 10k_C100 | 1k_C500 | 5k_C500 | 10k_C500 | Avg |
|---|---|---|---|---|---|---|---|
| EOH Run 1 | 4.48% | 0.88% | 0.83% | 4.32% | 1.06% | 0.97% | 2.09% |
| EOH Run 2 | 7.56% | 3.33% | 2.62% | 7.22% | 3.19% | 2.50% | 4.07% |
| EOH Run 3 | 4.18% | 3.24% | 3.35% | 3.79% | 3.12% | 3.21% | 3.48% |
| EOH Avg | **5.41%** | **2.48%** | **2.27%** | **5.11%** | **2.46%** | **2.23%** | **3.33%** |
| Ours Run 1 | 3.58% | 0.85% | 0.41% | 3.67% | 0.82% | 0.42% | 1.63% |
| Ours Run 2 | 2.69% | 0.86% | 0.54% | 2.54% | 0.87% | 0.52% | 1.34% |
| Ours Run 3 | 2.64% | 0.94% | 0.69% | 2.54% | 0.94% | 0.70% | 1.41% |
| Ours Avg | **2.97%↑** | **0.88%↑** | **0.55%↑** | **2.92%↑** | **0.88%↑** | **0.55%↑** | **1.46%↑** |

*Table 15.* Impact of population size on Bin Packing tasks. Results are reported as error rates (%).

| Method | 1k_C100 | 5k_C100 | 10k_C100 | 1k_C500 | 5k_C500 | 10k_C500 | Avg |
|---|---|---|---|---|---|---|---|
| EOH (20) | 4.48% | 0.88% | 0.83% | 4.32% | 1.06% | 0.97% | 2.09% |
| Ours (4) | 3.23% | 0.80% | 0.43% | 3.96% | 1.27% | 0.89% | 1.76%↑ |
| Ours (20) | 3.58% | 0.85% | 0.41% | 3.67% | 0.82% | 0.42% | **1.63%↑** |

# H. Additional Validation Experiments

## H.1. Stability Evaluation of Multiple Runs

To evaluate the stability and consistency of our proposed method, we conducted repeated experiments on the **Bin Packing** task. Specifically, we ran three independent trials for both EoH and our method, and the results are summarized in Table 14.

In our proposed method, the seed heuristic strategies are not hand-crafted. Instead, they are automatically generated by the large language model , which introduces some degree of randomness between runs. Despite this randomness, our method consistently outperforms EoH in both effectiveness and stability. For example, on the **1k_C100**, **10k_C100**, and **10k_C500** test sets, the variance in our results is small, and the average performance is consistently better than EoH.

These results demonstrate that our dual-layer framework, combined with the differential evolution mechanism, effectively enhances both consistency and generalization. Moreover, the smaller variance in our method's results highlights its robustness against the randomness introduced by the seed generation process.

## H.2. Impact of Population Size on Experimental Outcomes

To further analyze the impact of population size on experimental outcomes, we conducted additional experiments on the **Bin Packing** task, testing our method with a reduced population size of 4. The results are summarized in Table 15.

As shown in the table, although the average performance slightly decreases with a smaller population size, our method still outperforms EoH (population size 20). For example, the average error of our method with a population size of 4 is **1.76%**, which is better than EoH's **2.09%**. This demonstrates that our dual-layer framework and differential evolution mechanism exhibit significant robustness and effectiveness, maintaining superior performance even with smaller populations.

## H.3. Performance Comparison with EoH on LNS Tasks

In this subsection, we present a comparison between the proposed LLM-LNS framework and existing method EoH, on large-scale combinatorial optimization tasks. While EoH focus on discovering strategies for combinatorial optimization problems, our LLM-LNS framework is specifically designed to address the challenges of large-scale MILP problems through its dual-layer self-evolutionary mechanism.

We evaluated the methods on the Set Covering (SC) problem, a minimization task, using two large-scale datasets:

*Table 16.* Performance comparison on the $SC_1$ dataset (200,000 variables and constraints). Results are reported as objective values (lower is better).

| Method | Instance$_1$ | Instance$_2$ | Instance$_3$ | Instance$_4$ | Avg |
|---|---|---|---|---|---|
| EOH-LNS | 16114.27 | 16073.72 | 16046.83 | 16074.26 | 16070.15 |
| LLM-LNS (Ours) | **15830.61↑** | **15801.19↑** | **15800.17↑** | **15800.17↑** | **15802.68↑** |

*Table 17.* Performance comparison on the $SC_2$ dataset (2,000,000 variables and constraints). Results are reported as objective values (lower is better).

| Method | Instance$_1$ | Instance$_2$ | Instance$_3$ | Instance$_4$ | Avg |
|---|---|---|---|---|---|
| EOH-LNS | 175358.59 | 174339.78 | 174782.76 | 174026.33 | 174978.20 |
| LLM-LNS (Ours) | **158901.57↑** | **158953.57↑** | **158712.64↑** | **158759.90↑** | **158831.42↑** |

- **$SC_1$**: Instances with 200,000 decision variables and constraints.

- **$SC_2$**: Instances with 2,000,000 decision variables and constraints.

EOH-LNS was selected as the primary baseline for comparison because it generally outperforms FunSearch on combinatorial optimization tasks, as reported in prior literature. This ensures a fair and representative evaluation of our framework. The experimental results are summarized in Tables 16 and 17, where LLM-LNS consistently outperforms EOH-LNS across all test instances. Specifically:

- On the **$SC_1$** dataset (200,000 variables and constraints), LLM-LNS achieves an average improvement of **1.67%** over EOH-LNS.

- On the **$SC_2$** dataset (2,000,000 variables and constraints), the improvement is more pronounced, reaching **9.20%** on average.

These results demonstrate the superior capability of LLM-LNS in solving large-scale optimization tasks, particularly in terms of solution quality. The improvements can be attributed to the dual-layer architecture, which effectively balances search diversity and solution convergence.

The results demonstrate that LLM-LNS consistently outperforms EOH-LNS, particularly on large-scale instances. This improvement is enabled by the dual-layer self-evolutionary mechanism, which dynamically balances exploration and exploitation:

- The **outer layer** generates diverse prompts to broaden search space coverage, preventing premature convergence to suboptimal solutions.

- The **inner layer** refines heuristic strategies and accelerates convergence by leveraging the evolved prompts, ensuring high-quality solutions.

This collaborative interaction between the two layers forms a dynamic feedback loop, enabling continuous learning and adaptation. While EOH-LNS demonstrates strong performance on small-scale combinatorial optimization tasks, its inability to balance exploration and convergence limits its scalability to larger and more complex problems, as evidenced by the significant performance gap on $SC_2$.

### H.4. Comprehensive Evaluation on TSPLib Instances

We evaluated our method on all 87 instances from the TSPLib benchmark to comprehensively assess its performance. As shown in Table 18, our method achieves better results than the EOH baseline on 43 instances, matches EOH on 39 instances, and performs slightly worse on only 5 instances. This demonstrates that our method is not only robust but also generalizes effectively across diverse TSP instances of varying sizes and complexities. On average, the gap from the best-known solutions is reduced from 6.93% for EOH to 6.25% for our method, representing an overall improvement of approximately 10%. These results highlight the superiority of our approach in minimizing the gap to optimality across a wide range of benchmark instances.

*Table 18.* Performance comparison between EOH and our method on TSPLib instances. Results are reported as the gap from the best-known solutions (%). Bold values indicate the better performance, with red for EOH and blue for ours. Green indicates identical performance.

| Instance | EOH Gap | Ours Gap | Instance | EOH Gap | Ours Gap | Instance | EOH Gap | Ours Gap |
|---|---|---|---|---|---|---|---|---|
| pr439 | 2.80% | **1.97%** | pla7397 | **4.28%** | **4.28%** | gr96 | **0.00%** | **0.00%** |
| rd100 | **0.01%** | **0.01%** | rl5934 | **4.25%** | **4.25%** | pcb442 | 1.15% | **0.96%** |
| u2319 | **2.34%** | **2.34%** | gil262 | 0.59% | **0.48%** | pcb3038 | **4.13%** | **4.13%** |
| lin105 | **0.03%** | **0.03%** | fl417 | 0.80% | **0.77%** | tsp225 | 1.39% | **0.00%** |
| fl1400 | 7.66% | **2.28%** | nrw1379 | 3.82% | **2.99%** | d2103 | **1.88%** | **1.88%** |
| kroA150 | **0.00%** | **0.00%** | pcb1173 | 5.07% | **2.91%** | d198 | 0.40% | **0.29%** |
| fl1577 | **5.03%** | **5.03%** | gr666 | 2.17% | **0.00%** | ch130 | **0.01%** | 0.70% |
| kroB100 | **0.00%** | **0.00%** | u1060 | 4.04% | **1.54%** | berlin52 | **0.03%** | **0.03%** |
| eil51 | **0.67%** | **0.67%** | rl1304 | 6.52% | **2.40%** | u2152 | **4.60%** | **4.60%** |
| ulysses16 | **0.00%** | **0.00%** | u724 | 2.85% | **1.13%** | kroD100 | **0.00%** | **0.00%** |
| linhp318 | 3.22% | **2.77%** | pr299 | 0.61% | **0.11%** | rd400 | 2.23% | **0.82%** |
| gr202 | 0.54% | **0.00%** | vm1084 | 3.64% | **1.74%** | rat575 | 3.11% | **1.88%** |
| d1655 | **5.79%** | **5.79%** | ch150 | 0.37% | **0.04%** | pr107 | **0.00%** | **0.00%** |
| kroB200 | **0.23%** | 0.44% | a280 | 2.06% | **0.34%** | d1291 | 6.53% | **2.54%** |
| gr229 | 1.15% | **0.00%** | pr264 | **0.00%** | **0.00%** | pr76 | **0.00%** | **0.00%** |
| d493 | 2.82% | **1.27%** | dsj1000 | 4.28% | **1.06%** | pr136 | 0.09% | **0.00%** |
| rat195 | **0.99%** | 1.37% | att532 | 220.07% | **215.43%** | kroA100 | **0.02%** | **0.02%** |
| ali535 | 0.67% | **0.00%** | ulysses22 | **0.00%** | **0.00%** | kroB150 | 0.08% | **0.01%** |
| bier127 | 0.26% | **0.01%** | kroC100 | **0.01%** | **0.01%** | eil76 | 1.53% | **1.18%** |
| pr124 | **0.00%** | **0.00%** | rl1323 | 4.35% | **1.93%** | p654 | 0.75% | **0.05%** |
| gr431 | 1.93% | **0.00%** | rl1889 | **4.08%** | **4.08%** | d657 | 2.85% | **1.02%** |
| eil101 | 2.59% | **2.08%** | fnl4461 | **4.63%** | **4.63%** | pr2392 | **4.19%** | **4.19%** |
| rat783 | 4.48% | **2.18%** | ts225 | **0.00%** | **0.00%** | u1432 | 4.84% | **3.02%** |
| u1817 | **4.62%** | **4.62%** | lin318 | 1.46% | **1.09%** | rl5915 | **3.96%** | **3.96%** |
| att48 | **215.43%** | **215.43%** | st70 | **0.31%** | **0.31%** | rat99 | **0.68%** | **0.68%** |
| fl3795 | **4.38%** | **4.38%** | burma14 | **0.00%** | **0.00%** | u159 | **0.00%** | **0.00%** |
| kroA200 | **0.25%** | 0.62% | u574 | 2.85% | **1.38%** | pr1002 | 3.27% | **1.16%** |
| pr152 | **0.00%** | 0.19% | gr137 | 0.11% | **0.00%** | pr226 | 0.10% | **0.06%** |
| vm1748 | **4.33%** | **4.33%** | pr144 | **0.00%** | **0.00%** | kroE100 | **0.00%** | **0.00%** |

The improvements are particularly evident on larger and more challenging instances. For example, on fl1400, our method reduces the gap from 7.66% (EOH) to 2.28%, showcasing its scalability and effectiveness in handling complex optimization problems. Similarly, on the pcb1173 instance, the gap decreases from 5.07% (EOH) to 2.91%, validating the ability of our method to outperform EOH on instances with higher complexity. Even on medium-sized instances such as pr439, our method demonstrates significant improvements, reducing the gap from 2.80

In addition to these improvements, we also observe instances where both methods achieve comparable performance. For example, on smaller problems such as eil51, ulysses16, and kroD100, both EOH and our method report identical gaps, demonstrating that our method maintains competitive performance even on instances where EOH performs optimally. Furthermore, the results highlight the consistency of our approach across various instance scales, from small to large.

There are only a few exceptions where EOH slightly outperforms our method. For example, on ch130, EOH achieves a gap of 0.01%, whereas our method reports 0.70%. However, these cases are rare, occurring in only 5 instances out of 87, and do not substantially impact the overall trend of improvement demonstrated by our method.

Overall, our method exhibits strong generalization across the TSPLib benchmark and consistently achieves lower average gaps compared to EOH. The significant improvements on larger and more complex instances further underscore the scalability and effectiveness of our dual-layer architecture. By balancing exploration and exploitation, our method demonstrates its capability to address the challenges posed by diverse and large-scale optimization problems, making it a reliable alternative to state-of-the-art methods such as EOH.

### H.5. Impact of the Backbone Algorithm on Performance

This section addresses whether the proposed method is sensitive to the choice of the backbone algorithm. Our study focuses on solving large-scale MILP problems, where heuristic methods play a critical role due to the complexity of the problem space. Among these methods, LNS has demonstrated significant advantages in scalability and efficiency, especially for large-scale problems. In this context, we selected ALNS (Adaptive Large Neighborhood Search) as the backbone of our framework. ALNS, as a variant of LNS, dynamically adjusts neighborhood sizes to balance exploration and exploitation, making it more effective than non-adaptive LNS methods, which often struggle with local optima in large-scale problems.

To validate this choice, we conducted experiments replacing ALNS with non-adaptive LNS in our framework. The results,

*Table 19.* Comparison of ALNS (adaptive) and non-adaptive LNS as the backbone algorithm in our framework. Results are reported as objective values (lower is better).

| Method | $SC_1$ | $SC_2$ | $MVC_1$ | $MVC_2$ | $MIS_1$ | $MIS_2$ | $MIKS_1$ | $MIKS_2$ |
|---|---|---|---|---|---|---|---|---|
| Without Adaptive | 15957.0 | 160510.8 | 26850.3 | 269701.8 | 23073.2 | 230497.4 | 36330.8 | 362496.3 |
| LLM-LNS (Ours) | **15802.7** | **158878.9** | **26725.3** | **268033.7** | **23169.3** | **231636.9** | **36479.8** | **363749.5** |

*Table 20.* Performance comparison of LLM-LNS using different LLMs on the 10k_C500 dataset. Results are reported as the gap from the best-known solutions (%). Lower values indicate better performance.

| LLM Model | $Run_1$ | $Run_2$ | $Run_3$ | Avg. |
|---|---|---|---|---|
| gpt-4o-mini | 0.42% | 0.52% | 0.70% | 0.55% |
| gpt-4o | 0.33% | 0.58% | 0.39% | 0.43% |
| deepseek | 0.83% | 0.52% | 0.38% | 0.58% |
| gemini-1.5-pro | 0.63% | 1.91% | 0.53% | 1.02% |
| llama-3.1-70B | 2.87% | 3.98% | 0.88% | 2.58% |

summarized in Table 19, show that ALNS consistently outperforms non-adaptive LNS across all tested MILP instances. For example, on the $SC_1$ problem, the objective value achieved by ALNS is **15802.7**, compared to **15957.0** for non-adaptive LNS. Similarly, on the $MVC_2$ problem, ALNS achieves an objective value of **268033.7**, whereas non-adaptive LNS reports **269701.8**. These results highlight the critical role of adaptive mechanisms in ALNS for leveraging the full potential of our framework.

### H.6. Robustness of LLM-LNS with Different LLMs

We conducted experiments to evaluate the robustness of the LLM-LNS framework across various large language models, including GPT-4o, GPT-4o-mini, DeepSeek, Gemini-1.5-Pro, and Llama-3.1-70B. These experiments were performed on the **10k_C500** dataset, and the results are summarized in Table 20.

The results demonstrate that the dual-layer structure of LLM-LNS adapts effectively to different LLMs, achieving reasonable performance across all tested models. GPT-4o consistently achieved the best results, showing the lowest average gap of **0.43%**, followed by GPT-4o-mini (**0.55%**) and DeepSeek (**0.58%**). Gemini-1.5-Pro and Llama-3.1-70B exhibited relatively weaker performance, with average gaps of **1.02%** and **2.58%**, respectively. These variations are likely due to differences in model architecture and pretraining quality. Nonetheless, the framework demonstrated strong general robustness, with all models performing adequately within the LLM-LNS structure.

These findings underscore the necessity of combining LLMs with a structured optimization framework to fully leverage their potential.

### H.7. Comparison with ReEvo

We conducted additional experiments to compare our proposed method with ReEvo (Ye et al., 2024), a contemporary hyper-heuristic framework that combines reflection mechanisms and evolutionary search. Both methods were evaluated on the Bin Packing problem using the lightweight language model GPT-4o-mini, with the number of iterations fixed at 20 and the population size set to 20.

In the experiments, ReEvo exhibited poor stability when using GPT-4o-mini. Out of 138 attempts, only 3 runs successfully completed all 20 iterations, while the remaining runs were prematurely terminated due to invalid offspring generated during certain generations. Upon analysis, we identified severe hallucination issues in ReEvo. Although its reflection mechanism was effective in capturing evolutionary directions through pairwise comparisons of parent strategies stored in short-term memory, any errors in reflection led to a rapid decline in the quality of subsequent offspring. For example, ReEvo frequently attempted to call nonexistent libraries or use invalid function parameters, resulting in invalid heuristic algorithms and the termination of the evolutionary process.

In contrast, our method adopts a differential memory mechanism that generalizes beyond pairwise comparisons. Rather than relying on two parent strategies, our approach analyzes differences across multiple parent solutions and their corresponding objective values. This richer context enables the language model to conduct contrastive learning, allowing it to internalize more robust and directional evolutionary signals. In addition, differential memory is embedded within a dual-layer agent architecture, where the upper-layer agent optimizes prompt structures while the lower-layer agent generates new strategies. This coordinated design facilitates deeper interaction between memory-guided learning and heuristic generation, significantly

improving both performance and stability.

To ensure a fair comparison, we selected the 3 successful ReEvo runs and compared their performance with our method. Under the default setting, ReEvo utilized an expert seed algorithm to initialize its population. However, after 20 iterations, the best-performing algorithm in ReEvo remained its initial expert seed algorithm, failing to generate superior heuristic strategies. Furthermore, when the expert seed algorithm was removed, ReEvo's solution quality deteriorated further, with its average performance on the Bin Packing problem falling significantly behind our method.

The experimental results are shown in Table 21. Our method demonstrates substantial advantages under the same settings. In terms of solution quality, our approach consistently outperformed ReEvo across all test instances of the Bin Packing problem, with even greater advantages in scenarios without expert seed algorithms. Additionally, our method exhibited significant stability advantages, consistently completing 20 iterations and generating high-quality heuristic strategies without being affected by the hallucination issues observed in ReEvo. The collaborative optimization between the agents in our dual-layer architecture effectively balances search diversity and efficiency, delivering superior performance and higher robustness.

*Table 21.* Performance comparison between ReEvo and our proposed method on the Bin Packing problem. Average percentages represent the error rates.

| | 1k_C100 | 5k_C100 | 10k_C100 | 1k_C500 | 5k_C500 | 10k_C500 | Avg |
|---|---|---|---|---|---|---|---|
| ReEvo Run1 | 3.78% | 0.80% | 0.33% | 6.75% | 1.47% | 0.74% | 2.31% |
| ReEvo Run2 | 3.78% | 0.80% | 0.33% | 6.75% | 1.47% | 0.74% | 2.31% |
| ReEvo Run3 | 3.78% | 0.80% | 0.33% | 6.75% | 1.47% | 0.74% | 2.31% |
| **ReEvo Avg** | **3.78%** | **0.80%** | **0.33%** | **6.75%** | **1.47%** | **0.74%** | **2.31%** |
| ReEvo-no-expert Run 1 | 4.87% | 4.08% | 4.09% | 4.50% | 3.91% | 3.95% | 4.23% |
| ReEvo-no-expert Run 2 | 4.87% | 4.08% | 4.11% | 4.50% | 3.90% | 3.97% | 4.24% |
| ReEvo-no-expert Run 3 | 4.87% | 4.08% | 4.09% | 4.50% | 3.91% | 3.95% | 4.23% |
| **ReEvo-no-expert Avg** | **4.87%** | **4.08%** | **4.10%** | **4.50%** | **3.91%** | **3.96%** | **4.24%** |
| Ours Run1 | 3.58% | 0.85% | 0.41% | 3.67% | 0.82% | 0.42% | 1.63% |
| Ours Run2 | 2.69% | 0.86% | 0.54% | 2.54% | 0.87% | 0.52% | 1.34% |
| Ours Run3 | 2.64% | 0.94% | 0.69% | 2.54% | 0.94% | 0.70% | 1.41% |
| **Ours Avg** | **2.97%↑** | **0.88%↑** | **0.55%↑** | **2.92%↑** | **0.88%↑** | **0.55%↑** | **1.46%↑** |

In conclusion, our method not only outperforms ReEvo in terms of experimental results but also demonstrates significant advantages in stability and robustness. By integrating differential memory into a dual-agent framework, our approach goes beyond short-term comparisons and enables deeper, contrastive learning across diverse search trajectories. This innovation opens up a new avenue for applying LNS to large-scale optimization problems, surpassing existing state-of-the-art solutions in both quality and reliability.

### H.8. Performance Comparison with EoH Using Different LLMs

To evaluate the adaptability and effectiveness of our proposed method across different language models, we conducted experiments comparing our framework with EoH on the *10k_C500* dataset using three LLMs: GPT-4o-mini, GPT-4o, and DeepSeek. EoH was chosen as the baseline based on existing literature, which suggests it generally outperforms FunSearch on combinatorial optimization tasks.

The results of the experiments are summarized in Table 22. Our method consistently outperformed EoH across all tested LLMs. Notably, our approach demonstrated significant advantages when using GPT-4o-mini and DeepSeek. For instance, with GPT-4o-mini, our framework achieved an average performance of **0.55%**, which is approximately four times better than EoH's **2.23%**. Similarly, under DeepSeek, our method achieved an average performance of **0.58%**, significantly outperforming EoH's **1.77%**.

One particularly interesting observation is the poor convergence of EoH under DeepSeek. In both $Run_2$ and $Run_3$, EoH's fitness function values during evolution were much lower than those achieved by our framework. This highlights the limitations of EoH's framework in adapting to certain LLMs, where errors in evolution can significantly impact its performance. In contrast, our dual-layer architecture, combined with differential evolution, demonstrates robust and stable performance across all tested LLMs.

These findings underscore the superiority of our approach in leveraging the capabilities of different LLMs for combinatorial optimization tasks. The dual-layer structure not only enhances adaptability but also ensures consistent performance, addressing the convergence and stability issues observed in EoH. We believe these results further validate the effectiveness

*Table 22.* Performance comparison between EoH and our proposed method on the *10k_C500* dataset using different LLMs. Average percentages represent the error rates.

| 10k_C500 | Run$_1$ | Run$_2$ | Run$_3$ | Avg. |
|---|---|---|---|---|
| gpt-4o-mini (EOH) | 0.97% | 2.50% | 3.21% | 2.23% |
| gpt-4o-mini (Ours) | 0.42% | 0.52% | 0.70% | **0.55%**↑ |
| gpt-4o (EOH) | 0.50% | 0.41% | 0.58% | 0.50% |
| gpt-4o (Ours) | 0.33% | 0.58% | 0.39% | **0.43%**↑ |
| deepseek (EOH) | 0.32% | 3.06% | 1.92% | 1.77% |
| deepseek (Ours) | 0.83% | 0.52% | 0.38% | **0.58%**↑ |

*Table 23.* Performance comparison between standalone LLMs and our proposed framework on the Bin Packing problem. Average percentages represent the error rates.

| | 1k_C100 | 5k_C100 | 10k_C100 | 1k_C500 | 5k_C500 | 10k_C500 | Avg |
|---|---|---|---|---|---|---|---|
| Sample Run1 | 5.32% | 4.40% | 4.44% | 4.97% | 4.27% | 4.28% | 4.61% |
| Sample Run2 | 7.51% | 2.30% | 1.74% | 9.47% | 4.58% | 3.99% | 4.93% |
| Sample Run3 | 5.32% | 4.40% | 4.44% | 4.97% | 4.27% | 4.28% | 4.61% |
| **Sample Avg** | **6.05%** | **3.70%** | **3.54%** | **6.47%** | **4.37%** | **4.18%** | **4.72%** |
| Ours Run1 | 3.58% | 0.85% | 0.41% | 3.67% | 0.82% | 0.42% | 1.63% |
| Ours Run2 | 2.69% | 0.86% | 0.54% | 2.54% | 0.87% | 0.52% | 1.34% |
| Ours Run3 | 2.64% | 0.94% | 0.69% | 2.54% | 0.94% | 0.70% | 1.41% |
| **Ours Avg** | **2.97%**↑ | **0.88%**↑ | **0.55%**↑ | **2.92%**↑ | **0.88%**↑ | **0.55%**↑ | **1.46%**↑ |

and scalability of our method across diverse settings.

### H.9. Comparison with Standalone LLMs

To further validate the effectiveness of our framework, we conducted a comparative experiment against standalone LLMs. Specifically, we replaced all crossover and mutation operations in our framework with instances where the problem information was directly input into a standalone GPT-4o-mini model, which independently generated new strategies and evaluated them. Both approaches were tested on the Bin Packing problem across 20 iterations, with the same total number of strategies generated in each case.

The results, summarized in Table 23, demonstrate that our framework significantly outperforms the standalone LLM approach across all test instances. On average, our framework achieves an error rate of **1.46%**, which is approximately **69% lower** than the standalone LLM's average error rate of **4.72%**. This improvement is primarily due to the dynamic interaction between the outer and inner layers in our framework, which balances exploration and exploitation, ensuring the generation of diverse and high-quality strategies. In contrast, the standalone LLM approach frequently generated redundant or identical strategies, thereby limiting its ability to effectively explore the solution space.

Additionally, we observed that the standalone LLM approach struggled to maintain diversity as the number of iterations increased, resulting in many duplicate strategies and a subsequent decline in optimization performance. In contrast, our framework, through evolutionary operations such as crossover and mutation, maintains diversity within the population, enabling it to achieve superior optimization outcomes with the same number of generated strategies.

These findings confirm that our framework not only improves decision-variable ranking and optimization compared to standalone LLMs but also addresses key limitations such as diversity and redundancy. By integrating evolutionary mechanisms into the LLM-based framework, our approach ensures more efficient use of computational resources and delivers superior performance across various problem instances.

### H.10. Efficiency Comparison of Evolution Time

To complement our performance evaluation, we conducted a detailed comparison of the evolution time between our method and a representative baseline, EOH. Both methods were run over 20 generations on the Bin Packing and Traveling Salesman Problem (TSP) tasks. Evolution time is measured as the total wall-clock time spent on generating and evaluating new prompt strategies during each generation. As our method operates in a learning-based paradigm, all evolutionary computations occur in the training stage and do not affect inference-time efficiency.

The results are summarized in Tables 24 and 25. Overall, our method achieves comparable or lower evolution time than EOH

*Table 24.* Evolution time (in minutes) comparison on the Bin Packing task over 20 generations.

| Method | 1 | 2 | 3 | 4 | 5 | 6 | 7 | 8 | 9 | 10 |
|--------|------|------|------|------|------|------|------|------|-------|-------|
| Ours | 9.9 | 10.5 | 11.1 | 11.7 | 12.3 | 12.9 | 49.3 | 51.1 | 51.7 | 53.8 |
| EOH | 17.9 | 38.5 | 46.8 | 47.4 | 69.5 | 78.2 | 88.1 | 88.7 | 108.7 | 109.3 |

| Method | 11 | 12 | 13 | 14 | 15 | 16 | 17 | 18 | 19 | 20 |
|--------|-------|-------|-------|-------|-------|-------|-------|-------|-------|-------|
| Ours | 56.4 | 57.1 | 59.5 | 61.5 | 62.3 | 80.0 | 83.6 | 136.9 | 137.7 | 144.7 |
| EOH | 110.0 | 110.5 | 111.3 | 111.9 | 117.2 | 117.8 | 118.4 | 137.4 | 138.1 | 138.7 |

*Table 25.* Evolution time (in minutes) comparison on the TSP task over 20 generations.

| Method | 1 | 2 | 3 | 4 | 5 | 6 | 7 | 8 | 9 | 10 |
|--------|------|------|------|------|------|------|------|------|------|------|
| Ours | 3.8 | 9.3 | 11.6 | 14.2 | 16.6 | 23.1 | 35.5 | 37.2 | 38.9 | 40.6 |
| EOH | 6.0 | 9.8 | 12.3 | 16.0 | 18.4 | 20.9 | 23.5 | 26.0 | 28.4 | 30.6 |

| Method | 11 | 12 | 13 | 14 | 15 | 16 | 17 | 18 | 19 | 20 |
|--------|------|------|------|------|------|------|------|------|------|------|
| Ours | 42.2 | 44.2 | 46.0 | 47.8 | 49.6 | 51.6 | 53.3 | 58.5 | 65.5 | 67.3 |
| EOH | 33.1 | 35.4 | 39.1 | 43.2 | 45.6 | 48.1 | 50.4 | 52.9 | 55.7 | 59.6 |

*Table 26.* Comparison of objective values on real-world datasets from MIPLIB 2017 under a 3000s time limit. Lower is better. "–" indicates no feasible solution was found. "*" indicates the method failed due to architectural incompatibility.

| Instance | Random-LNS | ACP | CL-LNS | Gurobi | GNN&GBDT | Light-MILPOpt | LLM-LNS (Ours) |
|----------|-----------|-----------|--------|----------|----------|---------------|----------------|
| dws (min.) | 189028.6 | 186625.3 | – | 146411.0 | * | 147417.7 | **143630.5** |
| ivu (min.) | 14261.3 | 9998.6 | – | 27488.0 | * | – | **3575.9** |

across most generations. Notably, in the Bin Packing task, our approach achieves faster evolution in the earlier generations and maintains efficiency in later ones, while EOH exhibits significant runtime increases as generations progress.

We observe that EOH tends to exhibit higher evolution time in many generations, particularly in the Bin Packing task. This is largely due to the nature of the algorithms it evolves: EOH often produces solutions with $\mathcal{O}(n^2)$ computational complexity, such as pairwise comparison-based packing or route construction procedures. These strategies, while potentially effective, impose significant overhead during evaluation. In contrast, our framework encourages the emergence of simpler, efficient strategies with linear $\mathcal{O}(n)$ complexity. By guiding evolution through performance-based feedback and maintaining diversity through controlled variation, our method reduces redundant exploration of expensive algorithms and ensures faster convergence. This advantage becomes especially clear in later generations, where EOH's runtime escalates, while our approach remains computationally manageable.

These findings confirm that our method not only outperforms in solution quality, but also sustains competitive or superior efficiency in the evolutionary process, making it a practical and scalable choice for large-scale combinatorial optimization.

### H.11. Evaluation on Realistic Industrial Benchmarks

To assess the practical value of our proposed LLM-LNS framework, we extend our evaluation beyond academic benchmarks to include two real-world problem classes from the MIPLIB 2017 dataset: dws and ivu. These classes are representative of large-scale industrial applications, each consisting of multiple instances with consistent structure and domain semantics.

Unlike many other MIPLIB instances that are either too heterogeneous or too small for learning-based heuristics to generalize meaningfully, the dws and ivu classes provide a favorable setting for systematic experimentation. We use the largest instance in each class as the test case, while the remaining instances are used for training when applicable. All methods are evaluated with a time budget of 3000 seconds.

The results are summarized in Table 26. LLM-LNS consistently achieves superior performance, producing better solutions than classical LNS variants, strong commercial solvers such as Gurobi, and recent learning-based approaches. Notably, it is the only method that finds high-quality solutions for both benchmarks, including the challenging ivu instance where Gurobi and other baselines fall short. These results highlight the ability of LLM-LNS to generalize to realistic and large-scale MILP problems.

*Table 27.* **Comparison of objective values on large-scale MILP instances across different methods.** For each instance, the best-performing objective value is highlighted in bold. The "–" symbol indicates that the method was unable to generate samples for any instance within 30,000 seconds, while "*" indicates that the framework could not solve the MILP problem.

| Method | $SC_1$ | $SC_2$ | $MVC_1$ | $MVC_2$ | $MIS_1$ | $MIS_2$ | $MIKS_1$ | $MIKS_2$ |
|---|---|---|---|---|---|---|---|---|
| Random-LNS | 16140.6 | 169417.5 | 27031.4 | 276467.5 | 22892.9 | 223748.6 | 36011.0 | 351964.2 |
| ACP | 17672.1 | 182359.4 | 26877.2 | 274013.3 | 23058.0 | 226498.2 | 34190.8 | 332235.6 |
| CL-LNS | – | – | 31285.0 | – | 15000.0 | – | – | – |
| CP-SAT (LNS only) | 16053.5 | 177565.1 | 26765.8 | 268773.4 | 23159.1 | 231091.3 | * | * |
| Gurobi | 17934.5 | 320240.4 | 28151.3 | 283555.8 | 21789.0 | 216591.3 | 32960.0 | 329642.4 |
| SCIP | 25191.2 | 385708.4 | 31275.4 | 491042.9 | 18649.9 | 9104.3 | 29974.7 | 168289.9 |
| CP-SAT (Full Solver) | 16036.8 | 177413.5 | 26771.8 | 269741.6 | 23152.4 | 230711.3 | * | * |
| GNN&GBDT | 16728.8 | 252797.2 | 27107.9 | 271777.2 | 22795.7 | 227006.4 | * | * |
| Light-MILPOpt | 16108.1 | 160015.5 | 26950.7 | 269571.5 | 22966.5 | 230432.9 | 36125.5 | 362265.1 |
| **LLM-LNS (Ours)** | **15802.7** | **158878.9** | **26725.3** | **268033.7** | **23169.3** | **231636.9** | **36479.8** | **363749.5** |

## H.12. Comparison with Advanced LNS in CP-SAT

To strengthen the comprehensiveness of our baseline comparisons, we additionally include the LNS capabilities of CP-SAT, a well-known solver developed by Google OR-Tools. CP-SAT contains a highly engineered portfolio of neighborhood search strategies, including random, sequential, and adaptive mechanisms, and has been recognized as a strong non-learning-based LNS system. This makes it a relevant point of reference for evaluating the effectiveness of LLM-LNS.

We evaluate two configurations of CP-SAT: the default full solver and a version with `use_lns_only=True`, both executed with 16 threads to activate its full range of neighborhood heuristics. These settings allow us to assess CP-SAT both as a general MILP solver and as a dedicated LNS system. Results are presented in Table 27, using the same time limit as other baselines.

CP-SAT demonstrates competitive performance on easier problems such as Maximum Independent Set (MIS) and Minimum Vertex Cover (MVC), achieving results that are comparable to or better than traditional heuristics. However, on structurally complex and large-scale MILP tasks like Set Covering (SC), its performance noticeably lags behind learning-based approaches. Additionally, due to architectural constraints, CP-SAT does not support general mixed-integer problems, and thus fails to provide results for the MIKS task.

Despite the strong engineering behind CP-SAT, our proposed LLM-LNS consistently outperforms all baselines across problem types. These findings reinforce the effectiveness and generalization ability of LLM-guided local search, particularly when facing high-dimensional combinatorial structures and diverse constraints.

# I. Limitations and Future Directions

While the proposed dual-layer self-evolutionary framework has demonstrated strong performance in solving large-scale MILP problems, we acknowledge several limitations that warrant further exploration and improvement. Below, we discuss these limitations in detail and outline potential future directions.

First, although the framework exhibits good generalization ability on MILP and certain combinatorial optimization problems, it is currently tailored to specific optimization scenarios. The design primarily focuses on MILP and does not directly extend to other types of optimization tasks, such as nonlinear optimization or dynamic optimization problems. Developing a more general agent structure that can adapt to a wider range of optimization algorithms and tasks remains an open challenge. Future work could explore more modular and flexible designs to enhance the adaptability of the framework for solving diverse and complex optimization problems.

Second, the current method leverages the generative capabilities of large language models (LLMs) and evolutionary mechanisms for heuristic strategy design. However, it does not fully incorporate domain knowledge or classical optimization expertise into the framework. In practical optimization tasks, domain-specific knowledge and traditional optimization techniques (e.g., heuristic rules or mathematical programming methods) often play a critical role. A key direction for future research is to explore how to effectively integrate the generalization capabilities of LLMs with optimization domain knowledge to create more efficient and robust algorithms. Such integration could not only improve computational efficiency but also reduce the resource overhead for solving ultra-large-scale problems.

Third, the fitness function currently employed for evaluating heuristic strategies relies on a simple unnormalized averaging of objective values. While this approach has proven effective for the random instances studied in this paper, where objective values likely possess similar orders of magnitude, it may introduce bias when dealing with more heterogeneous sets of problems. In such scenarios, instances with larger objective values could disproportionately influence the evolutionary process, potentially causing the heuristic to overfit to them. A pertinent direction for future work involves exploring more sophisticated fitness evaluation mechanisms. This could include the development of multi-metric fitness functions, incorporating measures such as the gap estimation integral or k-step improvement rates, to ensure a more balanced and comprehensive assessment of heuristic performance across diverse instance characteristics.

Finally, computational resource constraints remain a practical challenge for solving large-scale problems. While the proposed framework demonstrates good scalability, solving ultra-large-scale instances still requires significant computational time and hardware resources, which may limit its applicability in resource-constrained environments. Future research could focus on optimizing the computational complexity of the algorithm or designing more efficient resource allocation strategies to address these challenges.

