# OpenReview forum: "Large Language Model-driven Large Neighborhood Search for Large-Scale MILP Problems"
_ICML.cc/2025/Conference — ICML 2025 spotlightposter_

### Official Review · Reviewer_4X7L · 2025-02-20

**Overall Recommendation:** 3

**Summary:**

This paper proposes LLM-LNS, a framework leveraging Large Language Models (LLMs) to drive Large Neighborhood Search (LNS) for solving large-scale Mixed Integer Linear Programming (MILP) problems including Online Bin Packing, Traveling Salesman Problems, SC, MVC, MIS, and MIKS. The main novelty lies in a dual-layer self-evolutionary LLM agent that generates and refines heuristic strategies for neighborhood selection. Experiments validate superior performance over state-of-the-art methods like FunSearch, EOH, and Gurobi.

## Update after rebuttal
I maintain my review after the rebuttal.

**Claims And Evidence:**

### Key Points
1. The author claims that the proposed LLM-LNS framework surpasses traditional LNS methods, advanced solvers like Gurobi and SCIP, and modern ML-based frameworks. However, the dataset used in the comparison is four academic ones, so a more realistic dataset like MIPLIB 2017 should be used to validate its effectiveness.

**Essential References Not Discussed:**

A few recent works like "BTBS-LNS: Binarized-Tightening, Branch and Search on Learning LNS Policies for MIP" are closely related but are not discussed in the paper.

**Experimental Designs Or Analyses:**

I would like to know the settings of Gurobi and SCIP because most of the solving time is spent on proving optimality.

**Methods And Evaluation Criteria:**

The authors introduce a dual-layer LLM agent, which evolves heuristic strategies and evolutionary prompts. The framework adapts neighborhood size dynamically based on search progress, allowing for efficient exploration of large solution spaces.

**Other Comments Or Suggestions:**

None

**Other Strengths And Weaknesses:**

**Strengths:**
-  The proposed framework is highly innovative and demonstrates clear improvements over existing methods.

**Weakness:**
- more realistic large-scale datasets like MIPLIB 2017 should be tested.

**Questions For Authors:**

None

**Relation To Broader Scientific Literature:**

This paper aligns with recent advances in combining LLMs and optimization heuristics, following the trend of using machine learning to automate the design of optimization strategies.

**Theoretical Claims:**

None

---

> ### Author Rebuttal · Authors · 2025-04-01
>
> We sincerely thank Reviewer 4X7L for the thoughtful and constructive feedback. We are glad that you find the proposed LLM-LNS framework innovative and recognize its performance improvements over existing methods. We appreciate your suggestions regarding dataset realism, solver settings, and related work, and we address each of these points in detail below.
>
> **Claims And Evidence and Weakness:**
>
>
> **A1:** Thank you very much for the helpful suggestion. We fully agree that evaluating on realistic datasets is important. While MIPLIB 2017 is a valuable benchmark, many of its instances are either heterogeneous or not large enough to meaningfully evaluate LLM-driven heuristics. Therefore, we carefully selected two problem classes from MIPLIB — dws and ivu — which both contain multiple large-scale instances of the same type, allowing for a meaningful train/test split and consistent evaluation.
>
> Specifically, we used the largest instances from each class as the test set, and the remaining instances for training. All methods were evaluated under a 3000s time limit. The results below demonstrate that LLM-LNS generalizes well to realistic, large-scale industrial problems. We will include these experiments in the appendix and plan to evaluate broader MIPLIB categories in future work.
>
> |               | Random-LNS |ACP| CL-LNS |  Gurobi  | GNN&GBDT | Light-MILPOpt | LLM-LNS(Ours) |
> | :-----------: | :--------: | :------: | :----: | :------: | :------: | :-----------: | :-----------: |
> | dws(Minimize) |  189028.6  | 186625.3 |   -    | 146411.0 |    *     |   147417.7    | **143630.5**  |
> | ivu(Minimize) |  14261.3   |  9998.6  |   -    |  27488   |    *     |       -       |  **3575.9**   |
>
>
> **Experimental Designs Or Analyses:**
>
> **A1:** Thank you for the question. We used the default settings for both Gurobi and SCIP. To ensure a fair comparison, all baselines were run under the same computational constraint: 16 cores and 32 threads within a fixed time limit. We will include a description of these settings in the Experimental Setup section of the revised manuscript.
>
> **Supplementary Material:**
>
> **A1:** Thank you very much for the suggestion! We agree that moving Algorithm 1 into the main text would help clarify the overall workflow of our method. We will include it in the main paper if space permits during the revision.
>
> **Essential References Not Discussed:**
>
> **A1:** Thank you very much for pointing this out. BTBS-LNS is a recently accepted paper at ICLR 2025, which proposes a reinforcement learning-based LNS framework with a binarized-tightening and branching mechanism for solving MIP problems. While the approach is related in spirit, the code is not publicly available at this time. Moreover, the experiments in BTBS-LNS are mainly conducted on relatively small-scale problems (typically with tens of thousands of decision variables), and our previous experience suggests that reinforcement learning struggles to scale effectively in much larger problem settings, which are the main focus of our work. We will add a discussion of BTBS-LNS in the Related Work section, and if the code is released in time, we will consider including a comparative experiment in the final version.
>
> We sincerely appreciate Reviewer 4X7L’s insightful comments and constructive suggestions. They have helped us significantly improve the clarity and completeness of our paper. We have carefully addressed each point in our revision and will incorporate the proposed changes into the final version. Thank you again for your thoughtful review and valuable feedback.

---

### Official Review · Reviewer_r212 · 2025-03-13

**Overall Recommendation:** 4

**Summary:**

This paper describes an evolutionary LLM-based framework to produce heuristics represented as code. It extends previously-proposed evolutionary methods for heuristic search with two additional techniques: prompt evolution and directional evolution. Prompt evolution focuses on finding good LLM prompts to produce heuristic strategies, and directional evolution uses meta-prompts to provide strategy fitness values to LLMs as feedback to improve strategies. This is applied for both typical combinatorial heuristics, evaluated on online bin packing and the traveling salesman problem, and for variable selection heuristics in Large Neighborhood Search in MILP, evaluated in four typical MILP problem classes. Computational results are encouraging when compared with previous non-ML, ML, and LLM-based approaches.

## Update after rebuttal

I maintain my review after the rebuttal.

**Claims And Evidence:**

There are two main methodological contributions: 1, the strategy-search addition to the stack of evolutionary LLM layers for searching for heuristics to solve a combinatorial optimization problem, which enable the LLM to diversify and control its own search strategies, and 2, the use of LLM-based heuristic search in the context of Large Neighborhood Search. Their effectiveness is well-supported by a comprehensive set of experiments in various forms: only the LLM framework without the LNS for online bin packing and TSP in Sec. 4.1, ablation studies on each LLM strategy in Appendix G, and the full framework with LNS in Sec. 4.2. The appendix also contains other supporting experiments, such as variations in population size, variations in LLMs, and stability.

Experiments on online bin packing and TSP show that this stack of evolutionary methods can improve upon previous similar approaches by a reasonable margin, and the ablation studies confirm that both prompt evolution and directional evolution make a significant difference. These improvements may not always be consistent in the sense that turning off one or the other could be better, but they appear to be consistently better than not using either approach. The computational results on LNS are also very positive and encouraging. In general, the computational evidence is comprehensive and covers a good number of different scenarios.

**Essential References Not Discussed:**

I do not have further references to add.

**Experimental Designs Or Analyses:**

The experimental setup appears sound. As mentioned above, I consider these experimental results to be very positive, but I have a few comments and suggestions for improvement.

1. Looking at the results from Table 4, I first thought that you might be slightly overstating the contributions: my read of Table 4 is that LLM-LNS is a clear advantage over Gurobi and SCIP, but it is only a slight advantage to Light-MILPOPT. However, I believe your results are actually stronger than it may first appear from reading the main text.

    * This is more visible in Fig. 8: even if the results are fairly close at the time limit, LLM-LNS can obtain very good solutions early on, which is valuable in practice. I would like to make three recommendations to emphasize the strength of your own results: 1, show Figure 8 in the main text if you can find some space, 2, report the primal integral as well, and 3, include a sentence on this in the main text. The primal integral is a common way in the OR/MILP literature to quantify heuristic performance over time (I left another comment on this below).

    * Furthermore, I would actually consider simply being competitive with the ML-based approaches a success: in certain use cases, the result of LLM-based approaches can be preferable over typical neural network approaches because they are significantly easier to deploy and maintain. From a software engineering / systems perspective, it is easier to copy-and-paste and maintain a short snippet of a code rather than needing to store and maintain neural network data for an algorithm. Of course, they have different obstacles in the "training" step, so this opinion might not be prevalent or suitable for every use case, but this is definitely a relevant advantage for those interested in practical deployment. Perhaps this is something you might want to note in the paper.

Next, I discuss some areas that could have been better covered:

2. I was looking for results on the amount of time and compute needed to produce these heuristics, and I could not find them. I am not sure if I missed them, but I believe these are important to include (even if you only have estimates). Please include them if they are not in the paper, or let me know where they are if they already are.

3. If I understand correctly, I do not see a per-subproblem time limit in LNS. This is a little surprising for a number of reasons: 1, practical LNS implementations typically have some time limit in subproblem solves so it does not blow up, especially when they accidentally set the neighborhood size too high; 2, although the fitness function would filter these out, you may be wasting time running slow code proposed by LLMs; 3, as noted in Section F.1, this is risky if you are trying to generalize to larger instances. I believe a subproblem time limit can help make the approach more robust. That said, it is encouraging to hear that this method works well even without a per-subproblem time limit. I understand that this is not something that you can do at rebuttal, but it feels omitted and a brief discussion about this somewhere in the paper would be useful to at least highlight that this is something that can be done.

4. Another topic that I felt could be better covered in the paper is the fitness function. The paper proposes simple unnormalized averaging. This works well for the random instances studied in this paper because all objective values likely have about the same order of magnitude, but this is not always true for more heterogeneous sets of problems. In those cases, instances with larger orders of magnitude in objective value may be much more emphasized than those with smaller ones, and thus the heuristic may overfit to them. While simple averaging is fine for this work, I feel it is appropriate to include a short discussion about this in the fitness function section in the Appendix.

5. There is little discussion on the input parameters to the variable scoring heuristic, which I feel is an important topic. Looking at the examples, I see names like "site", "value", "constraint", but I cannot figure out what exactly they are. Even those that are self-explanatory, it would be nice to know how they are represented. Could you please describe what they are in the paper? In particular, why did you choose these parameters and not others? Could you have done better if you have chosen others, or even omitted some (as this affects how the LLM behaves)? Please add a discussion of this topic to the paper.

**Methods And Evaluation Criteria:**

The LLM strategy search makes sense: the heuristic search itself might require some guidance of which directions to explore. The example in Figure 3 is useful to understand what it is doing, and it is very interesting to see how the prompt strategy can guide the heuristic strategy towards certain known algorithmic directions. While it is always tricky to truly evaluate LLM behavior with high-level methods like these, the ablation tests suggest meaningful improvements. I suspect that this could also lead to more interpretable heuristics since the heuristic search is directed by broad algorithmic strategies, but this study is not present in the paper (it would be very interesting to see, but I do not expect it for this rebuttal).

The LNS research direction is also promising. Based on my own experience as an OR practitioner, I have had the prior belief that if there is any entry point to combine LLM-based function search with OR solvers, LNS is probably one of the most suitable ones. Variable selection strategies in LNS are flexible enough so that improvements in functions can produce significant impact, simple in scope enough to let LLMs work without a lot of complexity, and leverage the full generality of OR solvers. It is encouraging to see this in action and working well.

As far as I know, both of these approaches are novel, though I am more familiar with the LNS literature than the LLM one.

The computational evaluation appears to be careful and generally comprehensive. The experiments cover a good variety of both problems and baselines. For the MILP instances, I tried to get a sense of how realistic these instances are, but unfortunately I could not find how these instances were generated. Based on the number of variables and constraints, they seem rather sparse (e.g. 3x constraints on independent set or vertex cover means that the average degree of the graph is 6). That said, while ideally it would be better to have more realistic applications, it is not a significant concern.

I do have some concerns related to the LNS baselines:

1. I am surprised that your CL-LNS baseline is performing so poorly, which is not what you would see in (Huang et al., 2023b). Do you know why? I do notice that you are solving much larger instances than in that paper, but it is still a little surprising and I wonder what exactly is the issue.

2. If the behavior of CL-LNS is indeed correct, then I would argue that this paper is missing something closer to state-of-the-art LNS. If random LNS is your best LNS baseline, then I do not think you have SOTA LNS in your experiments. What I suggest is adding CP-SAT as a baseline to the paper. While CP-SAT does a lot more than LNS, it contains a competitive LNS implementation and it does not require you to have to implement yet another LNS algorithm. You have two options here, use it as a full blown solver (which would belong in the Gurobi/SCIP category), or if you want only its LNS capabilities for purposes of comparison, you can turn on the use_lns_only parameter. Make sure to increase the number of workers from the default as the various types of LNS are only active with a certain number of workers (I suggest 16, or 32 if you want; this may depend on how many threads other baselines are using for a fair comparison). It is of course not an ML-based approach, so it may not perform better than other ML-based approaches, but I expect it to be a better baseline than random LNS, since it implements random LNS and other LNS techniques.

**Other Comments Or Suggestions:**

Here are other minor comments, mostly on presentation:

1. In Figs 7-10, Light-MILPOPT seems to sometimes become worse. I am guessing you are taking the current solution, but this is not the right way to present the result, since you can always keep the best solution. Please correct these plots to report the best solution value found so far in each time step, rather than the current one.

2. Table 4 is difficult to read since improvements on each column depend on whether it is a minimization or maximization problem. An option here is to use primal gap instead, which is traditional in MILP. See, for example, Table 2 in the CL-LNS paper (Huang et al., 2023b in your paper). If you do not want to use it, at least add some indicator on whether you are minimizing or maximizing in each column.

3. As previously discussed, primal integral would be great to see as well (see Berthold, "Measuring the impact of primal heuristics"). You seem to have the data to compute that based on Figs. 7-10. Please note comment 1: use the data for the best solution found so far to compute the primal integrals, not the current value like Light-MILPOPT is showing in the submitted version.

4. The ablation study is important to understand the impact of the prompt evolution and the directional evolution. Space permitting, consider adding a sentence in the main text summarizing the results, and point to the appendix.

5. Could you see if you could fit into the start of Sec. 4.1 a couple of sentences explaining the heuristic framework for online bin packing and TSP, i.e. a very brief summary of what you have in Appendix C.4? In my readthrough, I got confused as I thought you were using LNS for these problems, but you are not using LNS here. I believe this misunderstanding is easy to make, so I would even encourage to explicitly add in Sec. 4 or 4.1 that this is not using LNS for these problems.

6. Could you mark in Table 17 which TSPLib instances were used for training? This should be easy enough to add and it would be helpful to get some more information on generalization.

7. In Table 4, please add to the caption the difference between set 1 and 2.

8. Sec. 4.1.2: I presume that you are using the routing solver in Google OR-Tools and not CP-SAT. Please include for clarity, since there are multiple solvers in OR-Tools.

9. Section C.2: Could you please add here that $T$ is for the entire LNS solve (i.e. not for each subproblem), and that you solve each subproblem to optimality? I believe this is the case.

10. Section C.5 seems to be missing information on the instance generation process.

11. Would you be willing to provide in the final version the optimal heuristics for all six problems? This can be included with the code. It would be interesting to see if there is anything interpretable in those heuristics. Furthermore and optionally, it might be insightful to highlight in the paper the algorithmic directions that appeared in prompts of the best heuristics for each problem class.

**Other Strengths And Weaknesses:**

The paper is generally comprehensive and well-written. Overall, I believe this paper is a solid step towards effectively leveraging LLMs in practical OR applications.

**Questions For Authors:**

All my questions are throughout the above sections. While I am recommending acceptance, there are a few gaps in the paper and I am assuming the authors can provide a reasonable rebuttal. Please go through them carefully and address them, as I believe it can strengthen the paper.

**Relation To Broader Scientific Literature:**

I believe the paper does a decent job at relating to previous work.

**Theoretical Claims:**

The paper has no theoretical claims.

---

> ### Author Rebuttal · Authors · 2025-04-01
>
> We sincerely thank Reviewer r212 for the detailed, thoughtful, and constructive feedback. We greatly appreciate your recognition of the contributions and your suggestions for improving the clarity, robustness, and completeness of the work.
>
> **Methods And Evaluation Criteria:**
>
> **A1:**  We agree that fully evaluating high-level LLM behavior is challenging. As suggested, we will include the full evolution traces of both strategies and prompts in the appendix to improve interpretability. Regarding instance realism, as noted in our response to Reviewer 4X7L, we have added experiments on two large-scale MIPLIB problem classes to further validate the robustness of our method.
>
> **A2:** CL-LNS was designed for small-scale MILPs and struggles to scale to our large instances with hundreds of thousands of variables. Training takes over 8 hours per instance, and inference is 30× slower than our method, making it impractical. Following your suggestion, we added CP-SAT (16 threads, both original and use_lns_only) as a new baseline. As shown in Table 1 at the supplementary link, CP-SAT supports only integer programs and performs well on easier problems like MIS and MVC. However, on more complex large-scale tasks such as SC${}_{2}$, its performance lags behind several other baselines. **Our method still outperforms all baselines across tasks.** Due to time constraints, further analysis of CP-SAT—including using it as a subsolver to enhance LLM-LNS—will be included in future work.
>
> **Experimental Designs Or Analyses:**
>
> **A1:** We agree your suggestions help better highlight our method’s strengths. We will revise the paper to include Figure 8 in the main text if space permits. Following your advice, we also report the primal integral in Supplementary Table 2, which further confirms the effectiveness of LLM-LNS. We appreciate your point on the deployment advantage of LLM-based methods and will add a note on this practical benefit in the revision.
>
> **A2:**  Our method evolves prompt strategies during training, not inference, so it does not affect solving efficiency. Although we add a prompt evolution layer, its total time remains comparable to EOH, with minor overhead from fitness evaluation via testing. Evolution times (in minutes) for BinPacking and TSP over 20 generations are reported in Rebuttal to Reviewer G8Pp, and will be added to the appendix.
>
> **A3:** You're absolutely right—setting a time limit per subproblem is important for robustness. In our implementation, we do impose a maximum time per LNS iteration: 100 seconds for problems with ~100K variables, and 200 seconds for ~1M variables. We will clarify this in the experimental setup section of the paper.
>
> **A4:** We agree that unnormalized averaging may be biased on heterogeneous datasets. In ongoing work, we explore multi-metric fitness (e.g., gap estimation integral, k-step improvement rate). We will briefly discuss this in the Appendix as a direction for future work.
>
> **A5:** The terms site, value, and constraint describe the constraint matrix: involved variables, coefficients, and RHS. We'll clarify this in the paper. In ongoing work, we observe that giving full constraint details may lead to strategies with high time complexity. We're exploring feature extraction to simplify inputs, and will discuss this in the Appendix.
>
> **Other Comments Or Suggestions:**
>
> **A1&A3:** We have updated the plots to report the best-so-far solution at each time step. The corrected results are shown in Appendix Figures 1–4. The primal integral results (based on the best-so-far solutions) have been added in Appendix Table 2.
>
> **A2:** We have added the minimization or maximization indicator for each problem in Appendix Table 1. If helpful, we can further include a primal gap comparison in the Appendix to make the results clearer.
>
> **A4&5&8&9:** We agree this will make the paper clearer, and will revise the corresponding sections.
>
> **A6:** We confirm that, following the same setup as EOH and FunSearch, we use five TSPLib instances (d198, eil76, rat99, rl1889, and u1060) as the training set for evolving our policies. None of the other TSPLib instances used in evaluation are seen during training.  We will update Table 17 to clearly mark this.
>
> **A7:** We have explained the difference between set 1 and set 2 in Appendix Table 5 (Sec. C.5). We'll also add a sentence in the main text to clarify this and point to the appendix.
>
> **A10:** We will revise Appendix C.5 to include a clear description of the instance generation process.
>
> **A11:** We will include the final heuristics in the code and highlight key algorithmic directions in the appendix to aid interpretability and better understand LLM behavior.
>
> Due to the rebuttal length limit, supplementary figures and tables are provided at https://anonymous.4open.science/r/Supplementary-Figures-and-Tables-8DBB/. If any part of our response is unclear or insufficient, we would be very happy to further clarify and continue the discussion.

---

> > ### Comment · Reviewer_r212 · 2025-04-03
> >
> > Thank you for the excellent rebuttal. I am happy that the authors are able to address all my concerns. I believe these changes will strengthen the quality of the paper, especially the addition of the CP-SAT and MIPLIB baselines. I maintain my recommendation of acceptance.

---

### Official Review · Reviewer_fpWD · 2025-03-14

**Overall Recommendation:** 4

**Summary:**

The authors propose a Large Language Model (LLM)-driven LNS framework for large-scale MILP problems. Their approach introduces a dual-layer self-evolutionary LLM agent to automate neighborhood selection, discovering effective strategies with scant small-scale training data that generalize well to large-scale MILPs. The inner layer evolves heuristic strategies to ensure convergence, while the outer layer evolves evolutionary prompt strategies to maintain diversity. Experimental results demonstrate that the proposed dual-layer agent outperforms state-of-the-art agents such as FunSearch and EOH. It also achieves superior performance compared to advanced ML-based MILP optimization frameworks like GNN & GBDT and Light-MILPopt.

## update after rebuttal：I'd like to keep the score unchanged.

**Claims And Evidence:**

Claims are supported by clear and convincing evidence.

**Essential References Not Discussed:**

References are complete.

**Experimental Designs Or Analyses:**

Experiments are sound.

**Methods And Evaluation Criteria:**

The evaluation criteria make sense.

**Other Comments Or Suggestions:**

None

**Other Strengths And Weaknesses:**

The paper was well written with extensive experiments and supporting material.

**Questions For Authors:**

None

**Relation To Broader Scientific Literature:**

The method is interesting and extends LLM reasoning to MILP.

**Theoretical Claims:**

No theoretical results.

---

> ### Author Rebuttal · Authors · 2025-04-01
>
> We sincerely thank Reviewer fpWD for the positive and encouraging feedback. We are glad that you found our method interesting and the experimental design sound. Your recognition of the contributions and the clarity of our work is truly appreciated and motivates us to further refine and extend our research.

---

### Official Review · Reviewer_G8Pp · 2025-03-19

**Overall Recommendation:** 4

**Summary:**

This paper proposes LLM-LNS, a Large Language Model-driven Large Neighborhood Search framework for solving large-scale MILP problems. The method introduces a dual-layer self-evolutionary LLM agent: an inner layer evolves heuristic strategies to ensure convergence, while an outer layer optimizes evolutionary prompts to maintain diversity and avoid local optima. Experiments demonstrate that LLM-LNS outperforms state-of-the-art solvers like Gurobi, ML-based LNS methods, and heuristic evolution frameworks such as FunSearch and EOH. The approach shows strong generalization from small-scale training instances to large-scale MILP problems, achieving superior performance in combinatorial optimization tasks like bin packing and the Traveling Salesman Problem.

**Claims And Evidence:**

see the section Strengths And Weaknesses

**Essential References Not Discussed:**

Essential References are well-discussed

**Experimental Designs Or Analyses:**

Reasonable experimental designs and analyses

**Methods And Evaluation Criteria:**

Yes

**Other Comments Or Suggestions:**

See the questions part.

**Other Strengths And Weaknesses:**

### **Strengths**

1. The paper is well-organized and easy to follow, with a clear structure that enhances readability.
2. Unlike many current works that primarily follow the pipeline of EoH, this paper introduces significant contributions to the LLM-based heuristic evolution pipeline. It proposes an outer layer to evolve prompting strategies, which is novel, and introduces directional evolution based on differential memory. Ablation studies confirm the effectiveness of these innovations.
3. The paper includes comprehensive experiments for the ablation study, thoroughly evaluating each component of the pipeline.

### **Weaknesses**

I believe this is a generally good paper. I really like the idea of prompt evolution. I have the following concerns about this paper, which if well-addressed will lead to an even better paper.

1. Although various problem classes are considered, the experiments are conducted on selected instances rather than using a standard instance distribution, as is typical in the learning-for-optimization community.
2. A key baseline, ReEvo [1], is omitted, which is essential given the state-of-the-art context of this work.
3. The reflection procedure appears similar to the directional evolution based on differential memory. Clarification of their differences is needed.

[1] ReEvo: Large Language Models as Hyper-Heuristics with Reflective Evolution

**Questions For Authors:**

Besides the questions in the Strengths And Weaknesses section, I also have the following questions:

1. How are the prompting strategies evolved? Are fixed strategies employed in the evolution process?
2. What is the “evolution time” for this pipeline, and how does it compare with previous works?

**Relation To Broader Scientific Literature:**

This paper situates itself within the broader scientific literature by addressing key challenges in solving large-scale Mixed Integer Linear Programming (MILP) problems, a critical area in optimization research.

**Theoretical Claims:**

N/A

---

> ### Author Rebuttal · Authors · 2025-04-01
>
> We sincerely thank Reviewer G8Pp for the thorough review and positive evaluation of our work.  Your encouraging comments and constructive suggestions are highly appreciated and have helped us further improve the clarity and completeness of the paper.
>
> In the following, we address your concerns and questions point by point.
>
> **Strengths And Weaknesses:**
>
> **A1:** Thank you for this valuable comment. We would like to clarify that the instances used in our experiments are not arbitrarily selected, but are generated based on standard formulations of canonical MILP problems, as described in Appendix C.5. For each problem class and size range, instances are randomly sampled from the same distribution. Training and testing sets are formed via random partitioning, which ensures fair evaluation and avoids selection bias. We will make this clearer in the main text and explicitly refer readers to Appendix C.5, where we have also added pseudocode for the instance generation process.
>
> **A2:**  Thank you very much for your suggestion! We agree that ReEvo [1] is an important baseline. As detailed in Appendix H.7, we conducted a comparative experiment on the Bin Packing problem under the same settings. When using lightweight models such as GPT-4o-mini, our method exhibited significantly better stability and achieved superior performance across all test instances. We plan to add a summary of this comparison in the main text and include a pointer to Appendix H.7 for clarity.
>
> **A3:**  Thank you for your insightful comment. While our differential memory mechanism shares conceptual similarities with the reflection-based evolution in ReEvo, there are key differences in both design and learning strategy. ReEvo employs a pairwise comparison of parent strategies based on short-term memory. In contrast, our approach learns from differences across multiple parents, enabling the discovery of more directional and effective evolutionary strategies. By leveraging the objective values of these strategies, the large language model performs contrastive learning to internalize beneficial search directions. Furthermore, differential memory is integrated into a dual-layer agent architecture, allowing deeper interaction between prompt optimization and strategy generation. We will clarify these differences in Appendix H.7.
>
> **Questions For Authors:**
>
> **A1:** Thank you for the question. Our method adopts a dual-layer self-evolutionary agent, where the outer layer evolves prompt strategies and the inner layer evolves heuristic strategies. In the outer layer, prompt strategies guide the LLM to generate new heuristics through fixed crossover and mutation templates. While these operators are fixed, the prompt strategies themselves evolve dynamically: they are evaluated based on the performance of the heuristics they produce, and low-performing prompts are pruned over time. This design enables adaptive exploration while maintaining diversity and preventing premature convergence.
>
> **A2:**  Thank you for the question. Our method follows a learning-based optimization paradigm, where evolution takes place during the training stage, not inference. Therefore, it does not affect solving efficiency.
>
> Although we introduce an additional layer for prompt strategy evolution, the overall evolution time remains comparable to EOH. The slight overhead mainly comes from evaluating new prompt strategies through testing, which is necessary to compute their fitness.
>
> To provide a concrete comparison, the total evolution time on the 20th generation is:
> - BinPacking: Ours – 144.7 minutes, EOH – 138.7 minutes
> - TSP: Ours – 67.3 minutes, EOH – 59.6 minutes
>
> We believe this small difference is acceptable given the improved performance, and again, it occurs only during training, not during actual problem solving.We will include these results in the appendix to provide a more detailed comparison. Thank you again for the helpful suggestion.
>
> Thank you again to Reviewer G8Pp for the valuable suggestions — they will undoubtedly help us improve the paper.

---

> > ### Comment · Reviewer_G8Pp · 2025-04-02
> >
> > Thank you for the detailed responses. I believe my concerns have been addressed, and I have decided to keep my score.

---

### Decision · Program_Chairs · 2025-05-01

**Decision:**

Accept (spotlight poster)

**Comment:**

This paper proposed a Large Language Model-driven Large Neighborhood Search (LNS) method for solving  Large-Scale MILP Problems. Reviewers acknowledged the non-trivial contribution and promising empirical results. They raised several concerns regarding more realistic testing instances, more baselines for comparison, and discussions w.r.t some existing works. Authors' responses successfully addressed most of these concerns. Based on the uniformly positive evaluations, I recommend acceptance.